# Maximum-likelihood model fitting for quantitative analysis of SMLM data

Yu-Le Wu [1,2], Philipp Hoess [1], Aline Tschanz [1,2], Ulf Matti [1], Markus Mund [1,3] & Jonas Ries [1]✉

Quantitative data analysis is important for any single-molecule localization microscopy (SMLM) workflow to extract biological insights from the coordinates of the single fluorophores. However, current approaches are restricted to simple geometries or require identical structures. Here, we present LocMoFit (Localization Model Fit), an open-source framework to fit an arbitrary model to localization coordinates. It extracts meaningful parameters from individual structures and can select the most suitable model. In addition to analyzing complex, heterogeneous and dynamic structures for in situ structural biology, we demonstrate how LocMoFit can assemble multi-protein distribution maps of six nuclear pore components, calculate single-particle averages without any assumption about geometry or symmetry, and perform a time-resolved reconstruction of the highly dynamic endocytic process from static snapshots. We provide extensive simulation and visualization routines to validate the robustness of LocMoFit and tutorials to enable any user to increase the information content they can extract from their SMLM data.

Single-molecule localization microscopy (SMLM), such as PALM (photo-activated localization microscopy[1]), STORM (stochastic optical reconstruction microscopy[2,3]) or the new MINFLUX[4] technology, enables nanometer optical super-resolution and has widespread applications in cell and structural biology. Because of its molecular specificity and high contrast, it ideally complements electron microscopy for in situ structural biology, that is, the study of the structure or relative arrangement of proteins in the cell. It thus can aid in probing the arrangement of proteins in complexes, even if they are too small or flexible for electron microscopy, and enables the investigation of dynamic and irregular structures (for a review, see ref. [5]). To gain reliable mechanistic understanding from the data, especially when large amounts of data are created using high-throughput SMLM[6–9], a quantitative analysis that can easily scale up is indispensable. The aim of such a quantitative analysis is to inform on the properties of the biological system or to probe functional differences between different conditions with statistical confidence.

In SMLM the primary data are a list of coordinates of fluorophores, often with additional information such as an estimate of the localization uncertainty. The application of standard image analysis algorithms to a rendered pixelated SMLM image is possible but is often limited in performance due to the unique information content in SMLM. Thus, algorithms that directly use these coordinates can exploit the additional information and can produce more accurate and robust results[10]. Many of these approaches have been developed and can be assigned to several classes (reviewed in ref. [10]). First, spatial descriptive statistics[11–13] analyze data based on one-dimensional (1D) profiles without the need for segmenting structures. Second, classification[14,15] assigns class labels to individual segmented structures. Third, geometric analysis includes the fitting of single or double Gaussians to line profiles[16–18], or the fitting of a circle to extract the diameter of ring-shaped structures[19,20]. Last, particle averaging or fusion, an approach extensively used in electron microscopy, yields a final model with improved resolution and signal by registering and averaging hundreds of particles. This approach has been applied in SMLM for averaging[21–26] and for reconstructing 3D averages from 2D images[27,28].

Neither of these approaches reflects the most typical scenario of SMLM data analysis. Usually, some aspects of the geometry underlying

[1]Cell Biology and Biophysics Unit, European Molecular Biology Laboratory (EMBL), Heidelberg, Germany. [2] Collaboration for joint PhD degree between EMBL and Heidelberg University, Faculty of Biosciences, Heidelberg, Germany. [3]Department of Biochemistry, University of Geneva, Geneva, Switzerland. ✉e-mail: Jonas.ries@embl.de

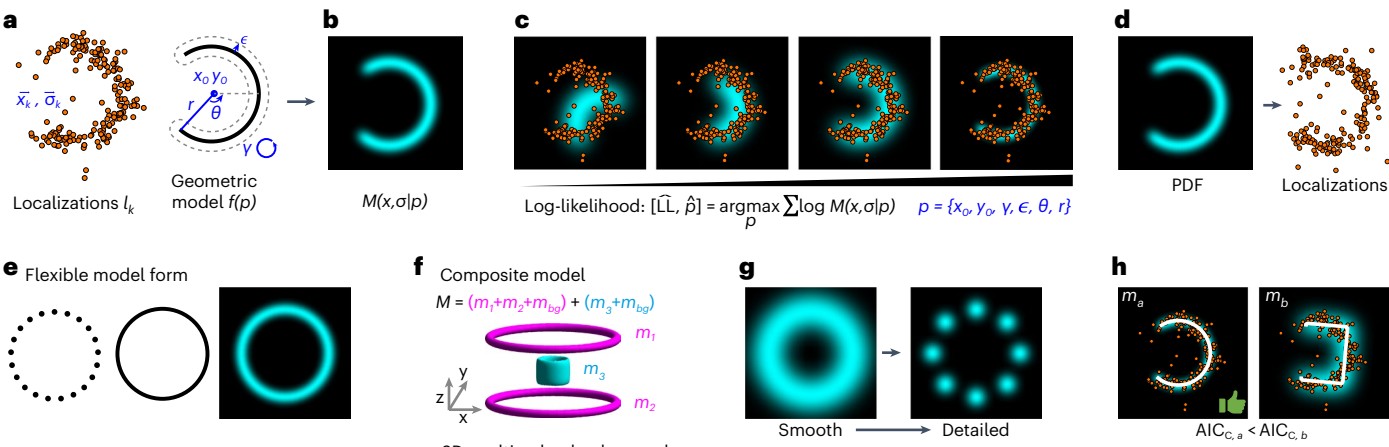

**Fig. 1 | Overview of LocMoFit. a–c**, Workflow of the fitting procedure. **a**, Inputs of LocMoFit are the spatial coordinate $\vec{x}_k$ and the localization precision $\vec{\sigma}_k$ of each localization $k$ and a geometric model $f$ parameterized by parameters $p$. **b**, First, the probability density function (PDF) $M(\vec{x}, \vec{\sigma}\,|\,p)$ of the input model is constructed. **c**, From the model PDF the likelihood of the model describing the data is calculated. A maximum likelihood estimation (MLE) routine searches in the parameter space and maximizes the log-likelihood to find parameter values that best describe the localizations. In the example, a 2D arc model (cyan), parameterized by positions $x_0, y_0$, rotational angle $\gamma$, extra uncertainty $\epsilon$, arc closing angle $\theta$ and radius $r$, is fitted to the single-color data (orange dots). **d–h**, Features of the framework. **d**, Simulation engine for validation. Labels are simulated as samples drawn from the PDF and localizations are then calculated based on fluorophore properties including photon count, re-blinks, and labeling efficiency. **e**, The framework supports flexible model forms including discrete/continuous models and images. **f**, It can assemble complex composite models from simple ones and supports 3D and multi-color data. In the example, the composite model $M$ is formed by combining two ring models ($m_1$ and $m_2$) and one cylindrical model ($m_3$), which are assigned to different channels, represented by different colors. Background models $m_{bg}$ are incorporated channel-wise. **g**, LocMoFit enables multi-step successive optimization to avoid local optima and to find a global optimum. In the example, a smooth, continuous ring model is used to robustly estimate approximate parameters. These are then passed on as initial parameters for a fit with a detailed eight-fold symmetry model with discrete corners. **h**, Model selection. Based on the corrected version of the Akaike information criterion ($AIC_C$) reported by LocMoFit, the model that best describes the data can be selected. In the example, the arc model $m_a$ has a smaller $AIC_C$ than the bucket model $m_b$, indicating that it is a better model for describing the example localizations.

the structure of interest can be inferred from visual inspection of the super-resolution images or from prior knowledge based on structural biology techniques. The data analysis task then consists of first selecting the most likely geometry from a class of possible models, and second, extracting precise parameters describing this geometry. Such analyses would be applicable to individual structures and thus could quantify biological and functional heterogeneities.

To support such a scenario we developed Localization Model Fit (LocMoFit; Fig. 1), a general framework to fit an arbitrary model to coordinate-based SMLM data. It identifies the most likely model from a class of models and estimates the most likely parameters of the model that describe the experimental structure. If the underlying geometry cannot be inferred, LocMoFit can be used for model-free particle averaging to calculate an average model under the assumption of identical structures. The framework also provides advanced visualization routines and a simulation engine, which allow for efficient validation and quality control. LocMoFit is based on maximum likelihood estimation, which is regularly used for fitting data points with distributions[29] and which has been shown to be applicable to SMLM data[30]. LocMoFit, written in MATLAB, has an application programming interface for integration into own code and can be easily extended by user-defined models. Seamless integration in SMAP[31], an open-source super-resolution microscopy analysis platform, provides access to many SMLM tools for localization, post-processing and quantification. Distributed as open source with numerous examples and extensive documentation, LocMoFit will enable many researchers to perform quantitative analysis of their SMLM data with unprecedented efficiency, accuracy and statistical power.

## Results

### Localization Model Fit

With LocMoFit we aimed to create a versatile framework for flexible, quantitative and rigorous analysis of coordinate-based data in SMLM.

Generally, LocMoFit directly analyzes localization point clouds of individual structures. For this, LocMoFit fits a geometric model $f(p)$ to a set of $K$ localizations $l_k = \{\vec{x}_k, \vec{\sigma}_k\}$ (Fig. 1a–c) in a region with a defined boundary that we call 'site', which corresponds to one biological structure or 'particle'. Such a geometric model can be built based on a priori knowledge from diffraction limited images, electron micrographs, or visual inspection of the SMLM images. $l_k$ are obtained by fitting camera images with a model of the point spread function and are described by their coordinates $\vec{x} = \{x, y\}$ and the coordinate uncertainties $\vec{\sigma} = \{\sigma_x, \sigma_y\}$ for 2D data and $\vec{x} = \{x, y, z\}$ and $\vec{\sigma} = \{\sigma_x, \sigma_y, \sigma_z\}$ for 3D data. Conceptually, LocMoFit can be seen as an extension of curve fitting to SMLM point clouds. We demonstrate the workflow using an arc site generated using the simulation functionality of LocMoFit (Fig. 1d). $f(p)$ describes the spatial distribution of the imaged fluorophores and is parameterized by the set of parameters $p$. Our approach is to use maximum likelihood estimation to find the set of parameters $\hat{p}$ that, together with $f(p)$, best describes the measured $l_k$ (Fig. 1c). For this, we first use $f(p)$ to calculate the probability density function (PDF) $M(\vec{x}, \vec{\sigma}\,|\,p)$ that describes the probability that, if we acquire a single localization $l$ with the uncertainty $\vec{\sigma}$ at random, it is found at the coordinate $\vec{x}$. The likelihood of obtaining the set $l_k$ of $K$ localizations in a measurement is then given by the product of individual probabilities:

$$L(p) = \prod_k M(\vec{x}_k, \vec{\sigma}_k\,|\,p). \tag{1}$$

We then use an optimization algorithm to find the parameters $\hat{p}$ that maximize $L(p)$:

$$[\hat{L}, \hat{p}] = \underset{p}{\mathrm{argmax}}\, L(p). \tag{2}$$

$\hat{L}$ denotes the estimate of the maximum likelihood. For efficiency and to prevent a small probability from being rounded to zero, the natural logarithm of the likelihood, the log-likelihood $LL(p)$, is used in practice (Fig. 1c).

The PDF $M$ is constructed from the geometric model $f(p)$. $f(p)$ is defined either in a continuous or a discrete form, or supplied as an image (Fig. 1e). A continuous $f(p)$ describes the shapes formed by the fluorophores such as 1D lines (for example, filaments or rings) or 2D surfaces (spheres, patches), while a discrete $f(p)$ describes the exact fluorophore positions (Methods).

LocMoFit can utilize the characteristics that each localization has its specific lateral and axial localization uncertainties. In this scenario, the model $\vec{v}_j = f(p)$ directly specifies the expected coordinates $\vec{v}_j$ of the in total $J$ fluorophore positions in the model. The likelihood that the localization $\vec{x}_k$ stems from the fluorophore $\vec{v}_j$ is described by a Gaussian function and depends on the distance between $\vec{x}_k$ and $\vec{v}_j$ and the localization precision $\vec{\sigma}_k$. To construct the model $M$ for this single localization, we sum over all model localizations $j$:

$$M(\vec{x}, \vec{\sigma} | p) = \frac{1}{J} \sum_{j=1}^{J} (2\pi)^{-\frac{3}{2}} \det(\Sigma)^{-\frac{1}{2}} \exp\left(-\frac{1}{2}(\vec{x} - \vec{v}_j)^T \Sigma^{-1}(\vec{x} - \vec{v}_j)\right).$$

(3)

$\Sigma = \text{diag}(\sigma_x^2, \sigma_y^2, \sigma_z^2)$ is the diagonal matrix of the square of localization uncertainties and $\det(\Sigma)$ is its determinant.

The parameters $p$ consist of intrinsic parameters $p^i$ that directly determine the shape of the model and extrinsic parameters $p^e$ that describe a rigid transformation and rescaling of the model. $p^e = \{\vec{x}_0, \vec{\alpha}, S, w_{bg}, \epsilon\}$ includes the position of the model $\vec{x}_0$, the orientation, described by the rotation angles $\vec{\alpha}$ around the coordinate axes, an optional global scaling factor $S$, and the proportional weight $w_{bg}$ of a constant background PDF $M_{bg}$ that accommodates the localizations that cannot be described by the geometric PDF. An optional extra uncertainty $\epsilon$ accommodates an uncertainty that cannot be described by the localization precision, such as a linkage error of the fluorophore (for example, due to immunolabeling with primary and secondary antibodies), small-scale deformations of the structure that are not described by the model or residual instabilities (vibrations, drift) of the microscope. From the optimization we obtain the parameter estimates $\hat{p}$ along with the 95% confidence intervals of each fit parameter.

To describe a more complex geometry, a composite model PDF $M_c$ (magenta only in Fig. 1f) can be formed by a linear combination of sub-models $M_m$ that share the same background:

$$M_c(\vec{x}, \vec{\sigma} | p) = \sum_m w_m M_m(\vec{x}, \vec{\sigma} | p_m) + w_{bg} M_{bg},$$

(4)

where the sum of weights $\sum_m w_m + w_{bg} = 1$ for normalization and $p = \{p_m, m = 1 \ldots N\}$ for a total of $N$ component models.

When fitting a composite model to more than one color at a time (for example, both colors in Fig. 1f), the model PDF can be constructed as

$$M_{mc}(\vec{x}_k^c, \vec{\sigma}_k^c | p^{mc}) = \sum_c M_c(\vec{x}_k^c, \vec{\sigma}_k^c | p^c)^{w_c}.$$

(5)

Note that each single-color PDF $M_c(\vec{x}_k^c, \vec{\sigma}_k^c | p^c)$ is evaluated only with the localizations of the corresponding color $c$. $w_c$ is the weight for each color and is by default set to 1 (Methods). Equation (5) is the general form of the model PDF, which can describe a vast class of biological structures. However, LocMoFit is not applicable to random structures that require too many parameters to describe (for example, highly variable topology such as the actin cortex).

To prevent the optimization from becoming stuck in a local maximum of the likelihood, LocMoFit enables the user to chain several fitting steps with different models (usually in the order from smooth to detailed), and pass on the parameter estimates from the previous step to the next one as the initial parameters (Fig. 1g).

Given that the likelihood itself is a measure of the goodness of fit, the model that best describes the data can be identified by comparing the log-likelihood or, more precisely, the corrected version of the Akaike information criterion (AIC$_C$)[32], of different models fitted to the same data (Fig. 1h).

The probabilistic likelihood $L(p)$ used in LocMoFit is closely related to cross-correlation[33,34] and the Bhattacharya cost function[25,26] previously used for SMLM (equation (17), Methods).

## Simulation and validation

We recommend a validation with simulations before applying a fitting pipeline to experimental data to investigate the pipeline's accuracy and robustness under defined experimental conditions. LocMoFit can generate simulated localizations from any model with a comprehensive simulation engine[20,31], using a realistic description of fluorophore blinking, background, labeling efficiency and random displacements of the localizations caused by linkage errors, drifts, and/or vibrations (Fig. 1d and Methods). Fitting these synthetic structures enables comparison of the fitted model parameters to the ground truth.

We systematically investigated how the precision and accuracy of the fit parameters depend on these conditions (Extended Data Fig. 1, Extended Data Fig. 2 and Methods) using the nuclear pore complex (NPC) as an example. Using cryo-electron microscopy the protein Nup96 has been shown to be distributed in two rings per NPC, and to have an eight-fold rotational symmetry with two protein copies per symmetric unit per ring[35]. Based on this prior knowledge, we constructed our detailed model of the NPC (Extended Data Fig. 1a). We simulated the localization data of NPCs with predefined parameters, the ground truth, as given in Supplementary Table 1. We acquired the parameter estimates by fitting the simulated data with the model and computed the errors of the estimates.

As shown in Extended Data Fig. 2, in general, parameter estimations (for example, position, rotation, ring radius or distance) are accurate and precise, indicated by close-to-zero mean errors and small spreads, respectively. This shows that the fitting is unbiased and reliable across a large range of experimental conditions. We found that the spreads of the errors correlate with the localization precisions, which depend on fluorophore brightness (Extended Data Fig. 2). Poorer localization precisions can lead to biases, especially in the extra uncertainty $\epsilon$, which describes the aforementioned random displacement of localizations. Labeling efficiencies that are too low resulted in some NPCs having one ring entirely unlabeled by chance. The remaining single ring is fitted well by the two-ring model with a small separation (Extended Data Fig. 3), leading to a bias towards smaller average ring separation. This highlights the importance of simulations for identifying potential factors to be considered when interpreting results. The other fitting parameters had negligible systematic errors.

In the current framework, multiple fluorophores per target molecule and repeated activation of a single fluorophore are not considered during fitting. Several localizations per molecule, however, do not have a noticeable impact on the accuracy of the parameter estimates (Extended Data Fig. 2c). Future extension to a probabilistic model of fluorophore blinking and non-stoichiometric labeling, possibly using a Bayesian framework[36], could exploit the additional information from multiple localizations per molecule to further improve robustness and accuracy.

To summarize, the simulation function in LocMoFit enables users to easily validate a data analysis workflow given specific experimental parameters and is an important step to ensure robustness.

## Extraction of structural parameters from individual sites

LocMoFit enables determination of the specific and meaningful parameters from individual sites without averaging, which can be used to gain structural insights into multi-protein assemblies and to investigate biological heterogeneity. We demonstrate this on two biological

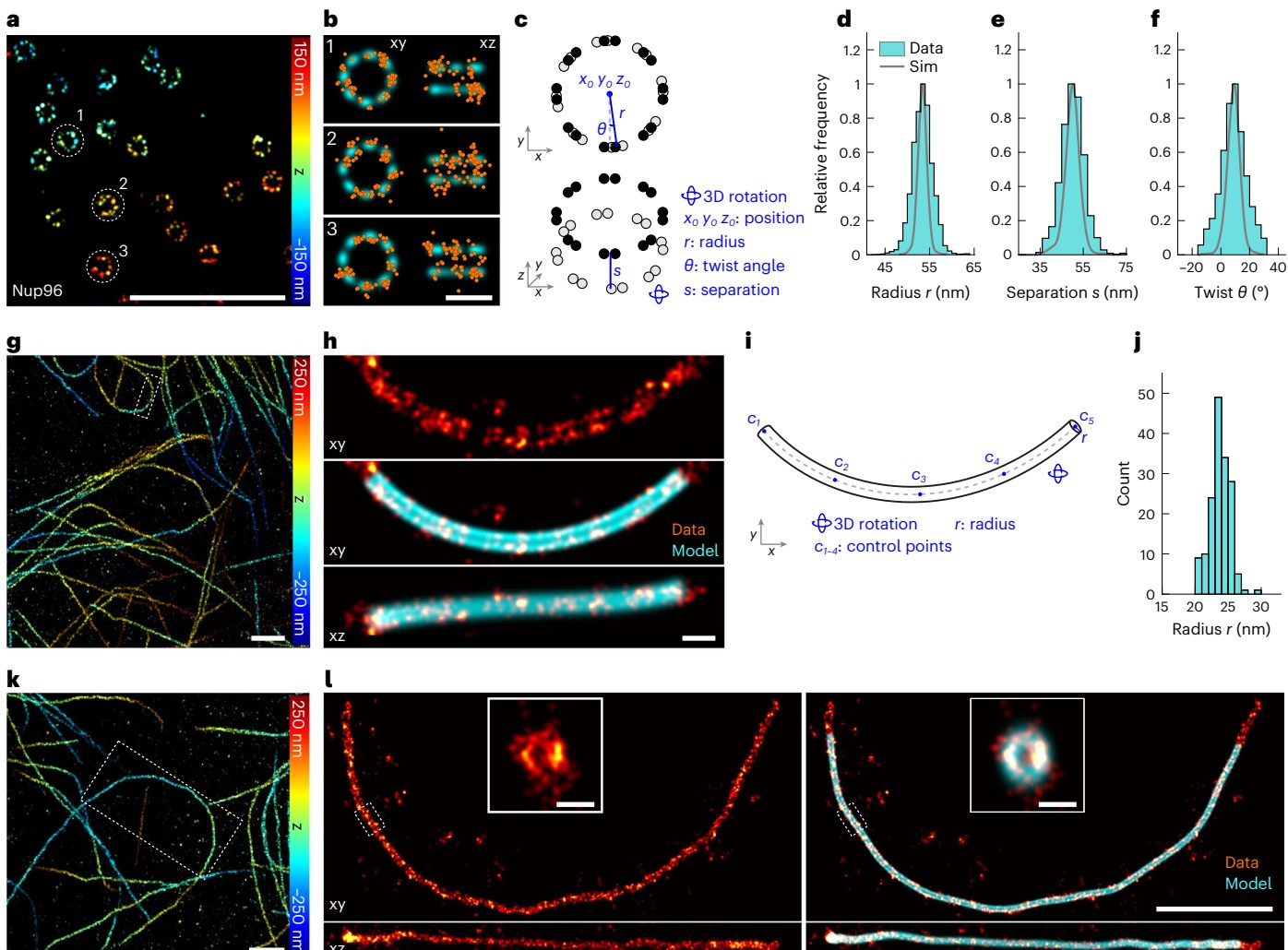

**Fig. 2 | Quantification of individual structures. a–f**, Nuclear pore complexes. **a**, Representative image of Nup96-labeled NPCs (Nup96-SNAP-AF647) in a 3D dataset (top view). **b**, Single NPCs (localizations in orange) as indicated in **a** are fitted with the eight-fold symmetry model (cyan) shown in **c**. The model is parameterized by the listed parameters (blue). **d–f**, Histograms of three fitted parameters: radius $r = 53.4 \pm 2.3$ nm (**d**), separation $s = 50.2 \pm 5.6$ nm (**e**) and twist $\theta = 8.8 \pm 9.0°$ (**f**). Sim, simulated data (gray, see also Extended Data Fig. 5). Simulation parameters are summarized in Supplementary Table 1. Sample size: sites, $n_s = 3{,}517$; cells, $n_c = 5$. $xy$ denotes the top view and $xz$ the side view in all parts of the figure. **g–l**, Microtubules. **g**, Representative image of immunolabeled microtubules in a 3D dataset (top view; original data from Speiser et al.[51]). **h**, One microtubule segment (red) as indicated in **g** is fitted by the linear-tube model (**i**). The fitted model is indicated in cyan. **i**, The linear-tube model parameterized by the listed parameters (blue), the control points $c_i$ define a cubic spline. **j**, Histogram of the fitted radius $r = 23.8 \pm 1.5$ nm, based on segments of 1 μm length. Sample size $n_s = 161$, $n_c = 1$. **k**, Top view of a region containing a 5.2-μm-long non-overlapping (boxed) curved segment (**l**). **l**, The long segment without (left) and with (right) the fitted model overlaid. Insets are the cross-sections of the boxed short segments. Reported values are mean ± s.d., based on $n_s$ sites in a total of $n_c$ cells. Scale bars: **a**,**g**,**k**,**l**, 1 μm; **b**,**h**, 100 nm; insets in **l**, 50 nm.

structures that have been used extensively as reference samples in SMLM: the NPC and microtubules.

We set out to characterize the heterogeneity of the NPC. We imaged Nup96 endogenously tagged with SNAP-tag in a genome-edited cell line[20,37] and obtained hundreds of NPC structures per field of view (Fig. 2a–f). After correcting depth-dependent aberrations[38] (Methods and Extended Data Fig. 4a–e), we fitted individual nuclear pores (Fig. 2b) with the NPC model (Fig. 2c) to extract structural parameters of NPCs: radius $r$ of the rings, separation $s$ and azimuthal 'twist' angle $\theta$ between the rings. The distribution of the single-structure measurements is unimodal for each parameter (Fig. 2d–f). For the mean values ± standard deviations we found $r = 53.4 \pm 2.3$ nm and $\theta = 8.8 \pm 9.0°$, in line with previously reported values based on a similar sample preparation[20]. Our direct quantification of the radius is more accurate than the previously reported value (59.0 nm) based on indirect immunolabeling and rendered 2D images[22], although both works achieved sub-ångström precision. To investigate whether the variation of the parameters is

technical or due to biological heterogeneities, we compared our results to simulations (Extended Data Fig. 5a–e), which included the experimentally measured mean extra uncertainty ($\epsilon = 6.4$ nm) as a random displacement of the localizations. Indeed, we found a larger spread of the experimental parameters (Fig. 2d–f), hinting at biological heterogeneity, that is, that the NPC has a variable size and twist angle on the nanoscale. Similar variabilities have been shown with atomic force microscopy[39] and cryo-electron tomography[40]. However, our model is still an approximation given that it does not describe all possible variations (for example, small local displacements). To assess how additional variations affect the parameter estimations, we fitted simulated elliptical NPCs (Extended Data Fig. 6a), a known deformation in SMLM data[34], with the ring approximation (that is, the model in Fig. 2c). As expected, the magnitude of the errors correlates with the deformation (Extended Data Fig. 6b–e). However, the errors were comparatively small even when the NPCs were visibly deformed, showing that an approximate model is sufficient to extract meaningful parameters. In addition, we

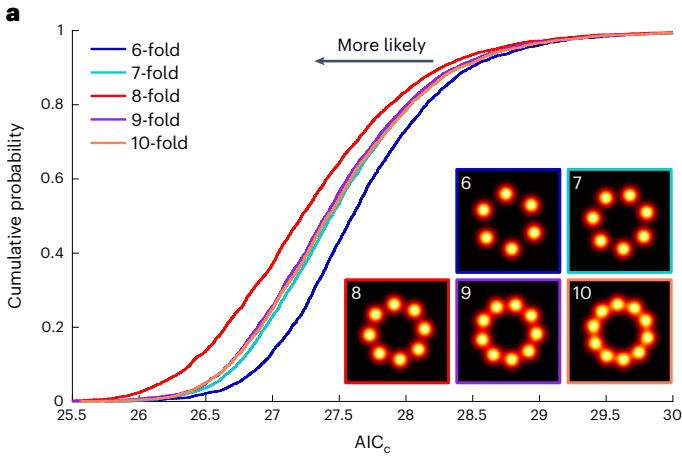

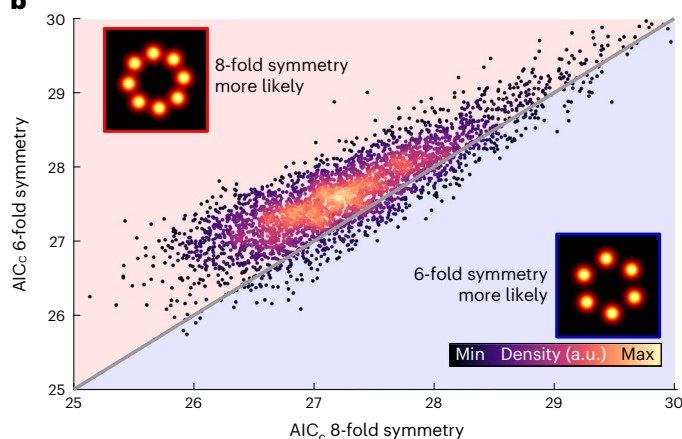

**Fig. 3 | Model selection. a**, Cumulative distribution of the normalized $AIC_C$ acquired by fitting the same experimental NPC dataset with models with different rotational symmetries. The $AIC_C$ was normalized by the number of localizations and assumes lower values for better fits. The (correct) 8-fold symmetry model corresponds to the lowest normalized $AIC_C$. **b**, Scatter plot of $AIC_C$ for fitting the same sites with six-fold and eight-fold rotational symmetry models. The gray diagonal line indicates equal $AIC_C$. Sample size: sites, $n_s$ = 3,517; cells, $n_c$ = 5.

found that the extra uncertainty parameter ϵ can reflect the average deviation between the data and the model and inform how well the data are approximated (Extended Data Fig. 6e).

Next, we demonstrate the analysis of extended structures with LocMoFit using the example of immunolabeled microtubules (Fig. 2g–l). Their apparent radius directly informs on the linkage error induced by the indirect immunolabeling[41] when compared with the true outer radius of microtubules (12.5 nm). Given that microtubules are generally curved, in the past the radius was usually measured only on short segments (less than 500 nm long) using a geometric fit to the cross-sectional profile[16,41], risking a bias from low labeling densities and residual curvature. In LocMoFit we implemented a model that describes a curved tube (Fig. 2i) and thus can trace extended (micrometer-long) curved microtubule segments (Fig. 2h). We measured the radius $r$ of the immunolabeled microtubules as 23.8 ± 1.5 nm, 11.3 nm larger than that of the microtubules themselves, and similar to the reported mean apparent radius of indirectly immunolabeled microtubules[41]. The fit still works for a longer segment (up to 5.2 µm; Fig. 2k,l) but requires long runtimes (~20 hours in this particular case), which has a cubic dependence on the arc length according to simulations (Extended Data Fig. 5g). For efficiency, one can fit different parts of a long segment with the micrometer-long model separately and stitch the results.

### Model selection

Selection of a model that faithfully approximates the biological structure is key to performing a meaningful analysis in LocMoFit. We can use LocMoFit to select the best out of a class of models by comparing the $AIC_C$ (ref. [32]) after fitting. $AIC_C$ is a derivation from maximum likelihood, with a penalty for the number of free parameters $P$ and with a correction for sample size, here the number of localizations $K$ (see Methods)[32]: $AIC_C = AIC + (2P^2 + 2P)/(K - P - 1)$ where $AIC = 2P - 2 \ln \hat{L}$. $\hat{L}$ is the maximum likelihood determined by equation (2). In practice, we would like to choose a model with fewer parameters but with a larger maximum likelihood. Therefore, the smallest $AIC_C$ indicates the best model when fitting the same data. To validate this idea, we fitted different models to each NPC in the Nup96 dataset (Fig. 2a,b). These models were rotationally symmetric with different symmetries (from six-fold to 10-fold, Fig. 3a). The model with eight-fold symmetry clearly has the lowest $AIC_C$ overall, in line with the known symmetry of the NPC[35]. To further validate the model-selection functionality of LocMoFit, we used its simulation engine to generate NPCs with different rotational symmetries.

We show that the cumulative distributions enabled identification of the correct symmetry, given that the matching symmetry always had the lowest $AIC_C$ (Extended Data Fig. 7a). At the single-site level, identification of the correct model is not always possible due to the relatively large variance of the $AIC_C$ (Fig. 3b). Therefore, the $AIC_C$ itself may not rule out all bad fits but can exclude completely wrong models. This is in line with the simulations (Extended Data Fig. 7b), in which a small but noticeable proportion (2%) of eight-fold symmetry NPCs had a lower $AIC_C$ when fitted with a six-fold symmetry model than an eight-fold symmetry model. A different symmetry (for example, the six-fold from the eight-fold), if present, stands out only when it has a comparably large population (Extended Data Fig. 7c).

Model selection can also be applied to investigate features of a more discrete structure. To demonstrate this, we simulated flexible lines consisting of different numbers of segments (Extended Data Fig. 8a–c). LocMoFit was able to recover the precise positions of the clusters along the line segments and to distinguish between the different number of segments using the $AIC_C$ (Extended Data Fig. 8d).

### Multi-color protein distribution maps

Multi-color microscopy is widely used for studying multi-protein assemblies[8,28,42,43]. However, the number of simultaneous labels in the same sample is still a bottleneck because of spectral overlap and the different optimal imaging conditions for different fluorophores, which limits routine multi-color SMLM to two or three colors. Also, the interpretation of hundreds of individual sites is challenging. Here, we show how to overcome this limit using LocMoFit by reconstructing average density maps of multi-protein assemblies from pairs of dual-color data (Fig. 4a–d). In this strategy we use one protein as a reference structure that is always imaged together with a second target protein, labeled in a different color. By fitting the reference protein we can determine the precise location and orientation of each site and thus register all sites within and across individual datasets for different target proteins. Here, we showcase this approach by determining the positions of five proteins in the NPC (Fig. 4c) using Nup96 as the reference (Fig. 4a–d). From a fit of the NPC model (Fig. 2c) to the Nup96 localizations in all datasets and sites we could calculate the average distribution of all target proteins (Fig. 4c) from individual sites (Fig. 4a,b) and integrate all target proteins into a single coordinate system as an average protein distribution map (Fig. 4d and Supplementary Video 1). Note that this approach greatly increases the effective labeling efficiency of the target protein and can produce high-contrast averages even for very

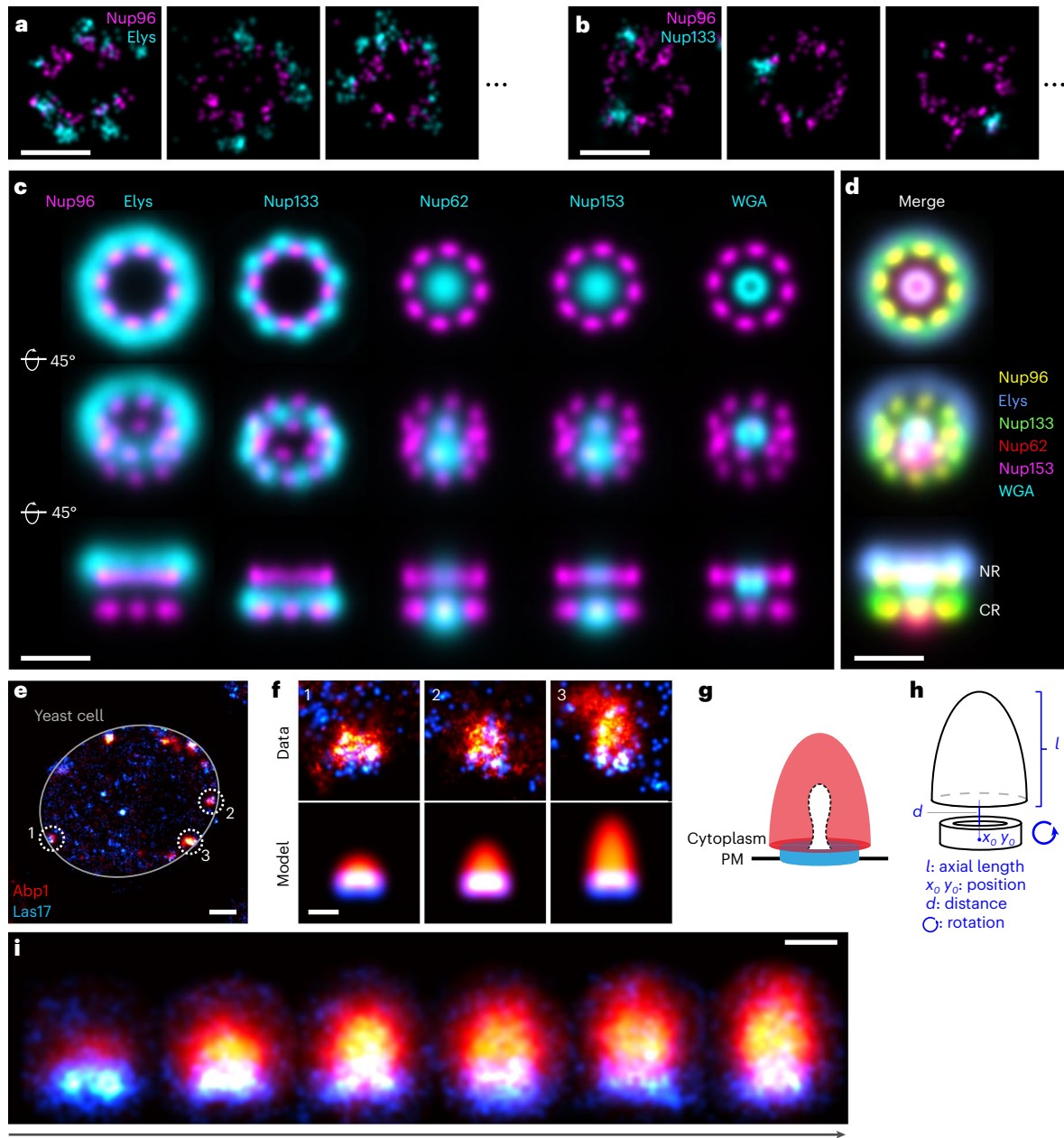

**Fig. 4 | Average protein distribution maps. a–d**, The nuclear pore complex. **a,b**, Representative images of individual sites showing Nup96-SNAP-AF647 and immunolabeled Elys-CF680 (**a**) or Nup133-CF680 (**b**). **c**, A model fit to the reference protein Nup96 enables the registration of all sites of one dataset and integration of the different dual-color datasets into one common coordinate system (**d**). CR, cytoplasmic ring; NR, nucleoplasmic ring. See Supplementary Video 1. Sample size: sites: Elys, $n_s$ = 1,875; Nup133, $n_s$ = 1,739; Nup62, $n_s$ = 2,263; Nup153, $n_s$ = 2,159; WGA, $n_s$ = 1,778; cells: $n_c$ = 3 for all. **e–i**, Dynamic dual-color reconstruction of endocytosis in yeast. **e**, Overview image of a single yeast cell showing Abp1-mMaple and Las17-SNAP-AF647. **f**, Individual endocytic sites are fitted with a dual-color model (**h**) that reflects the expected distribution (**g**) of Abp1 and Las17: we model Abp1 as a hemiellipsoid and Las17 as a thick ring and project these geometries in 2D. The fitted axial length of Abp1 is used as a proxy for pseudotime to sort individual endocytic sites according to their progression along the endocytic timeline. The fitted position and orientation are then used to average all sites in each time bin (**i**). Bin size: 21 sites. Sample size: $n_s$ = 130, $n_c$ = 51. A running average is shown in Supplementary Video 2. Scale bars: **a–d,f,i**, 100 nm; **e**, 500 µm.

poor labeling (compare Fig. 4b with the average), and thus can visualize structural details not apparent in single images. Given that we used a symmetric reference structure, its eight-fold rotational symmetry is transferred to the target proteins and any asymmetry is averaged out. To register target proteins that do not follow this symmetry, an asymmetric reference protein would also be required. Whenever templates are used for registration, the averages can be biased towards the template. This so-called 'template bias' poses a risk of wrongly visualizing structures present in the template that are not present in the particles[44]. This is the reason why we performed the registration

on the reference only, to keep the target structures free of this bias. As in any averaging approach, the underlying particles are required to be identical, otherwise only an averaged distribution is calculated. In the following we illustrate how a classification step can extend this approach to dynamic structures.

## Dynamic reconstruction

Most techniques for in situ structural biology, including SMLM with the highest resolution, are limited in their live-cell compatibility and thus cannot directly measure dynamic structural changes at the nanoscale.

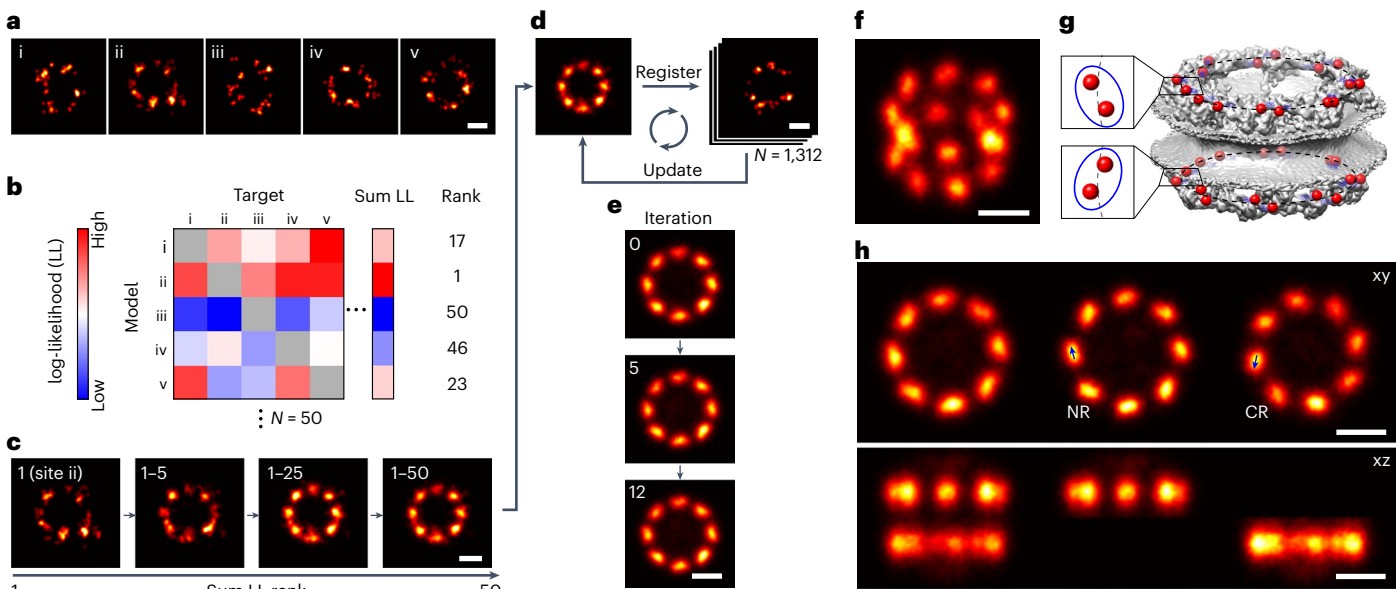

**Fig. 5 | Model-free particle averaging. a–e**, Workflow. **a**, Example NPC particles. We assumed that all sites are samples of the same underlying distribution. **b**, All-versus-all comparison. We first determine the site (in the example, the site ii) that best describes all of the other sites based on the rank on sum LL of the all-to-all matrix, where the 50 subset sites were fitted to each other. **c**, Construction of the initial template. The initial data-driven template is built based on sequential registration in the order of the sum LL rank. **d**, Iterative registration. The final fused particle is used to register all sites in the 1,312-site dataset. This procedure yields an updated fused particle, which is used to register the dataset again.

This process is iterated until it converges (**e**). **f–h**, The final average calculated from 1,312 particles without any assumption on the underlying geometry or symmetry in a tilted view (**f**), and for comparison the electron microscopy density (PDB ID: 5A9Q[35]) of the NPC with the C termini of Nup96 indicated in red (**g**, adapted with permission from ref. [20], Springer Nature). **h**, Top and side view, where the nucleoplasmic and cytoplasmic rings are shown together (left panel), or separately (middle, right panels). The two proteins per ring per symmetry unit give rise to tilted elongated average protein distributions in the averages (arrows in **h**). See Supplementary Video 3. Scale bars: 50 nm.

LocMoFit mitigates this bottleneck for SMLM by enabling a 'dynamic reconstruction', that is, the reconstruction of dynamic rearrangements of multi-protein assemblies based on static super-resolution snapshots taken in fixed cells. The idea is to use LocMoFit to extract features of the structure that can be used for pseudo-temporal sorting and to then average individual structures in each time bin. We illustrate this approach using the example of the machinery that drives clathrin-mediated endocytosis in yeast, which is known to have highly regular dynamics and composition[45]. From prior super-resolution and electron microscopy studies we know that the actin nucleation-promoting factor Las17 forms a ring at the plasma membrane[8] and that the actin-binding protein Abp1 decorates the dome-shaped actin network that elongates during endocytosis[45,46]. By fitting a model that reflects this geometry to dual-color 2D data (Fig. 4e–h), obtained by focusing on the midplane of yeast cells, we use the length of the Abp1 structures to sort all sites according to their progression along the endocytic timeline. We then distribute the structures evenly in individual time bins and use the fitted position and orientation for averaging to result in dynamic protein localization maps (Fig. 4i and Supplementary Video 2). In a recent work[47] we applied a similar analysis to elucidate how the clathrin coat is reshaped during endocytosis in mammalian cells (Extended Data Fig. 9). By geometrically quantifying single clathrin-coated pits, we were able to reconstruct the dynamics of the endocytic clathrin coat from thousands of 3D snapshots[47].

## Model-free averaging

When a structural prior is not available, the distribution map of a protein assembly can still be obtained by model-free averaging or particle fusion. This approach fuses particles that share the same underlying structure to form an average that approximates the underlying structure. Model-free averaging is widespread in electron microscopy[48]. The approach has been introduced to SMLM based on the alignment of particles using pairwise cross-correlations[25,26,33], without or with

adaptation. The adaptation was implemented because of the different data types between SMLM (sparse and coordinate-based) and electron microscopy (dense and intensity-based)[25]. Although this approach alone does not enable quantification of the geometry-specific structural parameters and heterogeneity, the final average can still serve as the basis for the construction of a geometric model.

In LocMoFit we can use individual particles as models for other particles to determine their relative position and orientation and use those in an iterative workflow for model-free particle fusion (Fig. 5). Here, we illustrate this based on Nup96 in the NPC. Given that the log-likelihood is a measure of the similarity, we can efficiently construct the initial template. From an all-against-all pairwise registration of a subset of particles we can identify the particle that has the highest degree of similarity to all other particles as a seed (Fig. 5b). We then cumulatively fuse particles in the order of their total similarity (Fig. 5c), to yield the initial template. These steps minimize the bias of seed selection while avoiding a computationally expensive all-to-all registration applied to the full dataset, as used in a previous study[26]. The initial template is then used to register the remaining particles in the dataset. The resulting average can then be used for the next round of registration (Fig. 5d). This step is iterated until the optimization converges (Fig. 5e).

The resulting 3D average of Nup96 calculated from 1,312 particles clearly resolves the two rings in the NPC and their eight-fold symmetry (Fig. 5f and Supplementary Video 3). In addition, it shows subtle structural details such as the elongated, tilted shape of the corners (Fig. 5h), which indicates that in each ring each symmetric unit is occupied by two Nup96 copies with slightly different radii (Fig. 5g). Previous template-free averages[26] of another nucleoporin (Nup107) with a similar structure seem not to be able to resolve this signature. In our work, as few as ~150 particles are sufficient to obtain a reasonable average (Extended Data Fig. 10a–d), and the labeling efficiency can be as low as 30% (Extended Data Fig. 10e–h).

Model-free averaging does not rely on models and is therefore free from the template bias. However, given that biological variability, such as different conformations, is expected in most experiments, the resulting distributions can be biased towards a sub-population, usually the dominant one. In the future a combination of averaging with classification[34], as we demonstrated when we reconstructed the dynamic protein distribution maps (Fig. 4i), could extend particle averaging to heterogeneous and dynamic cellular structures.

In summary, LocMoFit allows for bias-free high-quality 3D averaging without any assumptions on the underlying geometry and symmetry.

## Discussion

In this study we present LocMoFit, a powerful and general framework for extracting quantitative descriptors of cellular structures by fitting an arbitrary, parameterized model to SMLM data. This single-structure analysis (for example, Fig. 2 and Extended Data Fig. 9) will facilitate the investigation of the vast majority of cellular structures that are heterogeneous and complex. These structures are currently challenging to quantify with classical structural biology techniques such as electron microscopy, in which typically many identical structures are required to be averaged to reach sufficient signal-to-noise ratios. Thus, LocMoFit could be key in enabling SMLM as a complementary method for in situ structural biology.

The integration of large datasets into protein distribution maps can be a useful and complementary approach to a statistical analysis of parameters extracted from individual sites. LocMoFit can calculate such distribution maps by determining the precise position and orientation of a reference structure and use this to align target proteins, imaged in a second channel. By additionally evaluating a parameter that changes monotonically over time, LocMoFit can extend this approach to dynamic, time-resolved localization maps. This novel approach of reconstructing structure and dynamics from snapshots taken in fixed cells can add temporal information to all of the super-resolution technologies that are currently not live-cell compatible. This capability is highlighted by our recent work using LocMoFit to quantify the shape of single clathrin coats and to visualize their structural dynamics[47] (Extended Data Fig. 9). This solved a long-standing controversy about the mechanism of endocytic coat remodeling in mammalian cells.

A reliable image analysis pipeline relies on the choice of correct priors, good data quality and quality control. In the following we will discuss how these factors can improve the robustness of LocMoFit.

LocMoFit depends on the choice of a model that can represent the data. An incorrect model will still result in parameters, but these parameters then might become difficult to interpret or be meaningless. This then prompts the question of how to construct a meaningful model for a biological structure. Usually, a simple geometry or symmetry can be inferred based on visual inspection of the data or from prior knowledge based on other techniques. It is then crucial to define the parameters in a way that ensures that the model is as general as possible and can describe a large class of experimental structures. For instance, the models used in this study are not rigid templates, and their size and shape can be changed during optimization. In the case of competing models, the more likely model can be chosen based on its lower $AIC_C$[32] (Fig. 3). When a structural prior is missing, model-free particle averaging can generate a protein distribution map with the premise of an identical underlying structure. This analysis is also implemented in LocMoFit (Fig. 5a–e), and it enables us to reconstruct a 3D protein distribution map of Nup96 with exceptional quality (Fig. 5e–h) that showed features of individual proteins not visible in previous particle averaging approaches[26].

Of equal importance to the selection of the right model is the quality of the data, which must contain sufficient information to unambiguously define the multiple model parameters. In the case of low labeling densities, large localization errors or structures with few features, simple models with few free parameters have a lower risk of overfitting than complex models. Even a well-chosen model might not converge to the global optimum. In these situations, choosing appropriate initial parameters in a first fitting step with a simpler model, or even manually, can provide a good solution, as well as choosing an optimizer in LocMoFit that performs a parameter search over defined intervals instead of gradient descent.

LocMoFit is equipped with tools to validate the plausibility and robustness of an analysis workflow. One of the tools is visualization, which enables users to efficiently inspect the results of the fit, as we always recommend. Given that it is difficult to quantitatively evaluate a fitting workflow without knowing the ground truth of the data[49,50], LocMoFit provides a simulation engine that generates realistic SMLM coordinate data from a given model and known parameters. This enables investigation of the precision of the parameter estimates, the suitability of a model to fit the data of a specified quality and the impact of initial parameters on convergence. A future extension of LocMoFit to a probabilistic model of repeated fluorophore blinking and non-stoichiometric labeling could further improve robustness and accuracy, and deployment on clusters or graphics processing units could reduce runtimes.

LocMoFit is open source and is readily useable as part of the SMLM software platform SMAP[31], enabling users to easily fit their own data with any of the numerous predefined models using a graphical user interface. To this end, we provide detailed documentation, tutorials and example files. Alternatively, LocMoFit can be run independently of SMAP and provides an application programming interface for integration into own software. All models used in this study are ready to use, are available in the public domain and can be combined into complex composite models. New models can be created with basic programming expertise. We encourage users to deposit their own models into our Git repository to facilitate knowledge sharing.

LocMoFit will enable many researchers to greatly increase the information that can be extracted from their data and to develop new and complex data analysis workflows that drive biological discovery.

## Online content

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

## Methods

### LocMoFit framework

**Model fitting in LocMoFit.** LocMoFit fits a parameterized geometric model to a set of localizations from the same site through maximum likelihood estimation. LocMoFit requires two inputs: a parameterized geometric model $f(p)$ that describes the distribution of the fluorophores in the structure with the set of parameters $p$ of the model, and a set of $K$ localizations $l_k = \{\vec{x}_k, \vec{\sigma}_k\}$, where $\vec{x} = \{x, y, z\}$ are the coordinates of a detected emitter and $\vec{\sigma} = \{\sigma_x, \sigma_y, \sigma_z\}$ are the associated uncertainties. $\vec{x}_k$ and $\vec{\sigma}_k$ are typically obtained by fitting an experimental or Gaussian point spread function (PSF) model to the raw camera frames using maximum likelihood estimation[52,53].

To take into account localization uncertainties, we do not use $f(p)$ directly for fitting but instead we use a probability density function (PDF) $M(\vec{x}, \vec{\sigma}|p)$, which is derived from $f(p)$ as described in the next section. $M(\vec{x}, \vec{\sigma}|p)$ describes the probability of finding a single random localization $l$ at the coordinate $\vec{x}$ given an uncertainty $\vec{\sigma}$ and model parameters $p$.

If we measure a set $l_k$ of $K$ localizations and assume that they are random and independent variables of the PDF $M(\vec{x}, \vec{\sigma}|p)$, the likelihood to obtain precisely these localizations $l_k$ is simply the product of individual probabilities as shown in equation (1).

To find the set of parameters $\hat{p}$ that, together with $M(p)$ and therefore $f(p)$, best describes $l_k$, we maximize this likelihood using an optimization algorithm (see the Optimization procedure section) as shown in equation (2).

**Calculation of the probability density function.** Here, we discuss how to calculate the PDF $M(\vec{x}, \vec{\sigma}|p)$ from the geometric model $f(p)$. $f(p)$ can be defined as either a fluorophore density map, discrete fluorophore coordinates or a continuous fluorophore distribution.

In the first scenario, when defined as a fluorophore density map, the geometric model $d = f(\vec{x}, p)$ directly outputs the density $d$ of the fluorophore at the position $\vec{x}$. Here, $f(\vec{x}, p)$ is not necessarily normalized. Due to a limited localization precision $\vec{\sigma}$, the position of the localization coordinate is not equal to the true position of the fluorophore, but is instead randomly displaced by $\vec{\sigma}$. If the localization uncertainty has been included in $f(p)$, its PDF $M(\vec{x}|p)$ can be derived by simple normalization:

$$M(\vec{x}|p) = \frac{f(\vec{x}, p)}{\iiint f(\vec{x}, p)\,dx\,dy\,dz}. \tag{6}$$

If $f(p)$ does not incorporate the localization uncertainty, it can be used when constructing the PDF by convolving $f(\vec{x}, p)$ with a Gaussian function with standard deviations given by the mean of the localization precision $\langle\vec{\sigma}\rangle$ ($\otimes$ denotes the convolution):

$$M(\vec{x}, \vec{\sigma}|p) = \left[\frac{f(\vec{x}, p)}{\iiint f(\vec{x}, p)\,dx\,dy\,dz}\right] \otimes G(\vec{x}, \langle\vec{\sigma}\rangle). \tag{7}$$

In practice, the model $f(p)$ can be supplied as an image for a 2D fit, an image stack for a 3D fit, or directly as a function.

In the second scenario, when defined as discrete fluorophore coordinates, the geometric model $f(p)$ specifies the expected coordinates $v_j$ of the fluorophores so that $v_j = f(p)$. To derive the PDF for this case let us consider a simple 1D example, in which a fluorophore at position $v$ with a localization precision $\sigma$ is repeatedly localized, resulting in measured coordinates $x_k$. These measured coordinates then scatter around the true position with a standard deviation of $\sigma$, following a Gaussian distribution. Thus, the probability that the measured coordinate $x$ is caused by the fluorophore at position $v$ is[25,26,33]:

$$M(x, \sigma|v) = \frac{1}{\sqrt{2\pi}\sigma} \exp\left(-\frac{(x-v)^2}{2\sigma^2}\right). \tag{8}$$

If we have $J$ model fluorophore positions $\vec{v}_j$, the probability that they describe a single measured localization $l = \{\vec{x}, \vec{\sigma}\}$ is given by the sum of the individual probabilities (now for the 2D or 3D case) as in equation (3). The likelihood function $L(p)$ is then calculated according to equation (1) by multiplying the probabilities of all measured localizations. Compared with the first case (equation (7)), in which only the average localization precision is used to blur the model, here all of the localization precisions $\vec{\sigma}_k$ contribute individually to the PDF so that more precise localizations have a greater impact. Given that the stochastic nature of single-molecule imaging leads to a wide distribution of the localization precisions, this increases the accuracy by properly weighting the single localizations in the PDF.

In the third scenario, when defined as continuous fluorophore distributions, the geometric model in which fluorophores are distributed with constant density on a parametric line (for example, a filament or a ring) is given as $\vec{v} = f(\vec{u}, p)$ and the model to describe a parametric surface (for example, a spherical shell) is given as $\vec{v} = f(\vec{u}_1, \vec{u}_2, p)$. The vector variable $\vec{u}$ or $\vec{u}_1, \vec{u}_2$ parameterizes the line or surface, respectively. In practice, LocMoFit works with a discrete form $f_d(p)$ of the geometric function $f(\vec{u}, p)$. To discretize $f(\vec{u}, p)$, either LocMoFit can render $J$ fluorophores $\vec{v}_j$ on the line or surface based on $J$ vectors $\vec{u}_j$ across the range defined by the user and assign every point a weight $q_j$ inversely scaled to the local density, having $[\vec{v}_j, q_j] = f_d(p) = f(\vec{u}_j, p)$ or, alternatively, the user can define $J$ fluorophores $\vec{v}_j$ evenly distributed on the line or surface defined by $[\vec{v}_j, q_j] = f(p)$, with $q_j = 1$. In either case the maximum spacing $\delta$ between adjacent points defined in $\vec{v}_j$ is required to be smaller than the minimal localization precision of the $K$ localizations to retain continuity: $\delta < 0.75 \min_{k \in \{1...K\}} \{\sigma_{xk}, \sigma_{yk}, \sigma_{zk}\}$ To improve the computational speed by reducing the size of the model, LocMoFit also enables the user to define a minimum localization precision $\sigma_{min}$ so that any $\sigma_{xk}, \sigma_{yk}$ and $\sigma_{zk}$ smaller than $\sigma_{min}$ are set to $\sigma_{min}$. This setting increases the required spacing $\delta$ and reduces the required sampling rate (associated with $J$) and therefore the size of the model. With discrete positions of fluorophores $\vec{v}_j$, the convolution can be seen as placing Gaussian functions centered at all of the positions in $\vec{v}_j$. By having $\vec{v}_j$, we can utilize equation (3) to construct the PDF with the introduction of $q_j$:

$$M(\vec{x}, \vec{\sigma}|p) = \frac{1}{\sum_j q_j} \sum_{j=1}^{J} q_j (2\pi)^{-\frac{3}{2}} \det(\Sigma)^{-\frac{1}{2}} \exp\left(-\frac{1}{2}(\vec{x}-\vec{v}_j)^T \Sigma^{-1}(\vec{x}-\vec{v}_j)\right). \tag{9}$$

Equation (3) is then a special form of equation (9) with $q_j = 1$.

In this study we refer to a discrete model when it is constructed based on either equation (3) or equation (9), and to a continuous model when it is constructed based on either equation (6) or equation (7).

**Optimization procedure.** To find the set of parameters $\hat{p}$ that maximizes $L(p)$, the user can select either an evolutionary algorithm that searches parameters globally, a simplex-based derivative-free searching, or a gradient-descent optimizer. Before optimization the user can define which parameters to fit and which to set to a constant value, and their initial values and boundaries. The initial parameters can be either predefined values, values derived from user-defined rules, or values inherited from a previous fitting step.

For fitting, we classify the parameters $p$ into intrinsic parameters $p^i$ that directly determine the shape of the model, and extrinsic parameters $p^e = \{\vec{x}_0, \vec{\alpha}, \vec{S}, \epsilon, w_{bg}\}$ that describe the position of the model $\vec{x}_0$, the orientation, described by the rotation angles $\vec{\alpha}$ about the three axes, and optionally a global scaling factor $\vec{S}$, an uncertainty $\epsilon$ additional to the localization precision, and the weight $w_{bg}$ of a constant background PDF $M_{bg}$ to accommodate the localizations that cannot be described by the geometric PDF (see the next section). Here, the

rotation angles $\vec{\alpha} = \{\alpha, \beta, \gamma\}$ about the $x$, $y$ and $z$ axes, respectively, define the rotation matrix:

$$R =$$

$$\begin{bmatrix} \cos\beta\cos\gamma & -\cos\beta\sin\gamma & \sin\beta \\ \cos\alpha\sin\gamma + \cos\gamma\sin\alpha\sin\beta & \cos\alpha\cos\gamma - \sin\alpha\sin\beta\sin\gamma & -\cos\beta\sin\alpha \\ \sin\alpha\sin\gamma - \cos\alpha\cos\gamma\sin\beta & \cos\gamma\sin\alpha + \cos\alpha\sin\beta\sin\gamma & \cos\alpha\cos\beta \end{bmatrix}.$$
(10)

This parameterization of $R$ corresponds to the rotations about the $z$, $y$ and $x$ axes subsequently. $\vec{S} = \{s_x, s_y, s_z\}$ contains the scaling factors of the three spatial axes, defining the scaling matrix $S = \text{diag}(\vec{S})$. For a model in the continuous form we use the extrinsic parameters $p^e$ to reverse transform the localizations, which is computationally more efficient than to transform the model. Thus, during the optimization we first transform the localization coordinates as

$$\vec{x}' = R^{-1}S^{-1}(\vec{x} - \vec{x}_0).$$
(11)

For a discrete model we instead translate and rotate the model to avoid computationally costly rotation of the anisotropic multidimensional Gaussian (equation (3)), particularly in 3D. In this case, the fluorophore positions of the model $\vec{v}$ are transformed during optimization as:

$$\vec{v}' = SR\vec{v} + \vec{x}_0.$$
(12)

As a result of maximizing the likelihood with respect to $p^i$ and $p^e$, we obtain the parameter estimates $\hat{p}^i$ and $\hat{p}^e$ along with their 95% confidence intervals, based on the Hessian matrix $H$ estimated by fitting the log-likelihood function LL($p$) with a quadratic form $L_q(p)$ using random parameter values $p$ around the parameter estimates $\hat{p}$ as samples[30], with a fitted constant $a_0$:

$$\text{LL}(p) \approx \text{LL}_q(p) = (p - \hat{p})^T H (p - \hat{p}) + a_0.$$
(13)

The $b^{th}$ diagonal element of the inverse of $-H$ is the estimated variance of the $b^{th}$ fit parameters in $\hat{p}$:

$$\text{var}(\hat{p}_b) = \left[(-H)^{-1}\right]_{b,b}.$$
(14)

The 95% confidence interval of parameter $\hat{p}_b$ is then given as $\text{CI}(\hat{p}_b) = \hat{p}_b \pm 1.96 \times \sqrt{\text{var}(\hat{p}_b)}$.

**Background localizations and additional uncertainties.** In real-world experiments, unspecific background fluorophores, localizations from neighboring structures or large localization errors lead to localizations that are not described by the model. This mismatch can introduce a bias into the parameter estimates. We accommodate these so-called 'background' localizations with an evenly distributed (constant) PDF $M_{bg}$:

$$M_b(\vec{x}, \vec{\sigma}|p) = (1 - w_{bg})M(\vec{x}, \vec{\sigma}|p_m) + w_{bg}M_{bg}.$$
(15)

The set of parameters $p_m$ contains all elements of $p$ except for the background weight $w_{bg}$. $M_{bg} = d^{-D}$ where $d$ is the length of a site and $D$ is the dimension, so that the summed probability of $M_{bg}$ over the site is 1. $w_{bg}$ is the background weight that represents the fraction of localizations that are considered background. The total number $K_{bg}$ and density $\rho_{bg}$ of the background localizations can be obtained as $K_{bg} = K \cdot w_{bg}$ and $\rho_{bg} = K_{bg}/d^2$, respectively. In LocMoFit the user can choose whether to use the density $\rho_{bg}$ or the weight $w_{bg}$ as the fitting parameter. The difference between the total number of localizations $K$ and $K_{bg}$ is then the total number of localizations described by the model $K_m = K - K_{bg}$.

The localization precision $\sigma$ often underestimates the true spread of localizations in real-world experiments. The reason can be instabilities such as drifts or vibrations during the experiment, the size of the label that displaces the fluorophore from the target structure (linkage error), or biological variability that leads to a spread of the fluorophores that is not described in the model. These additional uncertainties, quantified by the parameter $\epsilon$, lead to an additional blurring (equation 7), with $\langle\sigma\rangle^2 \to \langle\sigma\rangle^2 + \epsilon^2$. In equation (3) we take $\epsilon$ into account with a modified covariance matrix:

$$\Sigma = \text{diag}\left(\sigma_x^2 + \epsilon^2, \sigma_y^2 + \epsilon^2, \sigma_z^2 + \epsilon^2\right).$$
(16)

$\epsilon$ can be specified by the user or used as an additional free-fitting parameter.

**Composite models.** LocMoFit enables the user to combine several simple models into a single one by adding up and re-normalizing the PDFs of each model (equation 4).

The sum of weights is 1: $\sum_m w_m + w_{bg} = 1$. $w_m$ represents the proportion of the localizations that can be described by the component PDF $M_m$. Using the weights we can estimate the number of localizations $K_m$ coming from a specific component model $M_m$ by $K_m = K \cdot w_m$.

Note that here we define the extrinsic parameters $p_m^e$ (except for the model weight $w_m$) of the $m^{th}$ component model ($m > 1$) with respect to the first component model, with a value of zero indicating the same transformation as the first component model. That is, the rigid transformation of the first component model (according to $p_1^e$) is first applied to all component models, followed by the rigid transformation of the $m^{th}$ component model (according to $p_m^e$) applied to only the $m^{th}$ component model.

The user can select which parameters are fixed in the models and which are fitted independently. This greatly facilitates the construction of complex models.

When fitting multi-color SMLM data, each localization is not only described by its coordinate and localization precision, but also by its color $c$. In this case we can define a separate model for each color channel and fit all of the models simultaneously, as shown in equation (5).

The weight for each color channel $w_c$ is introduced to minimize the effects of different numbers of localizations between different colors and can be assigned as $w_c = K_c^{-1}\sum_c K_c$, where $K_c$ is the number of localizations with the color $c$. $w_c$ is used as an exponent to normalize the different multiplications, which scales to the number of localizations, in equation (1). When the effects of different numbers of localizations are preferred, weighting can be switched off by setting $w_c = 1$. Note that each single-color PDF $M_c(\vec{x}_k^c, \vec{\sigma}_k^c|p^c)$ (with the background PDF $M_{bg}$, as described by equation (4) for an individual model or equation (5) for a composite model) is evaluated only with the localizations of the corresponding color.

**Chaining fitting steps for improved convergence.** For complex models with many fitting parameters, optimizers are limited when scanning the parameter space to find a global optimum and might become stuck in a local maximum of the likelihood. Thus, LocMoFit enables the user to chain several fitting steps with different models and to use the results of the previous step as the initial parameter for the next one. Note that the first step can involve user-defined rules and/or functions to provide initial parameter estimates. Then, the user can use a less complex model with strong blur (equation (16)), using a global optimizer before finetuning the fit with a simplex or gradient-descent optimizer on the precise model. In this way, LocMoFit efficiently finds the global maximum of the cost function $L(p)$.

**Computational complexity.** Maximum likelihood estimation fitting requires extensive computation. In our implementation, a fit to a single site takes seconds to minutes (5–10 s for an NPC and clathrin-mediated endocytosis site, 10–20 min for a micrometer-long microtubule, depending on the complexity of the model and the number of localizations of a site), enabling even large datasets with hundreds of sites to be analyzed in overnight runs on a standard central processing unit (for example, Intel Core i5-4460). In the future, deploying LocMoFit on clusters or graphics processing units could further improve performance.

**Likelihood and cross-correlation.** The likelihood $L(p)$ can be seen as a metric that describes the similarity between model $f(p)$ and data $l_k$ from the probabilistic aspect. By changing the multiplication in equation (1) to summation, we obtain another metric that is regularly used for pattern matching and represents the cross-correlation between model and data:

$$L_{cc}(p) = \sum_k M(\vec{x}_k, \vec{\sigma}_k | p). \tag{17}$$

When using a model $f(p)$ in the discrete form, by plugging its PDF (as in equation (3)) into equation (17), we obtain a similar form to the correlation between two sets of points derived by Schnitzbauer et al.[33], with the exception that we do not assign uncertainties to fluorophore coordinates in the model. Also, it is closely related to the Bhattacharya cost function and to its derivatives that were previously used for particle fusion[25,26] and for detecting structural heterogeneity[34] in SMLM. Therefore, the cross-correlation $L_{cc}(p)$ can also be used as the objective function in LocMoFit.

## Data analysis

**Models and fitting.** Model fitting requires segmented sites (see the section Segmentation of sites).

**Nup96.** We used three models to describe Nup96 in different fitting steps. The first model, $NPC_{m1}$, is a composite model of two identical rings, with a fixed radius, shifted along their common axis. The extrinsic parameters of the upper ring were fixed to those of the lower ring, except for the $z$ position. This model was implemented as a fluorophore density map. The second model, $NPC_{m2}$, is a dual-ring model that has two identical parallel rings, parameterized by the intrinsic parameters ring radius $r$ and ring separation $s$. This model was implemented as a discretized continuous fluorophore distribution. The third model, $NPC_{m3}$, was built using $NPC_{m2}$ as a backbone, with the continuous rings replaced by discrete fluorophore positions (Fig. 2c). Two of the fluorophores form a unit, which is evenly placed eight times on one ring rotationally to yield 32 positions in total. Given that the rings are no longer continuous, the twist $\theta$ between the two rings is also an intrinsic parameter, in addition to the two parameters inherited from the second model.

For simulating elliptical NPCs, the model $NPC_e$ was built as a derivative from $NPC_{m3}$. In $NPC_e$, the intrinsic parameter ring radius $r$ represents the average axes of the ellipse. A new intrinsic parameter, ellipticity $e$, which determines the lengths of the long and short axes $a$ and $b$, was incorporated so that $e = 1 - b/a$. Another new intrinsic parameter is the internal rotation angle $\Phi$, which is the rotational offset between the long axis and the first corner.

For fitting single-color NPC data (Fig. 2a–f) and elliptical NPCs (Extended Data Fig. 6), we chained these three fitting steps: first, fitting with $NPC_{m1}$ to approximately measure the orientations, positions and ring separations of the NPCs; second, fitting with $NPC_{m2}$ to refine the previously measured parameters and to measure radii; and third, fitting with $NPC_{m3}$ to measure the ring twist. In the last two steps, the extra uncertainty $\epsilon$ is defined as a free parameter to enable exploration of the parameter space during optimization. The initial parameters of a later step are inherited from the final parameters of the previous step. All parameter settings are summarized in Supplementary Table 2.

For fitting dual-color NPC data (Fig. 4a–d), Nup96 was fitted in two chained steps: first, fitting with $NPC_{m1}$ as for the single-color data, and second, fitting with $NPC_{m3}$, with intrinsic parameters fixed to the mean parameter values that were extracted from the single-color data (Fig. 2d–f). All parameter settings are summarized in Supplementary Table 3.

For the model selection, the fitting steps are the same as for single-color NPC data except that the different rotational symmetries were used as specified in Fig. 3a.

**Microtubules.** We used two models to describe microtubules. The first model, $MT_{m1}$, describes a cubic spline in 3D. In this model, the spline is defined as piece-wise third-order polynomials that traverse through a set of odd number $N$ of equidistant control points, in the order $q = 1$ to $N$. The middle point ($q = q_0 = (N+1)/2$) is defined as the reference position $\vec{x}_{mid} = \{x_{mid}, y_{mid}, z_{mid}\}$. Starting from the middle point, the rest of the control points are defined in two directions, one from $q = q_0 - 1$ to 1 and the other from $q = q_0 + 1$ to $N$. Following these orders, the position of one control point ($q = q_n$) is defined by its distance $h$ from the previous control point and the azimuth $\theta_q$ and elevation angle $\varphi_q$, defined relative to the previous control point. The second model, $MT_{m2}$, uses the first model as a backbone, rendering rings, centered at equidistant points on the backbone spline, perpendicular to the backbone (Fig. 2i). Thus, the radius $r$ of the rings is an intrinsic parameter in addition to the ones inherited from the first model. Both models were implemented as discretized continuous fluorophore distributions. In this study, unless specified otherwise, we used the number of control points $N = 5$ and the distance between points $h = 250$ nm for micrometer-long segments, and $N = 27$, $h = 200$ nm for the 5.2 µm segment.

For simulating the long curved microtubule (Extended Data Fig. 5f–h), the tubular model $MT_{m2}$ was used with the positions $\vec{x}_n = \{x_n, y_n, z_n\}$ of the control points $q_n$ directly specified.

Microtubule segments were fitted with two chained steps. For the segments used in Fig. 2g–j the first step involved fitting with $MT_{m1}$, with a large free extra uncertainty $\epsilon$ to estimate the central line of microtubule segments, and the second step involved fitting with $MT_{m2}$ to refine the path of the microtubules and to measure the radius. The initial parameters of the second step were inherited from the final parameters of the first step. For the segments used in Fig. 2k,l and Extended Data Fig. 5f–h, one additional intermediate between the two steps was included to further refine the path by fitting with $MT_{m1}$ again with a smaller free extra uncertainty $\epsilon$. All parameter settings are summarized in Supplementary Table 4 and Supplementary Table 5.

**Endocytic structures.** For fitting endocytic sites, we used a composite model formed by a two-component model: projections of a 3D hemiellipsoid and a thick ring onto the 2D imaging plane (Fig. 4g). This model was implemented as a discretized continuous fluorophore distribution. In the imaging plane, the base of the hemiellipsoidal projection is limited to below the thick-ring projection. The hemiellipsoidal projection is parameterized by the half long and half short axes $a$ and $b$ of a hemiellipsoid. The thick-ring projection is parameterized by the thickness $t$ and the inner and outer radii $r$ and $q$ of the ring. This model was fitted to the yeast endocytic sites in the dual-color dataset (Fig. 4e–i). The hemiellipsoid was fitted only to the localizations in the mMaple channel (Abp1) and the thick ring to the localizations in the AF647 channel (Las17). All parameters are summarized in Supplementary Table 6.

**Line segments.** The model of line segments LS is implemented as a discrete model, the model points of which are the vertices of connected line segments (Extended Data Fig. 8c). The positions of these vertices are defined in the same way as the control points in $MT_{m1}$. The model was fitted in only one step using a global optimizer. All parameter settings are summarized in Supplementary Table 7.

**Simulation.** We performed realistic simulations based on a two-state (bright and dark) fluorophore model plus bleaching[54]. First, we defined model parameters, which can be fixed numbers or uniformly distributed random variables within specified boundaries. Second, using the defined model parameters we generated protein positions for each simulated site by taking all of the $N$ positions (for example, 32 positions for the eight-fold symmetry model of the NPC) of proteins defined in a point model or randomly drawn $N$ samples from a specified PDF with no uncertainty. Third, with a probability $p_{label}$, a fluorescent label was created at a protein position. Fourth, an extra uncertainty was introduced by

adding random displacements in $x$, $y$ and $z$ to localizations as normally distributed random variables. The source of the uncertainty includes, for example, linkage error, drift and vibration. Fifth, each fluorophore appeared at a random time and lived for a time $t_i$, determined as a random variable from an exponential distribution. Sixth, a label had a probability $p_{react}$ to be reactivated and then appeared at a random later time point, otherwise it was bleached. Seventh, when it was on, a fluorophore had a constant brightness. Thus, the brightness in each frame was proportional to the fraction of the time in which the fluorophore was on in each frame. Eighth, the emitted photons in each frame were determined as a random Poisson variable with a mean value corresponding to the average brightness in the frame. Ninth, for each frame we calculated the CRLB (Cramér–Rao lower bound) in $x$, $y$ and $z$ from the number of photons and the background photons based on a 3D cubic spline PSF model derived from bead calibrations[53]. And last, this error was added to the true $x$, $y$ and $z$ positions of the fluorophores as normally distributed random values with a variance corresponding to the respective calculated CRLB. Simulation parameters are summarized in Supplementary Table 1.

The simulated localizations were processed with the same data analysis pipeline as the experimental data.

**Reference-based averaging of multi-color data.** To create the average density map of the NPC, in each site only Nup96 localizations were fitted, as described in Model fitting in LocMoFit. Each site was transformed to the orientation and position of the model so that all of the sites were in the same coordinate system. The averages were reconstructed from the transformed localizations of all of the sites.

A technical limitation of this example is the use of indirect immunolabeling. Here, varying epitope accessibility and non-random orientation of the antibodies can result in systematic differences between protein distribution maps and true distributions of the proteins, which in principle can be overcome with improved labeling schemes.

For the dynamic reconstruction of clathrin-mediated endocytosis in yeast, all of the sites were sorted by the fitted length of the hemiellipsoid describing Abp1 localizations. The orientation of each site was aligned to the direction of the membrane invagination, and the estimated position of the Las17 ring model defined the origin. Each time bin was then created from the localizations of 21 aligned sites. The movie of the dynamic reconstruction (Supplementary Video 3) was generated by a moving average across the aligned sites over pseudotime: each frame comprised 15 sites and the step size was 1 site.

**Model selection.** In LocMoFit we provide $AIC_C$ (ref. [32]) as the metric for performing model selection. In general, a model with more free parameters tends to fit better. Therefore, instead of using the maximum likelihood $\hat{L}$ as the metric, $AIC = 2P - 2\ln\hat{L}$ was suggested for penalizing the number of free parameters $P$ (ref. [32]). In practice we would like to choose a model with fewer parameters and which also has a larger maximum likelihood. Therefore, the smallest AIC indicates the best model when fitting the same data. To avoid overfitting caused by small sample size, $AIC_C$ includes an additional penalty: $AIC_C = AIC + (2P^2 + 2P)/(K - P - 1)$, where $K$ is the sample size[32]. When $K \to \infty$, the additional penalty term approaches zero so that $AIC_C$ converges to AIC. In LocMoFit the sample size $K$ is the number of localizations. For visual comparison of $AIC_C$ we normalize by the number of localizations $K$.

**Model-free averaging.** For model-free averaging of Nup96 particles, we generated an initial model from a subset of $n$ particles $P_1 \dots P_n$, $n = 50$. To this end, we defined the localization coordinates of each particle as the fluorophore positions of a point model and fitted each model to all other particles in the 50-particle subset. Prior to that all-against-all pairwise fitting, we optionally set the initial parameters of positions and rotations according to the fit with a continuous dual-ring model. This step narrows the search range of rotations during the pairwise fitting and reduces the tendency to form bright 'hot spots' or overlaps

of denser corners that have been enhanced by a wide search range of rotations. Next, based on LL values acquired by all-against-all pairwise fitting, we then built a similarity matrix $M$. Next, we cumulatively fused the particles in the order $R$ of their total similarity: each particle $P_{[R=i]}$ was registered to the fused particle $T_{[R=i-1]}$ starting with the highest-ranked particle $T_1 = P_{[R=1]}$. This initial model $T_{[R=i-1]}$ was then used to register the remaining particles $P_{51} \dots P_k$ in the $k$-particle dataset. The resulting average was then used as the new template $\tilde{T}$ for the next round of registration. This step was iterated until convergence and yielded the final average $T$. See Algorithm 1 for the pseudocode.

**Algorithm 1. Model-free averaging**

*Input*: $k$ individual particles $P_1 P_2 \dots P_k$, each contains localization coordinates and uncertainties
*Output*: final average $T$
Procedure:

 // all-against-all pairwise registration among an $n$-sites subset of particles with $n < k$
 // $k$ is the total number of particles in the dataset
 **for** each pair $\{i,j\} \in \{1 \dots n\}$
 **if** $i \neq j$
 // *LocMoFit*($A$, $B$) represents fitting $A$ to $B$ through ***LocMoFit***
 matrix $M[i,j] \leftarrow$ maximum log-likelihood of ***LocMoFit***$(P_i, P_j)$
 **else**
 $M[i,j] \leftarrow 0$
 **end if**
 **end for**
 vector $R \leftarrow$ rank(rowsum($M$))

 // forming the first data-driven template
 $T \leftarrow P_{[R=1]}$ // taking the particle with the highest total similarity as the seed
 **for** $i = 2 \dots n$
 $P' \leftarrow P_{[R=i]}$ registered to $T$ through ***LocMoFit***$(T, P_{[R=i]})$
 $T \leftarrow P' \cup T$
 **end for**

 // iterative optimization of the average particle until no further improvement
 // the optimization stops when $J$ unimproved iterations reached
 $S \leftarrow -inf$ // initializing the current best score $S$
 $\tilde{T} \leftarrow T$ // use the current $T$ as the initial template for the iterative registrations
 **repeat**
 **for** $i = 1 \dots k$
 $L_i \leftarrow$ maximum log-likelihood of ***LocMoFit***$(\tilde{T}, P_i)$
 $P'_i \leftarrow P_i$ registered to $T$ through ***LocMoFit***$(\tilde{T}, P_i)$
 **end for**
 $\tilde{S} \leftarrow$ sum($L_1 \dots L_k$) // the current score
 $\tilde{T} \leftarrow P'_1 \cup P'_2 \dots P'_k$ // $\tilde{T}$ is the template for the next iteration
 **if** $\tilde{S} > S$ // if there is an improvement
 $T \leftarrow \tilde{T}$ // $T$ is the current best average
 $S \leftarrow \tilde{S}$
 $j \leftarrow 0$
 **else** // if there is no improvement
 $j \leftarrow j + 1$
 **end if**
 **until** $j = J$ // stops when no improvement for $J$ consecutive times
 **return** $T$ as the final average

## Sample preparation

**Preparation of coverslips.** The 24 mm round glass coverslips were cleaned overnight in stirring methanol/hydrochloric acid (50/50). They were then rinsed repeatedly with Milli-Q water until the pH of the washing solution remained neutral. They were then placed overnight in a laminar flow cell culture hood to dry before being sterilized by ultraviolet irradiation for 30 min.

For yeast samples, the coverslips were subsequently plasma cleaned for 5–10 min. A drop of 20 µl concanavalin A (ConA) solution (4 mg ml$^{-1}$ in PBS) was added to each coverslip, spread out with a pipette tip, and left to incubate for 30 min in a humidified atmosphere. Then, the remaining liquid was removed and the coverslips were dried overnight at 37 °C. Prior to use, the remaining salts were washed off with Milli-Q water.

**Sample seeding.** Cells were seeded on clean glass coverslips 2 days before fixation to reach a confluency of ~50–70% on the day of fixation. They were grown in growth medium (DMEM; catalog no. 11880-02, Gibco) containing 1× MEM NEAA (catalog no. 11140-035, Gibco), 1× GlutaMAX (catalog no. 35050-038, Gibco) and 10% [v/v] fetal bovine serum (catalog no. 10270-106, Gibco) for approximately 2 days at 37 °C and 5% $CO_2$. Before further processing, the growth medium was aspirated and samples were rinsed twice with PBS to remove dead cells and debris.

**Imaging buffers.** Yeast samples were mounted in $D_2O$ blinking buffer (50 mM Tris-HCl pH 8, 10 mM NaCl, 100 U ml$^{-1}$ glucose oxidase, 0.004% [w/v] catalase, 10% [w/v] D-glucose, 20 mM cysteamine, in 90% $D_2O$).

NPC samples were imaged in 50 mM Tris-HCl pH 8, 10 mM NaCl, 100 U ml$^{-1}$ glucose oxidase, 0.004% [w/v] catalase, 10% [w/v] D-glucose and 35 mM cysteamine.

**Preparation of NPC samples.** For single-color imaging, coverslips containing Nup96-SNAP-tag cells (catalog no. 300444, CLS Cell Line Service) were rinsed twice with warm PBS. Prefixation was carried out in a 2.4% [w/v] formaldehyde in PBS solution for 40 s before the samples were permeabilized in 0.4% [v/v] Triton X-100 in PBS for 3 min. Complete fixation was carried out in 2.4% [w/v] formaldehyde in PBS for 30 min followed by three 5 min washing steps in PBS after fixation. Subsequently, the sample was incubated for 30 min with Image-iT FX Signal Enhancer (catalog no. I36933, Thermo Fisher Scientific) before staining with SNAP dye buffer (1 µM BG-AF647 (catalog no. S9136S, New England Biolabs) and 1 µM dithiothreitol in 0.5% [w/v] BSA in PBS) for 2 h at room temperature. To remove unbound dye, coverslips were washed three times for 5 min in PBS. At this point, the sample was ready for single-color super-resolution imaging.

For simultaneous dual-color imaging with immunostaining, samples were further blocked with 5% [v/v] normal goat serum (NGS) (catalog no. PCN5000, lifeTech) in PBS for 1 h. Binding of primary antibody (Elys (catalog no. HPA031658, Atlas Antibodies, 1:50), Nup133 (catalog no. HPA059767, Atlas Antibodies, 1:150), Nup62 (catalog no. 610498, BD Biosciences, 1:150), Nup153 (catalog no. ab24700, Abcam, 1:60)) was achieved by incubation with the respective antibody diluted in 5% [v/v] NGS in PBS for 1 h. Coverslips were washed three times for 5 min with PBS to remove unbound antibody and subsequently stained with CF660C- or CF680-labeled anti-rabbit antibody (catalog no. 20813/no. 20818, Biotium) or anti-mouse antibody (catalog no. 20815/no. 20819, Biotium) diluted 1:150 in PBS containing 5% [v/v] NGS for 1 h. After three washes with PBS for 5 min each, the sample was postfixed for 30 min using 2.4% [w/v] formaldehyde in PBS, rinsed with PBS, quenched in 100 mM NH$_4$Cl for 5 min and washed three times for 5 min with PBS.

For simultaneous dual-color imaging with wheat germ agglutinin (WGA) staining, cells on a coverslip were fixed, permeabilized and stained with SNAP dye as described above. The sample was then incubated for 10 min with 400 ng ml$^{-1}$ WGA-CF680 (catalog no. 29029-1,

Biotium) in 100 mM Tris pH 8.0, 40 mM NaCl, and rinsed three times with PBS.

Before imaging, samples were mounted on a custom sample holder in appropriate imaging buffers (see Imaging buffers). The holder was sealed with parafilm.

**Strain and sample preparation for yeast.** The yeast strain expressing Abp1 tagged with mMaple[55] and Las17 tagged with SNAP$_f$tag[56] has been described previously (JRY0014; ref. [8]). In brief, the two proteins were tagged at their carboxy termini at the endogenous loci[57]. The strain was verified using colony polymerase chain reaction and fluorescence microscopy.

Prior to the day of imaging, yeast cells were inoculated from single colonies on plates into 10 ml YPAD (yeast-extract peptone adenine dextrose: 1% [w/v] yeast extract, 2% [w/v] bacto peptone, 0.004% [w/v] adenine hemisulfate, 2% [w/v] D-glucose in Milli-Q water) medium in a glass flask, and grown overnight at 30 °C with shaking. The next morning, the culture was diluted into 10 ml YPAD medium in a glass flask to an optical density at 600 nm (OD$_{600}$) of 0.25, and grown for 3 more hours at 30 °C, typically reaching an OD$_{600}$ of 0.6–1.0.

For sample preparation, 2 ml of the culture were collected by centrifugation at 500 ×$g$ for 3 min, resuspended in 100–150 µl YPAD and pipetted onto a ConA-coated coverslip. During all following incubation steps, the samples were protected from light. The cells were allowed to settle for 15 min in a humidified atmosphere. Next, the coverslip was directly transferred into the freshly prepared fixation solution (4% [w/v] formaldehyde, 2% [w/v] sucrose in PBS). After 15 min of fixation with gentle orbital shaking, the sample was quenched in 100 mM NH$_4$Cl in PBS for 15 min. Quenching was repeated one more time before the coverslips was washed once in PBS for 5 min. Next, cells were permeabilized for 30 min by addition of the permeabilization solution (0.25% [v/v] Triton X-100, 50% [v/v] Image-iT FX, in PBS). The coverslip was washed twice in PBS for 5 min and then transferred face down onto a drop of 100 µl staining solution (1 µM SNAP Surface Alexa Fluor 647, 1% [w/v] BSA, 1 mM dithiothreitol, 0.25% [v/v] Triton X-100, in PBS) on parafilm. After staining for 90 min, the sample was washed three times in PBS for 5 min each.

## Microscopy

**Microscope setup and imaging.** All SMLM data of the NPC in mammalian cells were acquired on a custom-built widefield setup described previously[8,58]. In brief, the free output of a commercial laser box (Light-Hub, Omicron-Laserage Laserprodukte) equipped with Luxx 405, 488 and 638 and Cobolt 561 lasers and an additional 640 nm booster laser (iBeam Smart, Toptica) were collimated and focused onto a speckle reducer (catalog no. LSR-3005-17S-VIS, Optotune, Dietikon) before being coupled into a multi-mode fiber (catalog no. M105L02S-A, Thorlabs). The output of the fiber was magnified by an achromatic lens and was guided through a laser cleanup filter (390/482/563/640 HC Quad, AHF) to remove fluorescence generated by the fiber. Before being focused into the sample to homogeneously illuminate an area of ~1,000 µm$^2$, the beam was reflected into the high numerical aperture (NA) oil immersion objective (HCX PL APO 160×/1.43 NA, Leica) by a dichroic mirror (TIRF (total internal reflection fluorescence) Quad Line Beamsplitter, zt405/488/561/640rpc, Chroma). A cylindrical lens (f = 1,000 mm; catalog no. LJ1516L1-A, Thorlabs) was used to introduce astigmatism for 3D SMLM. Emitted fluorescence was collected through the objective, filtered by a 700/100 bandpass filter (catalog no. ET700/100m, Chroma) and imaged onto an Evolve512D EMCCD (electron multiplication charge-coupled device) camera (Photometrics). For the filter setup for dual-color imaging, see below (Ratiometric dual-color SMLM). The $z$ focus was stabilized by an infrared laser that was totally internally reflected off the coverslip onto a quadrant photodiode, which was coupled into closed-loop feedback with the piezo objective positioner (Physik Instrumente). Laser control, focus

stabilization and movement of the filters were performed using a field-programmable gate array (Mojo, Embedded Micro). The pulse length of the 405 nm laser (laser intensity ≈ 28 W cm⁻²) was controlled by a feedback algorithm to sustain a predefined number of localizations per frame. The microscope was controlled by µManager[59] through the Easier Micro-Manager User interface (EMU[60]). Typical acquisition parameters are ~100,000 frames, a frame rate of 100 ms, and a laser intensity of 6 kW cm⁻² as a good compromise between localization precision and imaging time[37]. Samples were mounted and imaged until almost all of the fluorophores were bleached and no further localizations were detected under continuous ultraviolet irradiation.

**Pixel size calibration.** The effective pixel size of the microscope was calibrated by translating fluorescent beads, immobilized on a coverslip, with a calibrated sample stage (SmarAct) that operated in a closed loop. From the measured translation of many beads the pixel size could be calibrated with a high accuracy.

**Ratiometric dual-color SMLM.** For ratiometric dual-color imaging of AF647 and CF680, the emitted fluorescence was split by a 665LP dichroic mirror (catalog no. ET665lp, Chroma), filtered by a 685/70 (catalog no. ET685/70m, Chroma) bandpass filter (transmitted light) or a 676/37 (catalog no. FF01-676/37-25, Semrock) bandpass filter (reflected light) and imaged side by side on the EMCCD camera. The color of the individual blinks was assigned by calculating the ratio of the intensities in the two channels.

**Dual-color SMLM in yeast.** The yeast dual-color data were acquired on a microscope with a commercial laser box (iChrome MLE, Toptica) with 405 nm, 561 nm and 640 nm lasers and a 640 nm booster laser (Toptica), which were coupled via single mode. The output of the fiber was collimated, focused on the back focal plane of the TIRF objective (60×/NA 1.49, Nikon), and adjusted for epi illumination. The emitted fluorescence was laterally constricted by a slit, split by a dichroic mirror (640LP, ZT640rdc, Chroma), filtered by the respective bandpass filters (transmitted/AF647: 676/37, FF01-676/37-25, Semrock; reflected/mMaple: 600/60, NC458462, Chroma), and imaged on two parts of the EMCCD camera (iXON Ultra, Andor). The focus was stabilized as described for the system above. Raw data were acquired with a 30 ms exposure time. The images acquired in the two channels were merged using a transformation that was determined using images of beads that are fluorescent in both channels (TetraSpeck).

## Data processing
SMLM data analysis was conducted using previously published algorithms with custom software written in MATLAB (super-resolution microscopy analysis platform, SMAP[31]), available as open source at github.com/jries/SMAP.

**3D bead calibration.** TetraSpeck beads (0.75 µl from stock, catalog no. T7279, Thermo Fisher) were diluted in 360 µl Milli-Q water, mixed with 40 µl 1 M MgCl₂ and put on a coverslip in a custom-manufactured sample holder. After 10 min, the mix was replaced with 400 µl Milli-Q water. Using Micro-Manager, approximately 20 positions on the coverslip were defined and the beads were imaged, with z stacks acquired (−1 to 1 µm, 10 nm step size) using the same filters as used in the intended experiment.

**Fitting and post-processing.** Two-dimensional data were fitted with a symmetric Gaussian PSF model with the PSF size, $x$, $y$, photons per localization and the background as free-fitting parameters using maximum likelihood estimation[53]. 3D data were fitted using an experimentally derived PSF model from the 3D bead calibration with $x$, $y$, $z$, photons per localization, and the background as free-fitting parameters using maximum likelihood estimation[53].

Fitted data were first grouped by merging localizations persistent over consecutive frames within 35 nm from each other (with an allowed gap of one dark frame) into one localization with its position calculated by the weighted average of individual $x$, $y$ and $z$ positions. Photons per localization as well as the background were summed over all frames in which the grouped localization was detected. Data were then drift corrected in $x$, $y$ and $z$ by a custom algorithm based on redundant cross-correlation. From the spread of the redundant displacements we estimated the accuracy of the drift correction to be better than 1.5 nm in $x$ and $y$ and 2 nm in $z$.

To exclude bad fits and to reject molecules far away from the focal plane, the filtering was applied depending on the type of data.

3D data of Nup96 were filtered based on lateral localization precision ([0,5] nm), $z$ position (boundaries defined to exclude localizations away from the nuclear envelope), log-likelihood (lower boundary defined to exclude the left tail of the distribution), and frames (boundaries defined to exclude the approximately 1,000 very first and last frames).

3D data of microtubules were filtered based on lateral localization precision: ([0,5] nm) and frames ([30,000, 90,000], also for efficiency).

3D dual-color data of NPCs were filtered based on lateral localization precision ([0,10] nm for Nup96 and [0,5] nm for target proteins), log-likelihood (lower boundary defined to exclude the left tail of the distribution), and frames (boundaries defined to exclude the ~1,000 very first and last frames).

2D dual-color data of endocytic sites in yeast were filtered based on localization precision ([0,25] nm), PSF size ([0,175] nm) and frames (boundaries defined to exclude the ~20,000 very first and last frames).

**Segmentation of sites.** All NPC images used in this work that are based on Nup96-derived data were segmented automatically in SMAP according to a previously published workflow[20]. For this, reconstructed images were convolved with a kernel consisting of a ring with a radius corresponding to the approximate radius of the NPC, convolved with a Gaussian. Local maxima over a user-defined threshold were treated as possible candidates. Candidates were cleaned up in three additional steps. In the first step, we fitted the localizations corresponding to each candidate with a circle and excluded structures with a ring radius smaller than 40 nm or larger than 70 nm. In the second step, the localizations were refitted with a circle of fixed radius to determine the center coordinates. Structures were rejected if more than 25% of localizations were closer than 40 nm to the center or if more than 40% of localizations were further away than 70 nm from the center, because these typically did not visually resemble NPCs or were two adjacent wrongly segmented NPCs. In the third step, sites with less than 30 localizations were removed to ensure sufficient sampling of the underlying biological structure. (Example low-quality NPCs are given in Extended Data Fig. 4f and their effects on the variations of parameters are shown in Extended Data Fig. 4g–i).

In images of microtubules, a circular boundary with a radius of 500 nm was used to crop microtubules into sites to obtain segments that were at least 1 µm long. In a site with more than one microtubule, a polygon mask was used to further retain only one segment of interest.

Endocytic sites in yeast were manually picked and rotated so that the direction of the invagination was pointing upwards.

**Correction of depth-dependent distortions.** We observed a depth-dependent distortion along the $z$ axis, as reported previously[38]. The distortion is reflected by the depth-dependent ring separations $s$ of NPCs (Extended Data Fig. 4a). Given that we previously measured the precise ring separation for Nup96 (ref. [20]), we used it as the standard to correct the distortion. By definition, the ring separation is the distance between the two rings of one NPC so that $s = |\vec{x}_{r1} - \vec{x}_{r2}|$, where $\vec{x}_{r1}$ and $\vec{x}_{r2}$ are the center positions of the two rings, respectively. Given that the orientation of an NPC is not necessary perpendicular to the $x$–$y$ plane, we measured the tilt angle radian of an NPC from the $z$ axis

as $\psi = \tan^{-1}\left(\left(1 - R_{3,3}^2\right)^{1/2}/R_{3,3}\right)$ where $R$ is defined by equation (10). We used this angle to derive the vertical component of the separation as $s_z = s\cos\psi = z_{r1} - z_{r2}$. We also calculated the expected $s_z$ as $E(s_z) = E(s)\cos\psi$, with $E(s)$ defined as 49.3 nm, the previously determined average separation of the Nup96 rings[20]. With these values, we can calculate for each NPC a scaling factor $s_f(z) = E(s_z)/s_z$. We found that the moving median of $s_z$ along the $z$ axis appeared as a quadratic-like curve. We then fitted a quadratic function $s_z = c_1 z^2 + c_2 z + c_3$ to the data. Given that the correction factor represents the change of the expected $z$ position over the change of measured $z$ position, $s_f(z) \approx \partial E(z)/\partial z$. We then defined $z_0$, which makes $s_f(z) = 1$, as the origin of distortion. The expected or undistorted $z$ position can then be acquired as $E(z) = \int s_f(z)dz$ with $E(z_0) = z_0$. The corrected $z$ position of each localization $k$ was then defined as $z'_k = E(z_k) - E(0)$ to keep the focal point zero. This correction was applied to all of the NPC datasets before further quantification.

### Reporting summary

Further information on research design is available in the Nature Portfolio Reporting Summary linked to this article.

### Data availability

All image data used in this study are available on the BioImage Archive data repository (accession number: S-BIAD563). Source data are provided with this paper.

### Code availability

Source code for LocMoFit (v1.1) is contained in Supplementary Software 1 and updated versions can be freely downloaded at https://github.com/jries/SMAP/tree/develop/LocMoFit. The software is documented at https://locmofit.readthedocs.io where detailed tutorials are also available.

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

### Acknowledgements

The authors thank I. Schoen and J.-K. Hériché for input into the manuscript, and I. Čavka, M. Theiss, S. Liu, M. Skruzny, L.-R. Mueller, T. Deguchi, C. Heidebrecht, S. Mojiri, S. Azizi, L. Voos, A. Zubas and T. Noordzij for feedback on the software and tutorials. This work was supported by the European Research Council (grant no. ERC CoG-724489 to J.R.), the National Institutes of Health Common Fund 4D Nucleome Program (grant no. U01 EB021223 to J.R.), the Human Frontier Science Program (grant no. RGY0065/2017 to J.R.) and the European Molecular Biology Laboratory.

### Author contributions

J.R. and Y.-L.W. conceived the approach, developed the methods and wrote the software. Y.-L.W., A.T., P.H., J.R. and M.M. tested the software. U.M., P.H., A.T. and M.M. acquired the data. Y.-L.W., P.H. and U.M. analyzed the data. Y.-L.W. and J.R. wrote the manuscript with input from all of the authors.

### Competing interests

The authors declare no competing interests.

### Additional information

**Extended data** are available for this paper at https://doi.org/10.1038/s41592-022-01676-z.

**Correspondence and requests for materials** should be addressed to Jonas Ries.

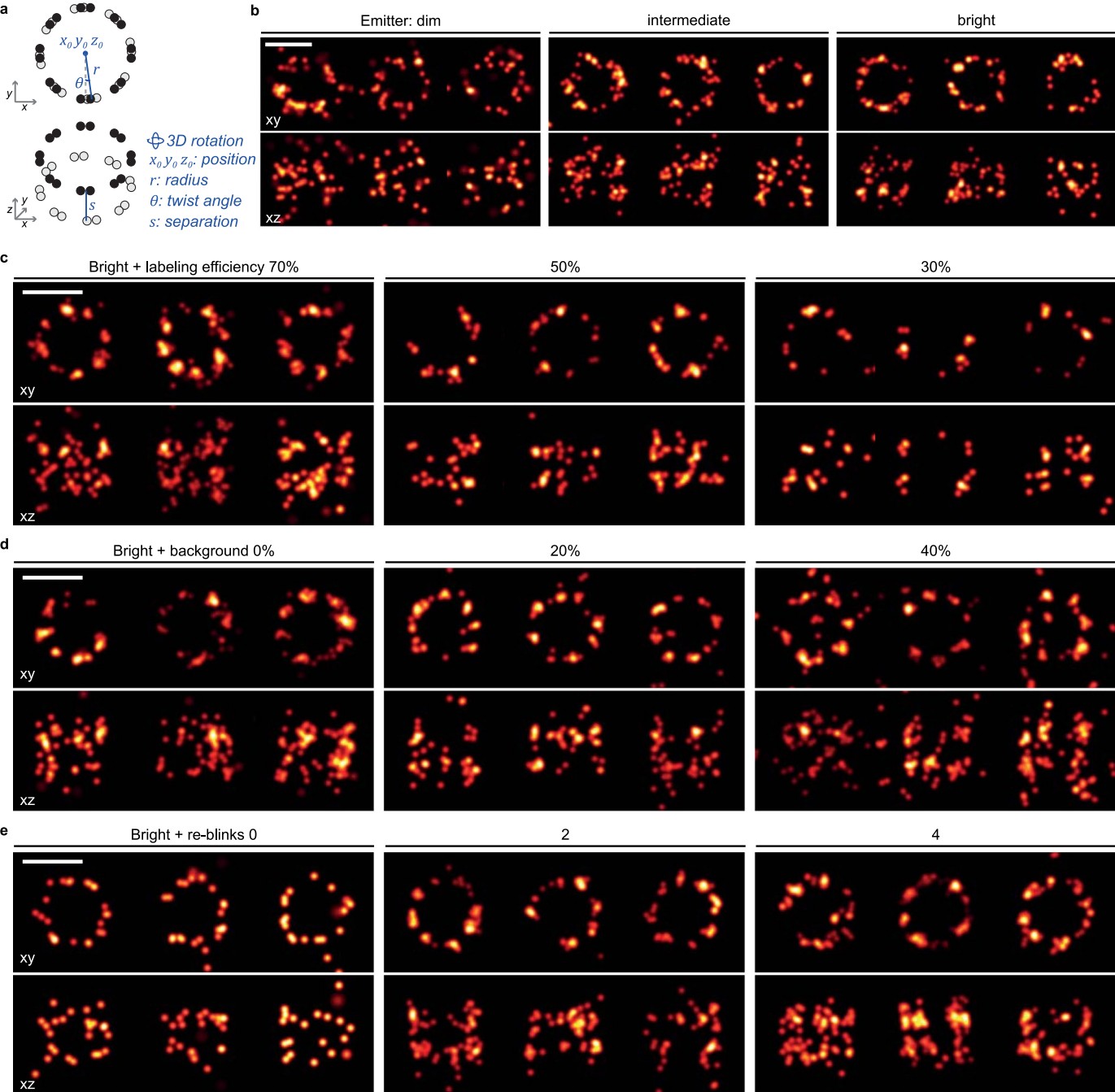

**Extended Data Fig. 1 | Example sites simulated across various conditions.**
**a**, The nuclear pore complex (NPC) model used for simulations. **b**–**e**, Example simulated NPCs across different brightness of fluorophores (**b**), labeling efficiency (**c**), background localizations (**d**), and re-blinks (**e**). In **b**, corresponding photon counts $n_{ph}$ and fluorescence background $bg$ in photons/pixel/localization were used (dim: $n_{ph}$=1,500, $bg$=30; intermediate: $n_{ph}$=5,000, $bg$=100; bright: $n_{ph}$=15,000, $bg$=300). The three conditions yielded median

localization precisions of 15.5, 8.0, and 4.6 nm, respectively. Unless otherwise indicated, we show the gallery of NPCs simulated using the following simulation parameters: photon count 15,000 and 300 background photons/pixel/ localization (corresponding to median lateral localization precision of 4.6 nm), labeling efficiency 60%, 10% background localizations, and 2 re-blinks. Detailed simulation parameters are listed in Supplementary Table 1. Scale bars: 100 nm.

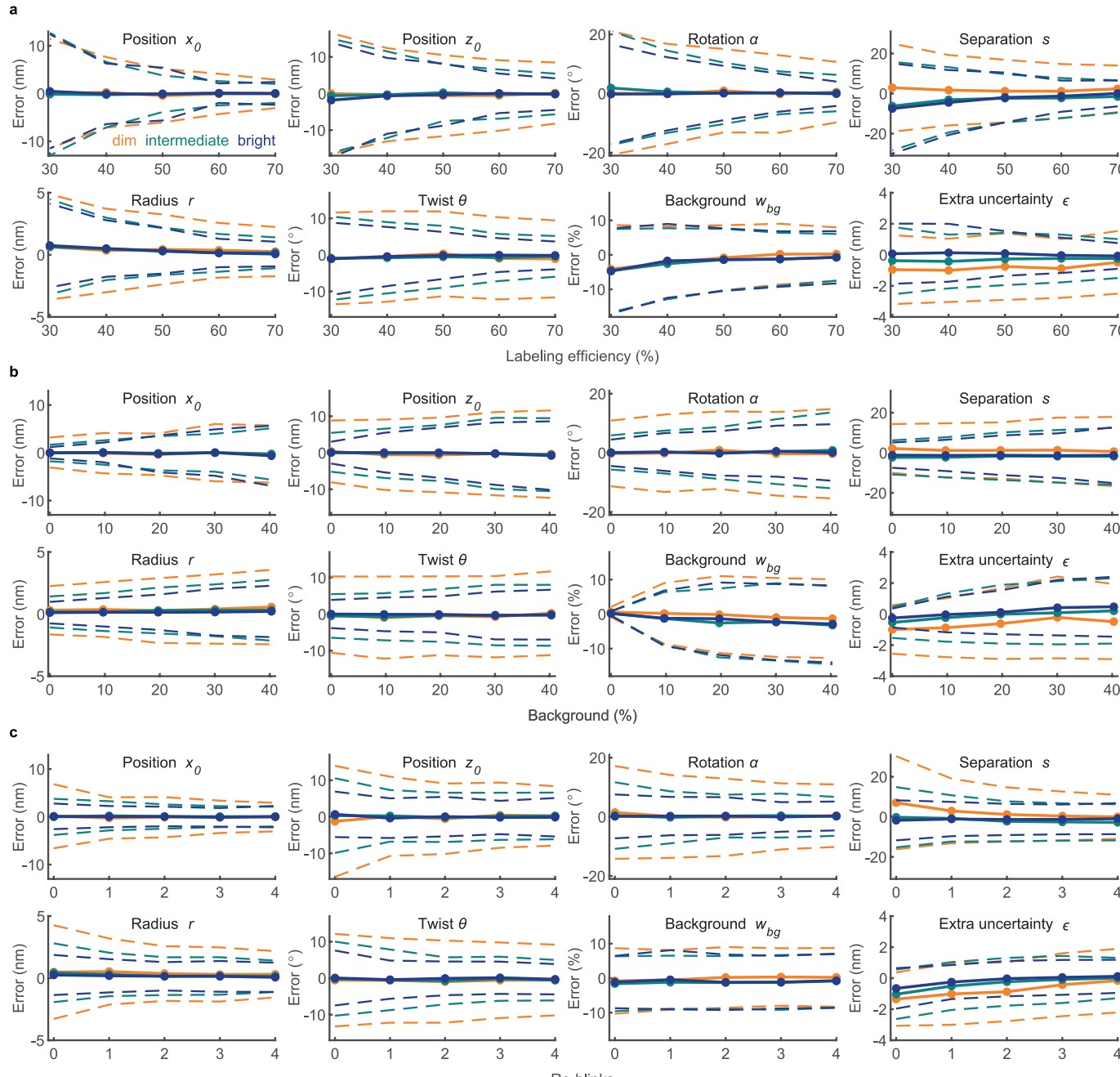

**Extended Data Fig. 2 | Errors of parameter estimations against factors contributing to data quality in the simulated datasets.** Each panel shows the estimation errors of one parameter with three different brightness of fluorophores (dim, intermediate, and bright), across different levels of either, labeling efficiency (**a**), background (**b**), or re-blinks (**c**). Example sites of some conditions are shown in Extended Data Fig. 1. The dots indicate mean errors (data value – ground truth value), which correspond to accuracy, and dashed lines indicate the standard deviations, which correspond to precision. Simulation parameters are listed in Supplementary Table 1. Sample size: $n_s$ = 1,000 for each dot.

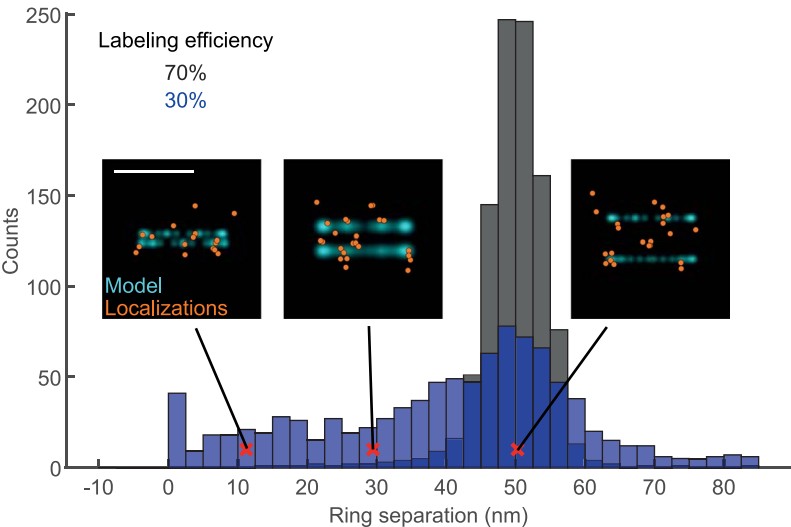

**Extended Data Fig. 3 | Distributions of the fitted ring separation at different labeling efficiencies in simulated data.** Three side-view examples of pores at different fitted ring separations, indicated by red crosses. All examples are from the dataset simulated with 30% labeling efficiency. Sample size: $n_s = 1,000$ for each condition. Scale bar: 100 nm.

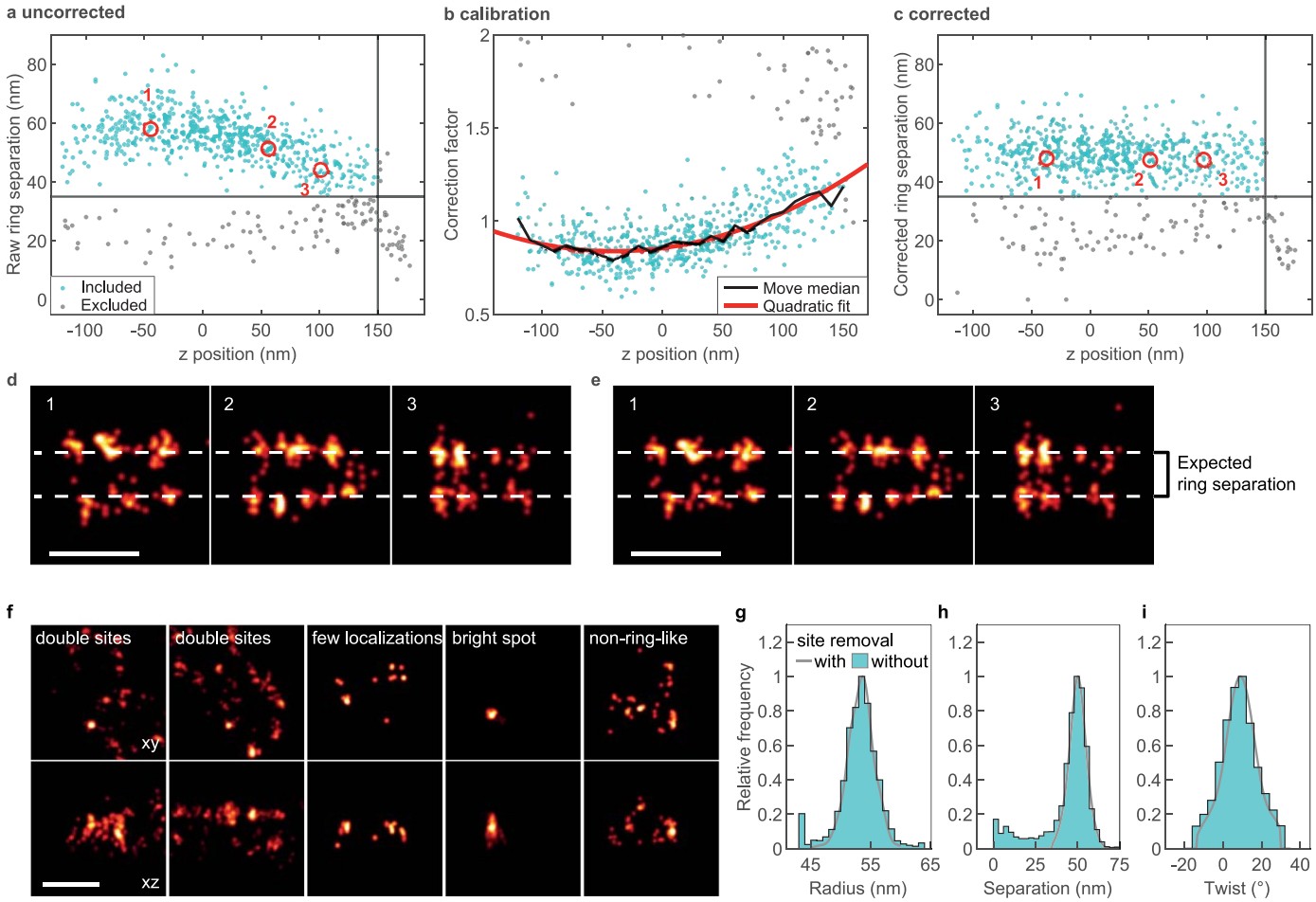

**Extended Data Fig. 4 | Data pre-processing and quality control of NPC data.**
**a**, A scatter plot showing the $z$-dependent spatial distortion along the $z$ axis, as the nuclear pore complex (NPC) ring separation appears to be dependent on the $z$ position. **b**, The median correction factor over the $z$ position can be approximated by a quadratic fit. The correction factor is defined as the expected ring separation divided by a fitted ring separation before the correction. **c**, Fitted ring separations over the $z$ positions after the correction based on the correction factor. NPCs with ring separations smaller than 35 nm (gray horizontal lines, likely stemming from NPCs with only a single ring labeled) and with $z$ positions further than 150 nm away from the focus (gray vertical lines) were excluded from

the curve fit and the following analysis. Each point represents one NPC. All the data points in **a**–**c** are from the same field of view. Side-view examples of sites at different $z$ positions before (**d**) and after (**e**) the correction, corresponding to the data points indicated by the numbered red circles in **a** and **c**. **f-i, Impact of removing low-quality or one-ring NPCs. f**, Example low-quality sites. **g-i**, Histograms of three fitted parameters with and without the removal. Statistics without the removal: radius $r = 53.1 \pm 3.3$ nm (**g**), separation $s = 43.7 \pm 15.3$ nm (**h**), and twist $\theta = 8.1 \pm 9.6°$ (**i**). Our workflow (see Methods) removed 1,712 (low quality: 602; single-ring: 1,110) out of 5,211 sites. Scale bars: 100 nm.

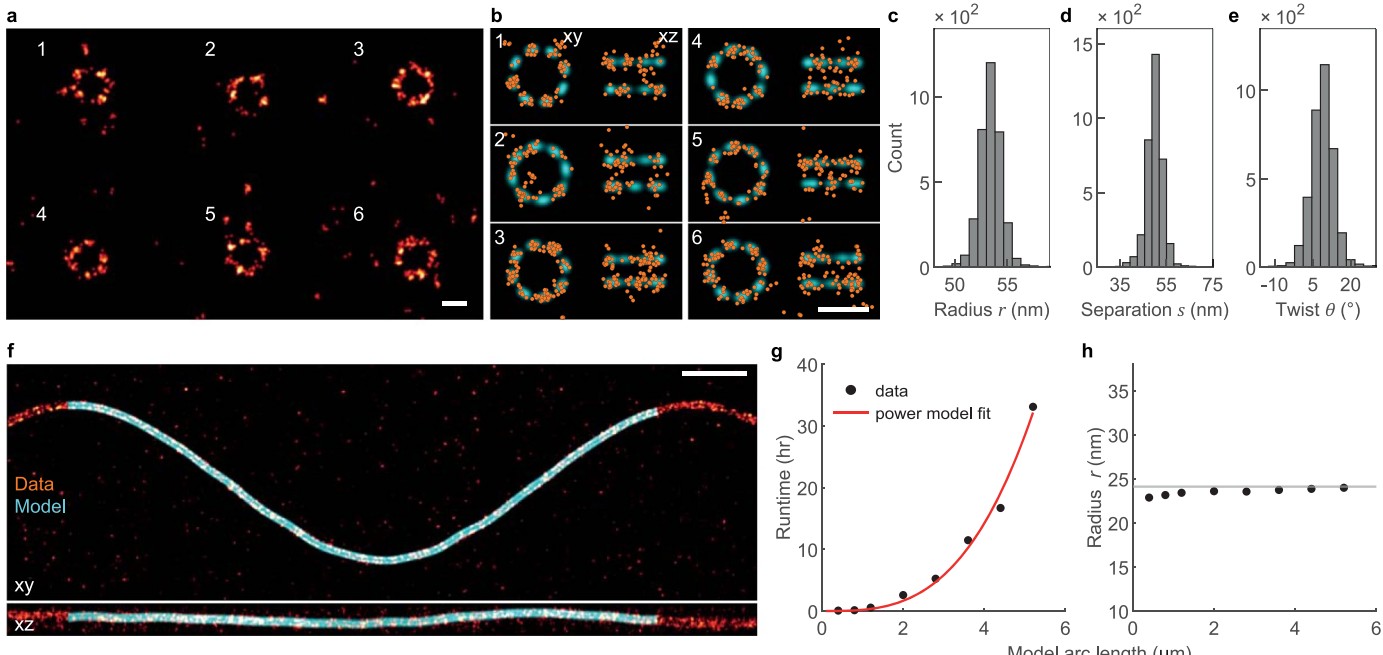

**Extended Data Fig. 5 | Extra information provided by simulations. a–e,** Simulated NPCs based on experimental parameters. **a**, Representative images of the simulated Nup96-labeled nuclear pore complexes (NPC) in 3D (top view). **b**, single NPCs (orange) as indicated in **a** are fitted with the eight-fold symmetry model. **c-e**, Histograms of three fitted parameters: radius $r = 53.4 \pm 1.1$ nm (**c**), separation $s = 50.0 \pm 3.8$ nm (**d**), and twist $\theta = 8.9 \pm 4.7°$ (**e**). Shown values are mean ± s.d. Sample size: $n_s = 3,539$. **f–h, Runtime estimates based on the simulated microtubule segments with increasing arc lengths**. The segments were derived from the same simulated long sinusoidal microtubule. **f**, The longest segment fitted with the model having the longest arc length (5.2 μm), as the example. **g**, relation between fitting runtime $t$ and the model arc length $n$ and fit with power model $t - n^\alpha$ resulting in $\alpha = 3.1$. **h**, Measured radii $r$ against different segment lengths. The horizontal line indicates the ground truth. Sample size: $n_s = 1$. Simulation parameters are listed in Supplementary Table 1. Scale bars: 100 nm (**a**,**b**), 500 nm (**f**).

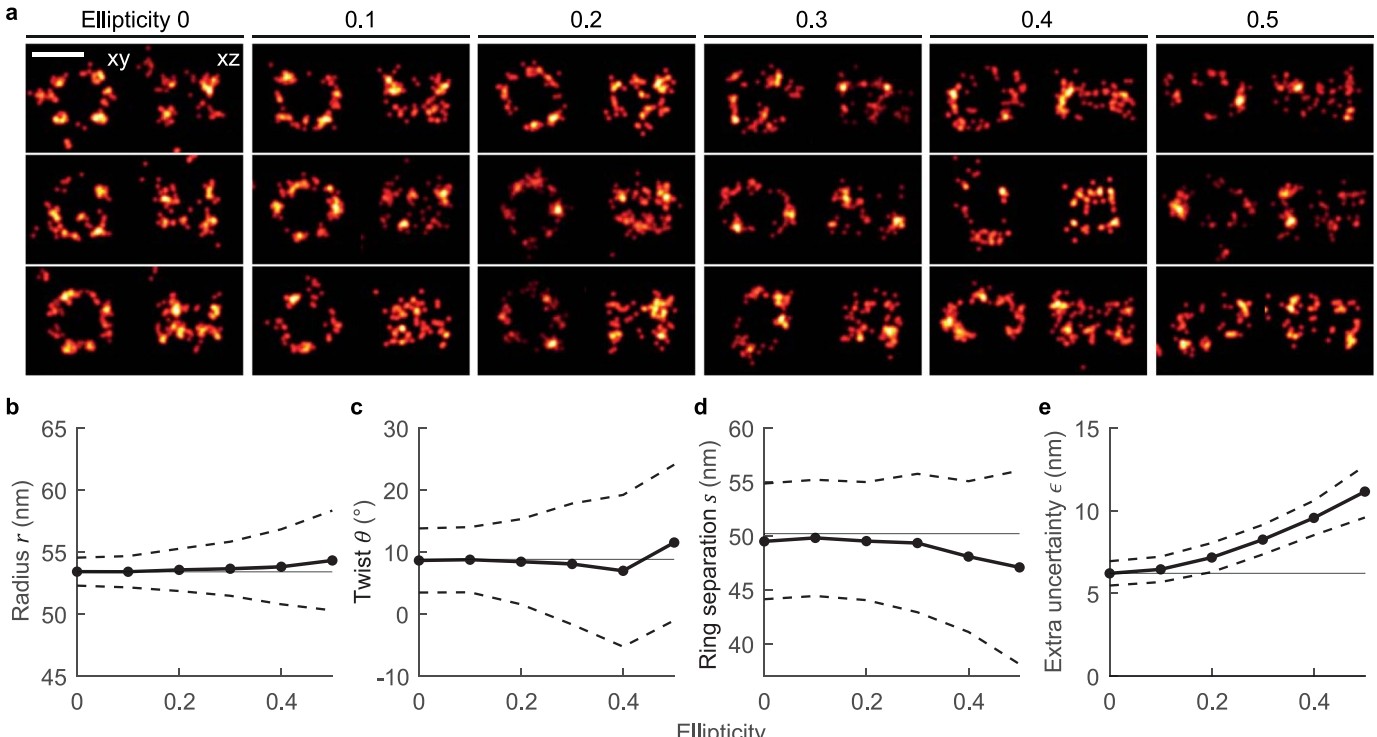

**Extended Data Fig. 6 | Parameter estimation errors of simulated NPCs with different ellipticities fitted with a ring approximation. a**, Example simulated NPCs with different ellipticities. The ellipticity is defined as $e=1\text{-}b/a$, where $a$ and $b$ are the lengths of the long and short axes, respectively. **b**–**e**, parameter estimations. Gray horizontal lines are the ground truths. The shown ground truth for the radius $r$ in **b** is the average of the two axes $a$ and $b$. **e** shows uncertainties that cannot be explained solely by localization precision. Such uncertainties are modeled by the single parameter extra uncertainty $\epsilon$ and can partially account for a model mismatch. Sample size: $n_s = 1{,}000$ for each dot. Scale bar: 100 nm.

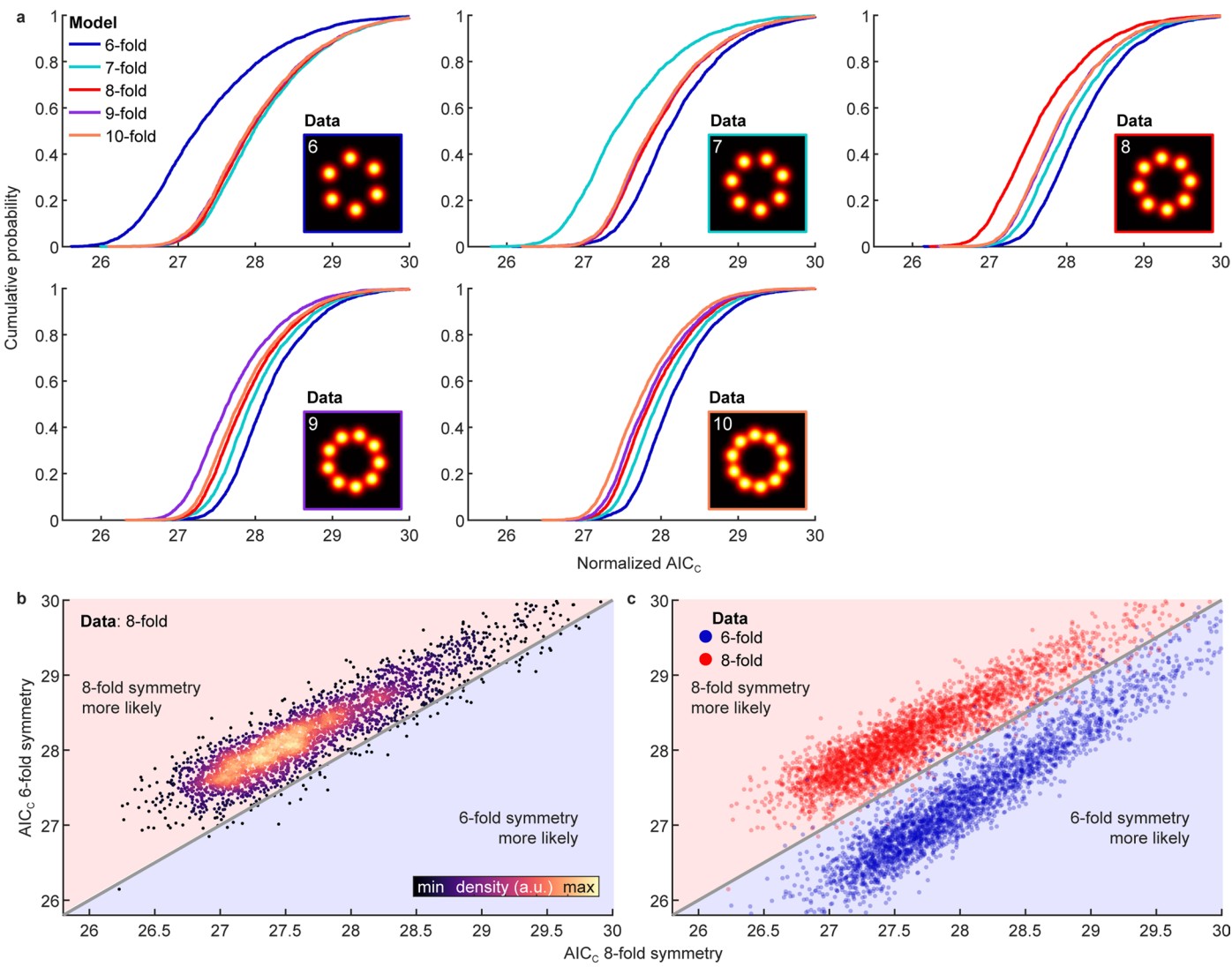

**Extended Data Fig. 7 | Validation of the model selection using simulations.**
**a**, Normalized AIC$_C$ of fitting simulated NPCs with different rotational symmetry
models. Each panel corresponds to a simulated NPC dataset generated from the
model indicated in the insets. Sample size: $n_s = 3,000$ for each panel. **b**, Scatter
plot showing normalized AIC$_C$ values of fitting the simulated 8-fold symmetric
sites with models having six-fold and eight-fold rotational symmetries. **c**, Scatter
plot showing normalized AIC$_C$ values of fitting both the simulated six-fold and
eight-fold symmetric sites with models having six-fold and eight-fold rotational
symmetries. The gray diagonal lines indicate equal normalized AIC$_C$ values. The
data points in **b** are displayed again in **c** for comparisons. Simulation parameters
are summarized in Supplementary Table 1.

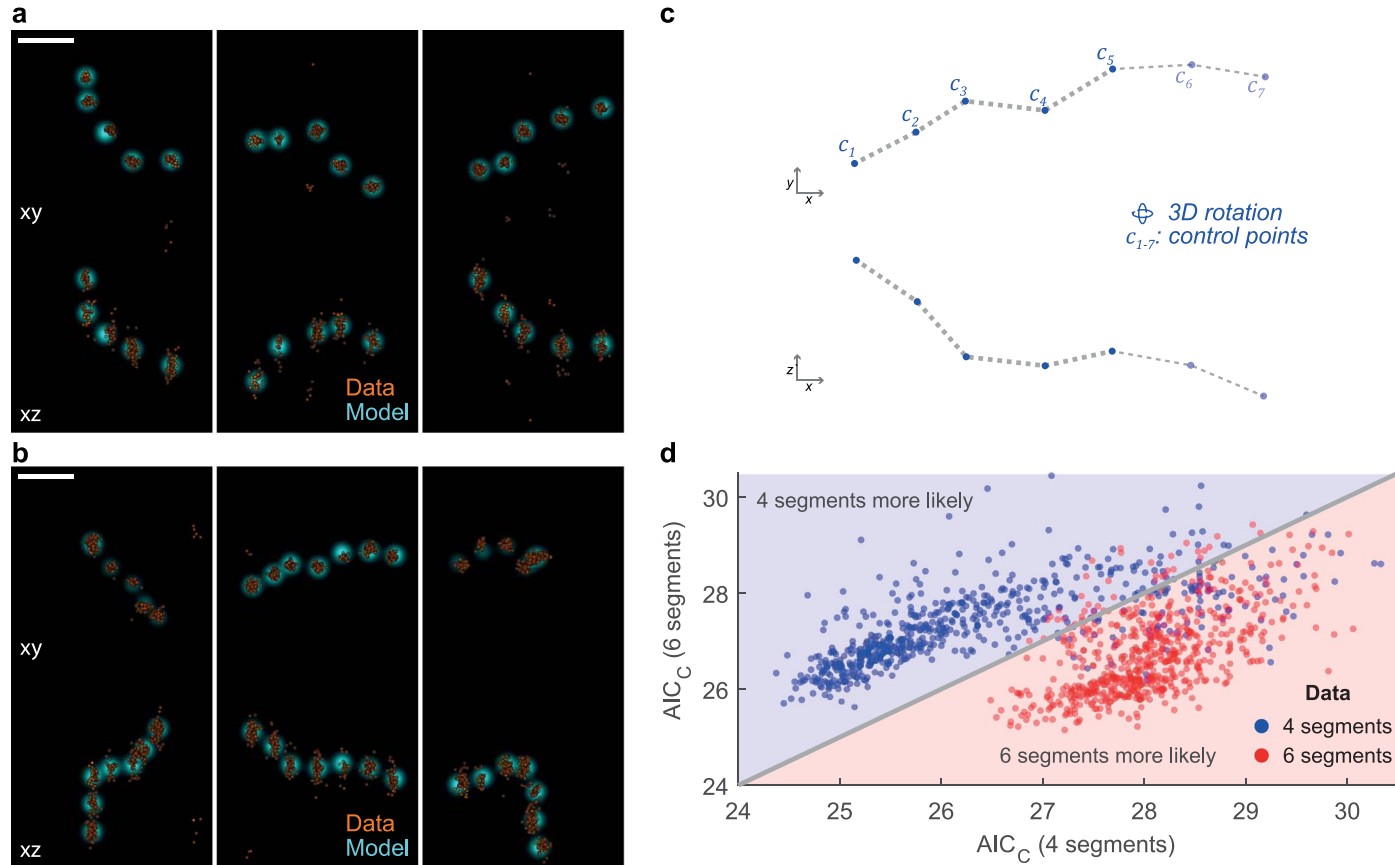

**Extended Data Fig. 8 | Model selection applied to line segments for selecting the number of segments. a, b**, simulated line segments [with four segments (**a**) and six segments (**b**)] fitted with the models used for generating the simulations (**c**). **c**, The line segment models used for the simulation and fitting. The four-segment model is composed of five control points (clusters) and the six-segment is composed of seven. **d**, Scatter plot showing normalized $AIC_C$ values of fitting both types of line segments with models having four and six segments. Sample size: $n_s = 600$ for each group. Scale bar: 100 nm.

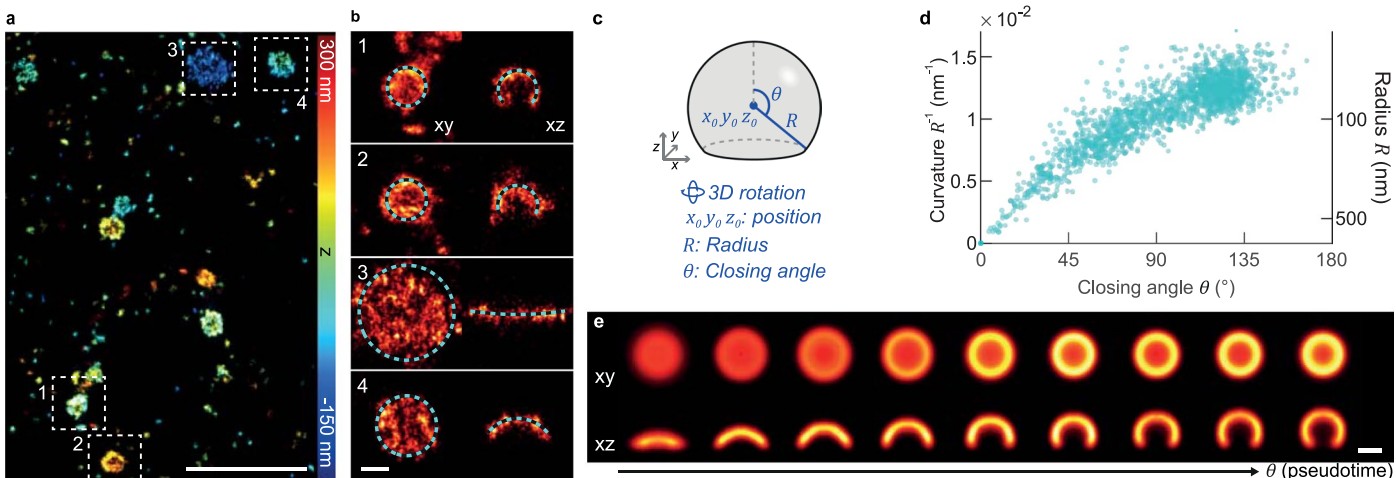

**Extended Data Fig. 9 | Quantification of mammalian clathrin coats.**
**a**, Representative images of the immunolabeled clathrin-coated pits at the bottom membrane of a SK-MEL-2 cell in 3D (top view). **b**, Single clathrin coats (red) as indicated in **a** are fitted with the spherical model (**c**), outlined. The *xz* side views show 30 nm thick central cross-sections. **d**, Scatter plot of the parameter radius $R$ and corresponding curvature $R^{-1}$ as a function of the closing angle $\theta$. **e**, The fitted closing angle $\theta$ is used as a proxy for pseudotime to sort individual coats according to their endocytic progression. In each time bin, clathrin coats are aligned based on their fitted position and orientation. Radii of clathrin coats are rescaled to the median radius of the bin except for the first bin. Here rotational symmetry around the *z* axis was imposed as the model is rotational symmetric. Bin size: 182 sites. Sample size: $n_s = 1,645$, $n_c = 13$. Scale bars, 1 μm (**a**), 100 nm (**b, e**). This figure was created based on data recently published by our group[48].

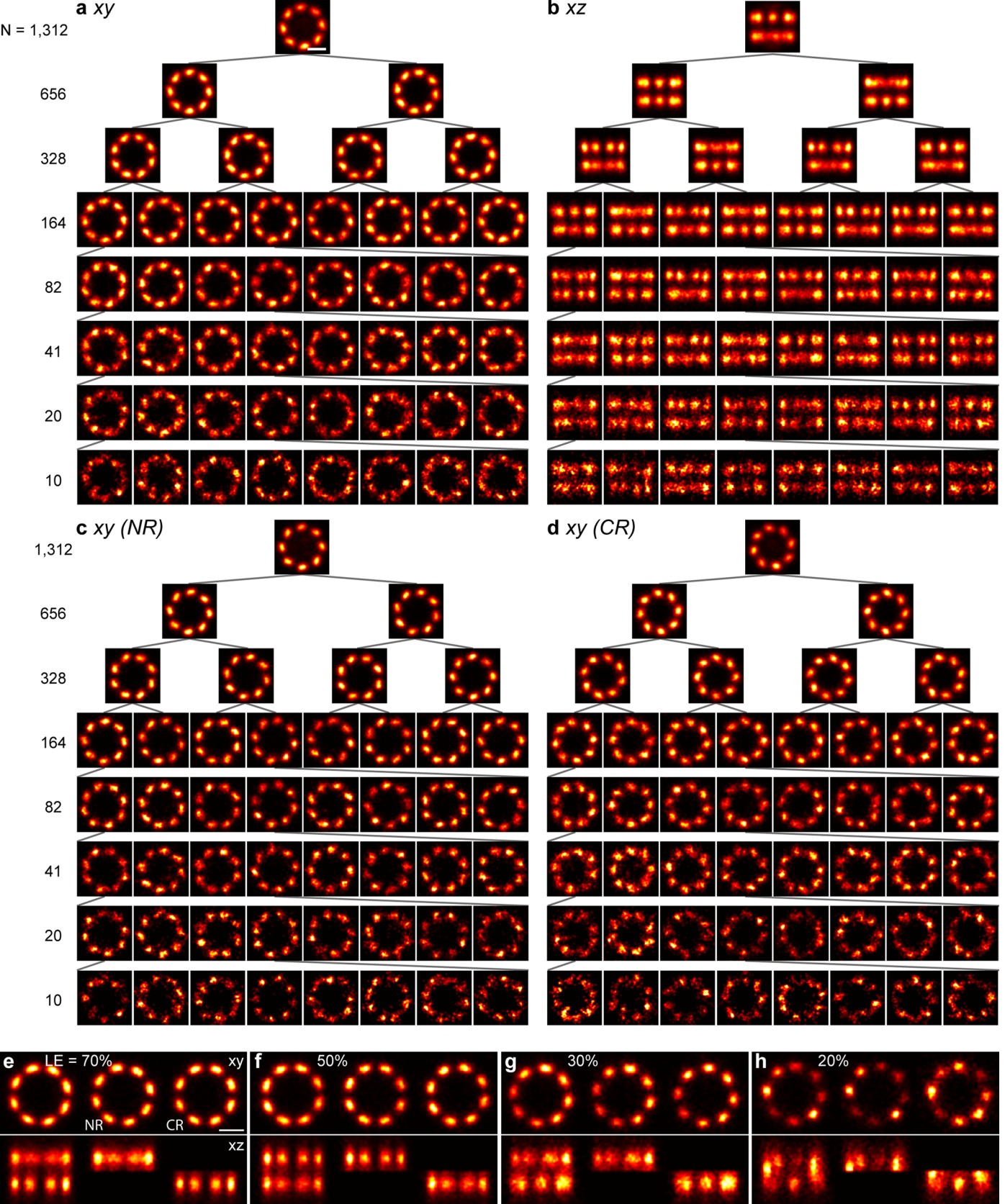

**Extended Data Fig. 10 | Requirements for calculating the high-quality template-free NPC average. a–d, NPC averages calculated based on different numbers of particles.** We hierarchically split the experimental 1,312 particles into subsets down to 10 particles per subset. We then analyzed each subset based on the same procedure as described in Fig. 5. **a**, Top view (xy) and **b**, side view (xz) reconstructions of the averages. **c**, Top view reconstruction of the nucleoplasmic ring (NR) and **d**, of the cytoplasmic ring (CR) are shown for visualizing outward tilts of individual corners. **e-h, NPC averages calculated based on particles simulated with different labeling efficiency.** We simulated NPCs according the same procedure as in Extended Data Fig. 5 with various labeling efficiency. Sample size: $n_s$ = 300. Scale bars: 100 nm.

# Reporting Summary

## Statistics

For all statistical analyses, confirm that the following items are present in the figure legend, table legend, main text, or Methods section.

| n/a | Confirmed | |
|---|---|---|
| ☐ | ☒ | The exact sample size (*n*) for each experimental group/condition, given as a discrete number and unit of measurement |
| ☐ | ☒ | A statement on whether measurements were taken from distinct samples or whether the same sample was measured repeatedly |
| ☒ | ☐ | The statistical test(s) used AND whether they are one- or two-sided *Only common tests should be described solely by name; describe more complex techniques in the Methods section.* |
| ☒ | ☐ | A description of all covariates tested |
| ☐ | ☒ | A description of any assumptions or corrections, such as tests of normality and adjustment for multiple comparisons |
| ☐ | ☒ | A full description of the statistical parameters including central tendency (e.g. means) or other basic estimates (e.g. regression coefficient) AND variation (e.g. standard deviation) or associated estimates of uncertainty (e.g. confidence intervals) |
| ☒ | ☐ | For null hypothesis testing, the test statistic (e.g. *F*, *t*, *r*) with confidence intervals, effect sizes, degrees of freedom and *P* value noted *Give P values as exact values whenever suitable.* |
| ☒ | ☐ | For Bayesian analysis, information on the choice of priors and Markov chain Monte Carlo settings |
| ☒ | ☐ | For hierarchical and complex designs, identification of the appropriate level for tests and full reporting of outcomes |
| ☒ | ☐ | Estimates of effect sizes (e.g. Cohen's *d*, Pearson's *r*), indicating how they were calculated |

*Our web collection on statistics for biologists contains articles on many of the points above.*

## Software and code

Policy information about availability of computer code

| Data collection | Micro-Manager (1.4 and 2.0) and our published software EMU (v1.1) |
|---|---|
| Data analysis | Matlab (2022a) and custom software available at https://github.com/jries/SMAP |

For manuscripts utilizing custom algorithms or software that are central to the research but not yet described in published literature, software must be made available to editors and reviewers. We strongly encourage code deposition in a community repository (e.g. GitHub). See the Nature Portfolio guidelines for submitting code & software for further information.

## Data

Policy information about availability of data

All manuscripts must include a data availability statement. This statement should provide the following information, where applicable:
- Accession codes, unique identifiers, or web links for publicly available datasets
- A description of any restrictions on data availability
- For clinical datasets or third party data, please ensure that the statement adheres to our policy

All image data used in this study are available on the BioImage Archive data repository (accession number: S-BIAD563).

# Field-specific reporting

Please select the one below that is the best fit for your research. If you are not sure, read the appropriate sections before making your selection.

☒ Life sciences  ☐ Behavioural & social sciences  ☐ Ecological, evolutionary & environmental sciences

For a reference copy of the document with all sections, see nature.com/documents/nr-reporting-summary-flat.pdf

# Life sciences study design

All studies must disclose on these points even when the disclosure is negative.

| | |
|---|---|
| Sample size | We did not perform sample-size calculation. The current sample size is considered sufficient because it already provides higher precision than technical accuracy. Except for the long microtubule segments (for each condition, one segment is used for demonstration), the smallest sample size in this work is 161 and most of experiments/conditions have sample sizes larger than 1,000, resulting in well-defined distributions or average images. |
| Data exclusions | Only nuclear pore complexes with low quality or with ring separation smaller than 35 nm were excluded from further analyses. The exclusion was necessary for precise measurements of parameters. The rationale for this exclusion was shown in Extended Data Figure 3 and Extended Data Figure 4f-i. |
| Replication | All attempts at replication were successful. Each analysis was performed on three different cells except for the microtubule data shown in Figure 2. The original microtubule data was from Speiser et al., 2021., where only one cell was acquired. |
| Randomization | We did not perform any randomization. Randomization is not relevant because no comparison of different biological conditions was performed in this work. |
| Blinding | We did not perform any blinding. Blinding is not relevant because no comparison of different biological conditions was performed in this work. |

# Reporting for specific materials, systems and methods

We require information from authors about some types of materials, experimental systems and methods used in many studies. Here, indicate whether each material, system or method listed is relevant to your study. If you are not sure if a list item applies to your research, read the appropriate section before selecting a response.

## Materials & experimental systems

| n/a | Involved in the study |
|---|---|
| ☐ | ☒ Antibodies |
| ☐ | ☒ Eukaryotic cell lines |
| ☒ | ☐ Palaeontology and archaeology |
| ☒ | ☐ Animals and other organisms |
| ☒ | ☐ Human research participants |
| ☒ | ☐ Clinical data |
| ☒ | ☐ Dual use research of concern |

## Methods

| n/a | Involved in the study |
|---|---|
| ☒ | ☐ ChIP-seq |
| ☒ | ☐ Flow cytometry |
| ☒ | ☐ MRI-based neuroimaging |

## Antibodies

| | |
|---|---|
| Antibodies used | Primary:<br>Anti-Elys (catalog no. HPA031658, Atlas Antibodies, 1:50),<br>Anti-Nup133 (catalog no. HPA059767, Atlas Antibodies, 1:150),<br>Anti-Nup62 (catalog no. 610498, BD Biosciences, 1:150),<br>Anti-Nup153 (catalog no. ab24700, Abcam, 1:60)) .<br>Secondary:<br>CF660C labeled anti-rabbit antibody (catalog no. 20813, Biotium),<br>CF660C anti-mouse antibody (catalog no. 20815, Biotium),<br>CF680 labeled anti-rabbit antibody (catalog no. 20818, Biotium),<br>CF680 anti-mouse antibody (catalog no. 20819, Biotium). |
| Validation | Validations were performed by the respectively indicated manufacturers. The validated applications of the antibodies are summarized bellow: Anti-Elys for immunohistochemistry (IHC) and immunofluorescence in cell lines (ICC-IF); Anti-Nup133 for IHC, western blot (WB), and ICC-IF; Anti-Nup62 for WB (routinely tested), IF, immunoprecipitation (tested during development), and IHC (not recommended); Anti-Nup153 for ICC-IF. These antibodies were used to perform immunofluorescence staining on specimens for demonstrating the described analysis software in the work. |

# Eukaryotic cell lines

Policy information about cell lines

| | |
|---|---|
| Cell line source(s) | U-2 OS Nup96-SNAP-tag (catalog no. 300444, CLS Cell Line Service, Eppelheim, Germany). |
| Authentication | None of cell lines were further authenticated. |
| Mycoplasma contamination | Cells were tested negative for mycoplasma contamination. |
| Commonly misidentified lines<br>(See ICLAC register) | No commonly misidentified cell lines were used. |

