## [Peer Review File · Nature Methods]

Peer Review Information

Manuscript Title: Maximum-likelihood model fitting for quantitative analysis of SMLM data

Corresponding author name(s): Jonas Ries

Reviewer Comments & Decisions:

Decision Letter, initial version:

Dear Jonas,

Thank you for submitting your manuscript entitled "Maximum-likelihood model fitting for quantitative analysis of SMLM data". We have given the paper our careful consideration but we regret that we cannot publish it in Nature Methods.

It is Nature Methods' policy to decline a substantial proportion of manuscripts without peer-review, so that they may be sent elsewhere without delay. Decisions of this kind are made by the editorial staff when it appears that papers are unlikely to succeed in the competition for limited space.

We read your paper with interest and were convinced your approach works well for quantitative analysis of SMLM structures. However, it was not clear to us that the approach is sufficiently enabling of new biological discovery relative to established tools to justify publication in Nature Methods.

Nevertheless, thank you very much for giving us the opportunity to consider your manuscript. I am sorry that we cannot be more positive on this occasion and hope that you will promptly find a more appropriate forum for presenting your work.

Sincerely,
Rita

Rita Strack, Ph.D.
Senior Editor
Nature Methods

P.S. You may want to consider *Nature Communications* as a potential venue for publication of your manuscript, please see the footnote below if you would like further information about this. Please note that the editorial team at Nature Communications will consider your manuscript independently.

Author Rebuttal to Initial decision:

On 20. Dec 2021, at 17:51, Jonas Ries <jonas.ries@embl.de> wrote:

Dear Rita,
thank you very much for reading our manuscript "Maximum-likelihood model fitting for quantitative analysis of SMLM data", in which we present LocMoFit, a conceptually new way for analyzing SMLM

data. I consider this work the most useful and transformative tool that we developed in the last years, and it has been essential for us to answer important biological questions that were out of reach before (e.g., Mund et al., bioRxiv 2021). However, with your decision letter, I realize we failed to bring across this message sufficiently.

In the following I would like to briefly summarize again how LocMoFit enables new biological insights compared to established tools. Maybe you find that with textual edits or new experiments you would like to reconsider this manuscript. Otherwise, brief feedback on how to highlight this impact better would be much appreciated to help us with submission to another journal.

Quantitative data analysis is a major bottleneck in SMLM. Unfortunately, current approaches to analyze individual structures are essentially limited to fitting of 1D line profiles or circles to the data, and thus can only extract simple descriptors like size or distance. Particle averaging, on the other hand, can generate a high contrast representation, which works only on identical particles. This means that currently no tool exists to analyze the vast majority of cellular structures, which are complex, three-dimensional, heterogenous and often dynamic. This inability to analyze our own data in biological projects prompted us to develop LocMoFit.

LocMoFit overcomes this major gap because it enables extracting many meaningful parameters from individual structures in SMLM data. In our manuscript, we attempted to showcase the power of LocMoFit on two general classes of biological questions:

First, we demonstrate the potential of LocMoFit for in situ structural biology of complex multi-protein assemblies. Among other examples, we measured the three key parameters (diameter, distance and rotational shift) of the octagons formed by proteins in individual nuclear pore complexes. This allowed us to assess complex structures and their heterogeneity in cells, which would have been impossible by current tools for SMLM and very challenging by classical structural biology techniques. Thus, LocMoFit could be the key ingredient to establish SMLM as a complementary technology for structural cell biology to enable quantitative measurements of such structures. This is further helped by improved particle averaging compared to current approaches (Fig. 4c,d and Fig. 5).

Second, we show that LocMoFit can perform architectural analyses of highly dynamic cellular structures from snapshots taken in fixed cells. The idea is to extract a parameter that changes over time and use it to pseudo-temporally sort all structures. This allows evaluating the dynamic progression of other parameters and to obtain key mechanistic insights. Using clathrin-mediated endocytosis as an example, we quantified how the curvature of the clathrin coat changes over time (EDFig. 7d), which already allowed us to solve a long-standing controversy about the mechanism of endocytic coat remodeling in mammalian cells in a separate paper (Mund et al., bioRxiv 2021). Importantly, LocMoFit also enables rendering dynamic reconstructions by combining pseudo-temporal sorting with precise position and orientation measurements, yielding nanoscale movies of complex rearrangements of molecular machineries (Fig 4i, EDFig 7e, and SMovie 2). Creating such dynamic, multi-color protein distribution maps using superresolution microscopy is impossible with current tools and represents a major advance. We regret that we did not communicate more clearly how LocMoFit offers an entirely new concept to reconstruct the dynamics of cellular processes from static SMLM data. This is particularly relevant when

live-cell SMLM is not feasible due to limited time resolution and phototoxicity. To further illustrate this approach, we would be willing to add additional data from an ongoing biological project, where we used LocMoFit's dynamic reconstruction feature to ask how dynamin is concentrated at the future membrane scission site while the clathrin coat is remodeled (attached figure and movie).

With LocMoFit, we want to enable a wide SMLM user base to obtain novel quantitative biological insights. Like with other tools that we developed, we are confident that LocMoFit will achieve that, because it proved to be broadly applicable to many cellular structures and complexes, and we packaged it into an easy-to-use software that enables users to create and exchange models of their structures of interest.

Decision Letter, first revision:

Dear Jonas,

Thank you for your letter detailing how you would respond to the reviewer concerns regarding your Article, "Maximum-likelihood model fitting for quantitative analysis of SMLM data". We have decided to invite you to revise your manuscript as you have outlined, before we reach a final decision on publication.

Regarding the comparison to PERPL, we think any experimental comparison should be removed. Instead, we think it should be cited and discussed appropriately in the introduction.

- * include a point-by-point response to the reviewers and to any editorial suggestions
- * please underline/highlight any additions to the text or areas with other significant changes to facilitate review of the revised manuscript
- * address the points listed described below to conform to our open science requirements
- * ensure it complies with our general format requirements as set out in our guide to authors at www.nature.com/naturemethods

* resubmit all the necessary files electronically by using the link below to access your home page

[Redacted] This URL links to your confidential home page and associated information about manuscripts you may have submitted, or that you are reviewing for us. If you wish to forward this email to co-authors, please delete the link to your homepage.

We hope to receive your revised paper within three months. If you cannot send it within this time, please let us know. In this event, we will still be happy to reconsider your paper at a later date so long as nothing similar has been accepted for publication at Nature Methods or published elsewhere.

OPEN SCIENCE REQUIREMENTS

REPORTING SUMMARY AND EDITORIAL POLICY CHECKLISTS

Please note that these forms are dynamic ‘smart pdfs’ and must therefore be downloaded and completed in Adobe Reader. We will then flatten them for ease of use by the reviewers. If you would like to reference the guidance text as you complete the template, please access these flattened versions at <http://www.nature.com/authors/policies/availability.html>.

DATA AVAILABILITY

Please include a “Data availability” subsection in the Online Methods. This section should inform readers about the availability of the data used to support the conclusions of your study, including accession codes to public repositories, references to source data that may be published alongside the paper,

unique identifiers such as URLs to data repository entries, or data set DOIs, and any other statement about data availability. At a minimum, you should include the following statement: “The data that support the findings of this study are available from the corresponding author upon request”, describing which data is available upon request and mentioning any restrictions on availability. If DOIs are provided, please include these in the Reference list (authors, title, publisher (repository name), identifier, year). For more guidance on how to write this section please see: <http://www.nature.com/authors/policies/data/data-availability-statements-data-citations.pdf>

CODE AVAILABILITY

Please include a “Code Availability” subsection in the Online Methods which details how your custom code is made available. Only in rare cases (where code is not central to the main conclusions of the paper) is the statement “available upon request” allowed (and reasons should be specified).

ORCID

Nature Methods is committed to improving transparency in authorship. As part of our efforts in this direction, we are now requesting that all authors identified as ‘corresponding author’ on published papers create and link their Open Researcher and Contributor Identifier (ORCID) with their account on the Manuscript Tracking System (MTS), prior to acceptance. This applies to primary research papers only. ORCID helps the scientific community achieve unambiguous attribution of all scholarly contributions. You can create and link your ORCID from the home page of the MTS by clicking on ‘Modify my Springer Nature account’. For more information please visit www.springernature.com/orcid.

Sincerely,
Rita

Rita Strack, Ph.D.
Senior Editor
Nature Methods

Reviewers' Comments:

Reviewer #1:

Remarks to the Author:

Wu et al develop LocMoFit, a maximum likelihood model fitting to extract quantitative parameters from structures imaged using SMLM. The idea is that if individual structures imaged using SMLM can be fit to a geometric model, then useful quantitative parameters can be extracted about the structure of interest (size, angle etc...). LocMoFit is applied to modelling NPCs, microtubules and CCPs in SMLM data. Authors repeat some of the previous applications and concepts they had already demonstrated using LocMoFit (such as registering/aligning 2-color images using the same reference structure to build a model of a multi-protein NPC complex; "dynamic" reconstruction of CCP endocytosis and model free averaging of SMLM structures). The theory part of the manuscript is well explained and allows for people who have a limited experience with the math behind the process to still understand everything. The build up is logical and goes from simple to more complex, being clear at every step.

LocMoFit is in principle an interesting idea. However, the paper and the code as presented have many shortcomings that will ultimately limit its broad applicability and usefulness. Below are our major comments on the manuscript:

- It is clear that LocMoFit will be useful when the exact geometric shape is known and there is little variability in that shape from one imaged structure to the next. However, when there is large variability or when the exact shape is not known a priori, it will be challenging to use or evaluate the appropriateness of this approach. In particular, when the quality of the SMLM data is low (low labeling efficiency etc...) it is not clear that this approach will be applicable. Authors say that the geometric model can be determined by "visual inspection" or "a priori knowledge" from EM. They showcase examples of the latter: NPCs, microtubules and CCPs, all of which have been extensively visualized with EM and the exact model to be used is known a priori. In fact, for some of the NPC images shown, one would never guess using visual inspection that the two ring model is the most appropriate model to use for the fit without a priori knowledge. Hence, LocMoFit would be most useful when a priori knowledge exists and its usefulness beyond that is questionable.

- Authors state “We usually validate a fitting pipeline with simulations before applying it to experimental data.” It is not clear how this works in practice. One must have an idea about the model of the experimental data before one can do this (only possible in a limited number of well-defined cases), and how can one be sure that the experimental data follows this model closely? How is this validity checked on the experimental data? This especially chimes in with the comparison between the simulated and experimental data. The simulated data has much more narrow distributions than the experimental data, and the authors explain this by saying some of the biological variability is unexplained. Could this actually be an indication that the model being used for fitting is not correct (or missing a contribution), and how would the authors compensate for this?
- The lack of a large availability of models for different type of data structures (either as images or as mathematical expressions) is a major limitation. More models should be included to make the method more useful to a broad readership E.g., mitochondria, other organelles (lysosomes, ER...), other nuclear structures etc. It is mentioned in the Discussion that people are encouraged to submit their models, but an extended database of models should already exist so people can start and build from existing models. For a biologist (target audience) it would not be a straightforward task to add a new geometrical model to the software.
- How do the biological variations that were unexplained by the model influence the model in itself? Does the model still hold, or does an additional source of variance have to be added to the model and the fitting redone?
- The microtubule section is not really explored all that much as the NPC data, and thus it feels like it was just added to have a second example. It would be nice to have some simulation studies done on this as well (e.g., what is the length of the sections that can be investigated now. It was arbitrarily taken at 1 μm , but how far can this be stretched?)
- In the multi-color section, the model for the structure obtained by (or determined by) LocMoFit is used to do image registration, which should in principle give a better result than when e.g. image correlation is obtained. However, it is not clear from the text how the procedure is actually done: do the authors make use of the multicolour capacity of LocMoFit, or is every channel modelled independently? (Would there be a difference between the two?)
- Also, the authors mention that the two-color helps when labelling efficiency is low, but this also has its limits: due to the symmetry of the reference structure, it can be rotated around the z-axis in multiple ways. If labelling efficiency is low of the second protein, and only 1 site is active in the images (e.g. images 2 and 3 in Figure 4b, but imagine this all the time), then false conclusions can be made from it. This statement should be nuanced a little, and is mainly useful when there is prior knowledge about the second protein. If nothing is known (new proteins/structures), then this should be considered carefully.
- The comparison between LocMoFit and PERPL is unfair. Using the same model to simulate (noisy) data and fit is always going to give good results (as long as SNR is still reasonable), whereas a method that works on a completely different principle is not going to perform as well. Especially when structures simulated in the data are round, PERPL will have an issue (due to the data reduction into 1 dimension). How do the two compare e.g. the microtubules structure. It should still be in favour of LocMoFit (as it

uses the exact model), but the differences are probably not going to be as big. The comparison/discussion for Extended Figure 8 is therefore not really a measure for how good LocMoFit is, but more about in which situations PERPL is not working well.

- The figures in Extended Data Figure 2 are very misleading and inaccurate. An error is always positive (as it is a distance/angle difference/etc. and therefore does not matter in which direction is it different). As the simulation is a random process, the number of 'negative' differences will be the same as the number of 'positive' differences and therefore always average out to 0. These errors should be in absolute values (and not be negative and positive) and will be more representative to the true error.

Minor comments on manuscript:

- Figure 5b is confusing: the authors select the most likely site and then start from there for building the template, but it is not clear if they maximize or minimize the log likelihood (I assume maximizing as they want the highest similarity), and even then, site iv is never the highest or lowest. So I do not understand how they got to site iv as the initial template (as per the caption).
- "Dynamic reconstruction": Dynamic is misleading as these are not really dynamic measurements.). The text mentions pseudotime, and this seems a more appropriate term.
- One of the advantages of LocMoFit is that the individual structures can be used for fitting instead of averaging over all the structures. This allows investigating the heterogeneity of the sample, and could thus be of high biological relevance. However, the entire last part of the manuscript is dedicated to model-free averaging to set up a model of the structures, which seems to nullify their previous point. I understand that you do not have to do both of, but it could potentially increase the impact of the paper if they start off with the model-averaging (something that has already been done for in situ structural biology with SMLM) and then continue with 'looking at the heterogeneity of the sample' (a more novel application).

Major comments on the software:

The software was not intuitive or easy to use and gave many warnings/errors that were not clear how to fix. In addition, it was not clear what results should be obtained and how the results should be accessed. The documentation is very long for SMAP and not easy to navigate and incomplete for LocMoFit.

See below for more detailed comments:

GUI in Matlab 2021b

- Opening the ROI manager loads partially outside the screen (15.4 inch laptop @ 1920 x 1080 pixels) Assumably because size is given in absolute pixel values, and not in relative pixel values.
- Lots of warnings of the same kind (when clicking on 'redraw' or 'all' in the ROI manager):
 1. Warning: Single-selection 'listbox' control requires that 'Value' be an integer within Character vector range

Control will not be rendered until all of its parameter values are valid

2. In gui/SEExploreGui/redrawall (line 246)

In gui.SEExploreGui>redrawsiteall_callback (line 959)

Warning: Single-selection 'listbox' control requires that 'Value' be an integer within Character vector range

Control will not be rendered until all of its parameter values are valid

3. Many many GUI warnings that won't all be copied here.

- Came across one error as well that could not be reproduced (also not sure if that is general SMAP functionality, or LocMoFit):

1. Intermediate dot '.' indexing produced a comma-separated list with 0 values, but it must produce a single value when followed by subsequent indexing operations.

Error in imline/getPosition (line 150)

```
pos = obj.api.getPosition();
```

Error in gui.SEExploreGui>plotline (line 844)

```
posin=obj.hlines.(posfield).getPosition;
```

Error in gui.SEExploreGui>anglebutton_callback (line 871)

```
plotline(obj,'rotationpos',obj.guihandles.angle);
```

Related documentation

Error while evaluating UIControl Callback.

- Error when clicking on 'evaluate current ROI':

1. Dot indexing is not supported for variables of this type.

Error in interfaces.SEEvaluationProcessor/getLocs (line 108)

```
sx=obj.site.image.rangex;
```

Error in ROIManager.Evaluate.generalStatistics/run (line 26)

```
locs=obj.getLocs({'locprecnm','PSFxn','xnm','ynm','phot','bg','numberInGroup'},'layer',k,'size',roisizeh);
```

Error in interfaces.SEEvaluationProcessor/evaluate (line 83)

```
out=obj.run(p);
```

Error in gui.SEEvaluationGui>evaluatesite (line 227)

```
module.evaluate(site);
```

Error in gui.SEEvaluationGui>preview_callback (line 211)

```
evaluatesite(obj,obj.SE.currentsite,1)
```

Error while evaluating UIControl Callback.

- Not clear from the tutorial how to perform a composite model. It stops at step 8.4, and could not really figure out how to advance from there. When some things were tried, errors were thrown, so

probably not following the correct steps. This ties in with the fact that the GUI is not really intuitive to use, and without a tutorial, it is difficult to have anything working properly.

- Simulating data according to the documentation does NOT work.

Cannot continue with the simulations. This is from a clean install, clean settings, Matlab 2021b

1. Error using LocMoFit/setModel (line 163)

Invalid: You are adding a 3D model for fitting 2D data.

Error in LocMoFit/addModel (line 224)

obj.setModel(model,lastMod+1);

Error in ROIManager.Evaluate.LocMoFitGUI>initmodel (line 1266)

fitter.addModel(geoModeltemp);

Error in ROIManager.Evaluate.LocMoFitGUI>loadmodel_callback (line 1205)

initmodel(obj, modelnumber);

Error while evaluating UIControl Callback.

2. Click Connect to LocMoFit and check LocMoFitGUI in the new window. -> Cannot find the 'connect to LocMoFit' button anywhere.

Reviewer #2:

Remarks to the Author:

The work by Wu and coworkers demonstrates an MLE approach to obtaining structural information from SMLM data. The authors use a combination of ground-truth models and the well characterised nuclear pore complex to validate their methodology. They eventually apply it to develop and benchmark a kinetic growth model for Clathrin-mediated endocytosis. The authors have open-sourced their code and is available as integration to SMAP developed by the Ries Lab as well as an API for third-party integration.

The simulations are extensive, the maths is solid, the manuscript is extremely well written and, overall, the work represents a huge leap forward for which I have to congratulate the authors. However, there are important shortcomings that have to be addressed before I can recommend this for publication.

1, the vision of the authors is to enable in-situ structural biology using SMLM. And, I agree that SMLM is one of few (perhaps the only method) that has the potential to analyse the structure of heterogenous macromolecular complexes inside the cell. However, talking about protein structure, measurements not the angstrom scale are not useful from a structural biology perspective; they are only useful in constructing kinetic growth models as nicely demonstrated by the authors. To what extent is the phrase 'in-situ structural biology' supported in light of the following:

a, none of the measurements in the paper, aside from the simulations, are reported with angstrom resolution (including those performed on the highly conserved nuclear pore complex).

b, this tool is most useful, as suggested by the authors, when applied to study the structure of heterogenous complexes which escape particle averaging by alternative techniques (e.g. CryoEM). However, the structure of the analysed complexes is not studied, one-by-one, at high resolution - instead they are either averaged (through the model free routine) or pooled in samples/bins for further analysis.

None of the above is a deterrent to publishing this beautiful work, but it is important to set the expectations right from the beginning - that this is a tool for analysing the nanoscale structure of protein complexes at the ensemble level as otherwise, for a structural biologist, this tool could image protein structures with angstrom resolution which is not the case.

2, the simulations in extended figure 2 are performed with one parameter changed at a time which is not informative for dictating the optimal experimental conditions that would yield measurements with the highest accuracy. To the non-expert reader, the reported method works well even when labelling efficiency is 30% - where in reality, that low labelling efficiency can be accompanied by reductions in the localisation precision, amplification in the background, etc.

Additionally, the very important parameters, linker length, density of localisations and low- and high-frequency drift are not simulated despite the fact that they could worsen spatial resolution by several tens of nano-metres.

The authors are asked to simulate these parameters alone and combined to guide the end-user in understanding the limitations of the method and/or choosing the right set of parameters that would provide them with an acceptable error in their quantitative structural measurements. This in addition to showing the reader the correlation between the error in these measurements and the sample size (i.e. number of sites/structures).

3, some readers will be familiar with the work of Szymborska et al. Science, 2013 where the radii of the different nucleoporins was reported with angstrom accuracy using SMLM (see figure 1F). It is important for the authors to establish the correlation between the accuracy of their, and the reported, measurements commenting on the discrepancy between both as well as the role of data curation in improving the accuracy of measurements.

4, the mathematical models used in this work are relatively simple. Complexes, particularly those of intrinsically-disordered proteins, assemble into complex geometries that can be modelled with discrete

models. The authors demonstrate discrete modelling in the context of model scoring of the conserved NPC structures, but not for more complex heterogenous structures. The authors are asked to perform simulations with heterogenous structures demonstrating the validity of true discrete modelling (i.e. flexible line segments) and model scoring.

5, it is important to establish the supremacy of MLE compared to the widely used LS/NLS in terms of accuracy or at least both approaches require the same amount of time to execute as with the MLE it requires 10 seconds per site which projects to several tens of hours for tens of thousands of structures.

Reviewer #3:

Remarks to the Author:

The manuscript presents a method to produce 3D structural models of proteins and protein complexes from localization coordinates - such as those obtained from single-molecule localization microscopy (SMLM). In particular, the method enables to fit arbitrary models to localization coordinates and to extract quantitative model parameters with error estimates. In addition, it is able to derive measures of quality of fitness that can be used to select which structural model is more appropriate. Also, it can rely on 2-color imaging to re-assemble multi-color protein distribution maps. Finally, the authors demonstrate the ability of their method to derive model-free averaging.

The manuscript is well written, and the method clearly described and validated. However, a number of issues need to be resolved to ensure that this method becomes of widespread interest to a community interested in using SMLM to reconstruct the 3D structures of protein complexes.

1) In their approach for multicolor protein distribution maps, the authors image 2 proteins simultaneously and use one as a reference structure. This approach is fine when there is no variation in the relative spatial distribution between proteins in a given structure. However, it is reasonable to imagine situations where this assumption will not be valid, and will lead to erroneous results. For instance, if there are two conformations with the same reference distribution but different distributions for target 1 and target 2, then the final model will contain 'assembled' distributions for these targets that will be wrong as the reconstructed 3-color structure actually does not exist in nature.

2) The software has been tested and validated in pretty stable and well known structures, mainly the NPC. It would be important to test how the method will perform in more realistic cases, where there is structural heterogeneity, different complex compositions, or where there is flexibility. At the end this package will be most useful if it can be applied to these kind of scenarios, rather than just to 'perfect' model systems that are interesting to a limited audience.

3) This last point, is in fact, a main claim of the method "As the fitting is performed on individual structures without averaging (e.g., Fig. and Extended Data Figure 7), this will help investigating the vast majority of cellular structures that are heterogeneous and complex.". However, I don't see where this has been tested or validated. In absence of this advantage, where is the gain with respect to cryo-EM?

4) I would have liked to see testing of the performance of the algorithm under different realistic experimental situations. For instance: how many particles or sites are necessary to converge? is this experimentally feasible? how does the model perform with lower efficiencies of detection?

5) The code is provided in MATLAB. If I understand correctly, this would require potential users to pay for a MATLAB license to use it. If this is the case, it would be important to provide a way in which users can bypass this, such as with a compiled dynamic library.

6) Literature on the previous methods used for particle averaging are not exhaustive and only seem to focus on those written by the author.

7) It is surprising to see a comparison between LocMoFit and PERPL in the Discussion. This comparison should be instead within the results section.

Author Rebuttal, first revision:

Reviewer #1:

Remarks to the Author:

Wu et al develop LocMoFit, a maximum likelihood model fitting to extract quantitative parameters from structures imaged using SMLM. The idea is that if individual structures imaged using SMLM can be fit to a geometric model, then useful quantitative parameters can be extracted about the structure of interest (size, angle etc...). LocMoFit is applied to modelling NPCs, microtubules and CCPs in SMLM data. Authors repeat some of the previous applications and concepts they had already demonstrated using LocMoFit (such as registering/aligning 2-color images using the same reference structure to build a model of a multi-protein NPC complex; "dynamic" reconstruction of CCP endocytosis and model free averaging of SMLM structures). The theory part of the manuscript is well explained and allows for people who have a limited experience with the math behind the process to still understand everything. The build up is logical and goes from simple to more complex, being clear at every step.

LocMoFit is in principle an interesting idea. However, the paper and the code as presented have many shortcomings that will ultimately limit its broad applicability and usefulness. Below are our major comments on the manuscript:

We would like to thank reviewer 1 for the detailed analysis of our manuscript that will help us a lot in improving it.

We would like to clarify that previous studies on registering or averaging NPCs (Heydarian et al., Nature Methods, 2018; Heydarian et al., Nature Communications, 2021; Jimenez Sabinina et al., Molecular Biology of the Cell, 2021; on which we are authors) used completely different concepts. Our recent preprint on the reconstruction of the clathrin coat during endocytosis (Mund & Tschanz et al., bioRxiv 2021) is directly based on this work. It does not include a description of LocMoFit but cites the preprint version of LocMoFit (Wu et al., bioRxiv 2021) for this purpose.

1. It is clear that LocMoFit will be useful when the exact geometric shape is known and there is little variability in that shape from one imaged structure to the next. However, when there is large variability or when the exact shape is not known a priori, it will be challenging to use or evaluate the appropriateness of this approach. In particular, when the quality of the SMLM data is low (low labeling efficiency etc...) it is not clear that this approach will be applicable. Authors say that the geometric model can be determined by “visual inspection” or “a priori knowledge” from EM. They showcase examples of the latter: NPCs, microtubules and CCPs, all of which have been extensively visualized with EM and the exact model to be used is known a priori. In fact, for some of the NPC images shown, one would never guess using visual inspection that the two ring model is the most appropriate model to use for the fit without a priori knowledge.

Hence, LocMoFit would be most useful when a priori knowledge exists and its usefulness beyond that is questionable.

We would like to first clarify that LocMoFit does not require knowledge of the exact shape of the structure, nor that the structure cannot be highly variable. A single geometric model in LocMoFit can describe any size of the structure and various shapes, which are encoded in the fitting parameters. For example, single models describe all endocytic shapes from patches to closed vesicles in ED Fig. 9, the entire temporal evolution of two protein distributions in yeast endocytosis (Fig. 4e-i), or any microtubule with any path or diameter (Fig. 2), respectively. Variability can be incorporated in a model to study this variability (e.g., the variability of the size or twist angle of the NPC in Fig. 2). The model selection feature of LocMoFit can help selecting the right model.

LocMoFit can be seen as an extension of curve fitting to SMLM point clouds (mentioned on Page 4, Lines 125-126). In curve fitting, a model function is fitted to data to estimate the best guess of parameters that describe the underlying process. Curve fitting is extensively used with approximate models, even if not every detail of the process is known. Often, simple approximations are chosen (linear, polynomial, or Gaussian models, or sigmoidal curves) that capture essential features of the data and improve robustness thanks to the smaller number of fitting parameters. In the same way, LocMoFit can give reasonable descriptions of a structure, even if the model does not include all details. To illustrate this, we simulated more complex structures (elliptical NPCs) and fitted them with the model assuming no ellipticity. The result shows that the ring approximation of the elliptical NPCs yields small estimation errors even when the NPC is largely distorted (mentioned on Page 7, Lines 276-283, illustrated in ED Fig. 6). As is the case in curve fitting, too many free fitting parameters carry the risk of overfitting, and the precision of the parameter estimates depends on the noise in the data. Thus, higher data quality

improves the parameter estimates in LocMoFit and allows fitting of more complex models. But as we show in ED Fig. 2 (which we now extended according to point 2 of reviewer 2; the simulations are now based on the experimental PSF and thus are more realistic), even low localization precisions or labeling efficiencies still lead to surprisingly good parameter estimates.

The reviewer is correct that some a priori knowledge is needed in LocMoFit to choose a suitable model. In practice, we do not think that this is a strong limitation: We could not think of many applications for which SMLM is used to quantify sub-cellular structures, but where it is impossible to guess some aspects of the underlying geometry. This can come from diffraction limited images, electron micrographs or visual inspection of the SMLM images (mentioned now on Page 4, Lines 121-122). When looking at ED Fig. 4d-e and Fig. 5, or just as at the SMLM data with a 3D viewer, the two-ring geometry for the NPC is very apparent. There are certainly use cases where LocMoFit is not suitable, i.e., for random structures that require too many features to describe (highly variably topology like the actin cortex or other dense networks). But for the vast number of subcellular structures studied with SMLM some hypotheses on the underlying geometry can be usually generated based on prior knowledge, or at least a simple approximate model can be inferred from visual inspection of the SMLM data itself. In all those cases, LocMoFit extracts the best values for parameters that describe the geometry. When this is not the case and the structure is featureless, LocMoFit is not applicable. This is now discussed on Page 5, Lines 185-188.

2. Authors state “We usually validate a fitting pipeline with simulations before applying it to experimental data.” It is not clear how this works in practice. One must have an idea about the model of the experimental data before one can do this (only possible in a limited number of well-defined cases), and how can one be sure that the experimental data follows this model closely? How is this validity checked on the experimental data? This especially chimes in with the comparison between the simulated and experimental data. The simulated data has much more narrow distributions than the experimental data, and the authors explain this by saying some of the biological variability is unexplained. Could this actually be an indication that the model being used for fitting is not correct (or missing a contribution), and how would the authors compensate for this?

We thank the reviewer for the comment that the validation and simulation pipeline are not sufficiently clear. We now clarify these points in the revised manuscript and extend tutorials to enable the user to perform these validations in practice.

As discussed under point 1, LocMoFit requires the choice of a model, which however can describe many shapes and which, in our view, is possible for many applications. The purpose of the validation pipeline is to investigate the accuracy and robustness of the fitting with this model under defined experimental conditions (photons, background, labeling efficiency, which all can be estimated from the data). Now this is mentioned on Page 6, Lines 207-211. In practice, the simulations provide the ground truth for testing an analysis workflow, given a specific geometry.

The reviewer raises another important point: what happens if the model is wrong? How can I know about this? This point is more difficult to assess. If the model is an approximation of the true underlying

structure, the parameters are meaningful and interpretable (see reply to comment 1). If an entirely wrong geometry is assumed, the likelihood from the fit (a measure for the goodness of fit) is decreased. Often, this value (and its derivative AICC) is not sufficient to rule out a bad fit by itself due to its large variability, but it can be used to exclude completely wrong models, and to choose better models (Fig. 3). The direct interpretation of the narrow distribution of simulations compared to the experiment (Fig. 2d-f) is that indeed in the cell not all NPCs have exactly the same radius or twist angle, something that has been shown before (Stanley et al., Life Science Alliance, 2018; Zimmerli et al., Science, 2021). Our formulation that this variability is not part of the model is incorrect. Of course, the different sizes and twist angles are perfectly described by the model. We now revise it on Page 7, Lines 274-276. But there could be additional variability, like deformations of the NPC. To test the effect of additional variability and therefore the impact of geometric approximations, we fitted simulated elliptical NPCs, a previously reported deformation, with the ring NPC model (see also reply to point 1). We found that the spread in the parameters indeed scales with the strength of the deformation (i.e., ellipticity), but overall the spread remains small. The result is included as new ED Fig.6 and described on Page 7, Lines 276-287. In the framework, a key parameter, the extra uncertainty ϵ is included, which in general informs how well the data are approximated by the model. We quantified ϵ in the experimental data and introduced it to the simulation as part of the random displacement to compensate partially for a variability that is not explicitly included in the model. For consistency, we now increased ϵ from 3 to 6.4 nm (measured from the data based on our fit) in the related simulations and updated Fig. 2d-f and ED Fig. 5a-e and ED Fig. 7.

3. The lack of a large availability of models for different type of data structures (either as images or as mathematical expressions) is a major limitation. More models should be included to make the method more useful to a broad readership E.g., mitochondria, other organelles (lysosomes, ER...), other nuclear structures etc. It is mentioned in the Discussion that people are encouraged to submit their models, but an extended database of models should already exist so people can start and build from existing models. For a biologist (target audience) it would not be a straightforward task to add a new geometrical model to the software.

We agree that LocMoFit so far has been lacking simpler and more general models that a less experienced user could start from for their analysis. We now added many general models (circle, line, sphere, ellipse, deformed spheres (ellipsoid), to fit intracellular organelles like lysosomes, endosomes, peroxisomes, etc.) Details of these models are now included in the LocMoFit manual and online documentation. However, determining shapes of larger organelles that can be extracted from, e.g., confocal images might not be a target application for SMLM and LocMoFit.

4. How do the biological variations that were unexplained by the model influence the model in itself? Does the model still hold, or does an additional source of variance have to be added to the model and the fitting redone?

As explained for point 1, most biological variations (e.g., in size or relative position of components) can be included in the model via flexible parameters. The fit then directly returns the value of the parameter

indicating this biological variation, which can be interpreted directly. As detailed in the reply to comment 2, additional variations that are not part of the model (e.g., small local displacements) are in many cases still approximated well by the model and the fitting results for the main parameters are still informative and reasonable but can lead to an additional uncertainty in the parameter estimates. To directly quantify such additional biological variability that is not captured by the model, one can attempt to extend the model to incorporate it. Depending on the quality of the data and the number of parameters needed to describe the variability, this might lead to overfitting. In this case, a simpler model is preferable that describes this variability with a single extra fitting parameter: the extra uncertainty ϵ (see Fig. 1c, ED Fig. 2 and ED Fig. 6e; also discussed in the reply to comment 2), which is a default parameter for every model.

LocMoFit allows adding a complex model as an additional step after the simpler model, improving the fitting with many parameters.

5. The microtubule section is not really explored all that much as the NPC data, and thus it feels like it was just added to have a second example. It would be nice to have some simulation studies done on this as well (e.g., what is the length of the sections that can be investigated now. It was arbitrarily taken at 1 μm , but how far can this be stretched?)

We admit that we included microtubules to demonstrate the wide applicability of LocMoFit to various and diverse biological structures. We chose 1 μm segments because on this scale most microtubules are curved and cannot be analyzed by cross-sectional profiles. To explore the maximum length that can be fitted with LocMoFit, we chose a curved 5.2 μm microtubule in the experimental data (now included as Fig. 2k-l and discussed on Page 9, Lines 315-316). We show that it can still be fitted using LocMoFit but requires a long runtime (~20 hours). Following the suggestion by the reviewer, we now added an analysis of a long simulated curved microtubule fitted with the model having different arc lengths. The result shows successful fits and a polynomial time complexity (Page 9, Lines 316-317 and ED Fig. 5f-h). To finish the analysis within reasonable runtimes, we now suggest to divide a long segment into multiple ~1 μm segments and fit them separately. This is now mentioned on Page 9, Lines 317-318.

6. In the multi-color section, the model for the structure obtained by (or determined by) LocMoFit is used to do image registration, which should in principle give a better result than when e.g. image correlation is obtained. However, it is not clear from the text how the procedure is actually done: do the authors make use of the multicolour capacity of LocMoFit, or is every channel modelled independently? (Would there be a difference between the two?)

In the multi-color section, we on purpose did not use dual-color registration, but only registered the reference channel. Now we explicitly mention this on Page 10, Lines 381-383. That means, we applied LocMoFit to one channel only to obtain positions and orientations of the structures and used those to register both channels together. We did this to avoid any model bias on the target proteins.

Multi-color registration is also easily possible with LocMoFit (see Fig. 4e-i) and can be used to obtain information about the relative spatial organization of components (e.g., twist between different NPC

components). For generating multi-color protein distribution maps, multi-channel registration could improve the registration accuracy, but we still chose not to do this to avoid the risk of a model bias.

7. Also, the authors mention that the two-color helps when labelling efficiency is low, but this also has its limits: due to the symmetry of the reference structure, it can be rotated around the z-axis in multiple ways. If labelling efficiency is low of the second protein, and only 1 site is active in the images (e.g. images 2 and 3 in Figure 4b, but imagine this all the time), then false conclusions can be made from it. This statement should be nuanced a little, and is mainly useful when there is prior knowledge about the second protein. If nothing is known (new proteins/structures), then this should be considered carefully. In the current example, the reference model is 8-fold rotationally symmetric, and this assumption on symmetry is transferred to the target proteins, resulting in symmetric target protein structures. However, this is not a general limitation of the approach, as asymmetric structures would be fitted with an asymmetric model for the reference to allow for proper registration without imposing symmetry on the target. We now discuss this point on Page 10, Lines 375-378.

8. The comparison between LocMoFit and PERPL is unfair. Using the same model to simulate (noisy) data and fit is always going to give good results (as long as SNR is still reasonable), whereas a method that works on a completely different principle is not going to perform as well. Especially when structures simulated in the data are round, PERPL will have an issue (due to the data reduction into 1 dimension). How do the two compare e.g. the microtubules structure. It should still be in favour of LocMoFit (as it uses the exact model), but the differences are probably not going to be as big. The comparison/discussion for Extended Figure 8 is therefore not really a measure for how good LocMoFit is, but more about in which situations PERPL is not working well.

We agree with the reviewer that the comparison with PERPL mainly shows that LocMoFit outperforms PERPL by a large margin on the structures investigated in the manuscript, but that it does not illustrate how good LocMoFit is by itself. We also agree that this comparison is somewhat unfair because PERPL has a different use case (it shines when segmentation is not easily possible, as it automatically averages over all structures in the FoV). But it is not unfair because of different models, as the reviewer suggests: we implemented the exact same models in PERPL as we used in LocMoFit. Therefore, the simulation did not favor LocMoFit because of the model. The authors of PERPL also used rotationally symmetrical structures (round in a sense) to showcase their analysis. Using a round structure for the comparison should be valid.

The reviewer exactly described how we reasoned the limits of PERPL. We included this comparison between PERPL and LocMoFit at the request of the editor and already then voiced our concerns. As agreed with the editor, we now completely took out the comparison with PERPL.

9. The figures in Extended Data Figure 2 are very misleading and inaccurate. An error is always positive (as it is a distance/angle difference/etc. and therefore does not matter in which direction is it different). As the simulation is a random process, the number of 'negative' differences will be the same as the

number of 'positive' differences and therefore always average out to 0. These errors should be in absolute values (and not be negative and positive) and will be more representative to the true error. We regret that we did not clarify what is shown in the figure but we cannot agree with the reviewer. Errors, defined as the difference between the ground truth value and the data, can be negative. The reviewer seems to refer to average errors, which often are reported as root mean square errors (RMSE) and are indeed positive. When evaluating the performance of an algorithm, two metrics are important: the precision (scatter of the data) and the accuracy (i.e., bias). The RMSE does not distinguish between the two. In ED Fig. 2 we instead report both values: the bias (which is the average of the difference between ground truth and data) and the precision (the standard deviation of the difference, closely related to the error that the reviewer refers to). The bias can be different from 0, e.g., we find a negative bias of ring separation s when labeling efficiency is low; ED Fig. 2a. To clarify this point, we modify the main text (Page 6, Lines 223-225) and the figure caption.

Minor comments on manuscript:

10. Figure 5b is confusing: the authors select the most likely site and then start from there for building the template, but it is not clear if they maximize or minimize the log likelihood (I assume maximizing as they want the highest similarity), and even then, site iv is never the highest or lowest. So I do not understand how they got to site iv as the initial template (as per the caption).

The confusion was caused by a typo: It should be (ii) instead of (iv). We corrected it in Fig. 5b.

11. "Dynamic reconstruction": Dynamic is misleading as these are not really dynamic measurements.). The text mentions pseudotime, and this seems a more appropriate term.

We believe that our formulation "dynamic reconstruction of multi-protein assemblies based on static super-resolution snapshots" clearly describes the concept and is not misleading. We achieve this dynamic reconstruction by a) pseudo-temporal sorting and b) spatial registration. Thus, pseudo-temporal sorting alone is not a synonym for "dynamic reconstruction". To avoid confusion, we explicitly describe what we mean by "dynamic reconstruction" on Page 12, Lines 408-409.

In general, we would like to point out that across biological disciplines there are many examples where "dynamics" are reconstructed from snapshots. For example, the developmental trajectory of immune cells can be constructed from single-cell sequencing data (Bendall et al., 2014), which are static snapshots as the experimental procedure requires cell lysis. Furthermore, when X-ray structures of different functional states of a protein are obtained, intermediate structures can be determined by molecular modeling (e.g., MacKerell et al., The Journal of Physical Chemistry B, 2018).

12. One of the advantages of LocMoFit is that the individual structures can be used for fitting instead of averaging over all the structures. This allows investigating the heterogeneity of the sample, and could thus be of high biological relevance. However, the entire last part of the manuscript is dedicated to model-free averaging to set up a model of the structures, which seems to nullify their previous point. I understand that you do not have to do both of, but it could potentially increase the impact of the paper

if they start off with the model-averaging (something that has already been done for in situ structural biology with SMLM) and then continue with 'looking at the heterogeneity of the sample' (a more novel application).

We chose the current flow because LocMoFit analyzes individual structures. This feature can then be used to perform model-free averaging, which however is only one specific application of LocMoFit. As the results we obtained with the model-free averaging are quite outstanding, we discussed this application in detail.

To ease the concern that this part reduces the impact of the first part, we mention the limitations of model-free averaging and how it can help build a geometric model on Page 12, Lines 437-439.

Major comments on the software:

The software was not intuitive or easy to use and gave many warnings/errors that were not clear how to fix. In addition, it was not clear what results should be obtained and how the results should be accessed. The documentation is very long for SMAP and not easy to navigate and incomplete for LocMoFit.

See below for more detailed comments:

We appreciate that the reviewer made an effort to test our software extensively and to note down warning/error messages. LocMoFit has been designed to be general and flexible and is therefore a complex tool. We now spent extra effort in making the user experience simpler. We also want to comment on the 'ease of use' that we claim, which is of course not an absolute measure. Software like Chimera, Pymol, SMAP, LocMoFit, and many others, allow biologists to analyze their data without extensive programming experience and without implementing their own algorithms. Thus, they are relatively easy to use. But they all require some dedication and time to learn (often several days), which however is still little compared to the time it takes to take and analyze the data.

The SMAP tutorial is very long as SMAP has extensive functionality. In the revised version, we now extended the LocMoFit tutorial to include all parts from SMAP that are relevant, so that a user does not have to consult the SMAP tutorial to follow the examples. We now restructured the tutorial and included outcome images of major steps to make it complete and easier to navigate. As with SMAP, we will support any future user directly with their questions and bug reports.

We tested LocMoFit and the tutorials extensively again to make sure that they can be followed without any errors.

GUI in Matlab 2021b

- Opening the ROI manager loads partially outside the screen (15.4 inch laptop @ 1920 x 1080 pixels) Assumably because size is given in absolute pixel values, and not in relative pixel values.

- Lots of warnings of the same kind (when clicking on 'redraw' or 'all' in the ROI manager):

1. Warning: Single-selection 'listbox' control requires that 'Value' be an integer within Character vector range

Control will not be rendered until all of its parameter values are valid

2. In gui/SEExploreGui/redrawall (line 246)

In gui.SEExploreGui>redrawsiteall_callback (line 959)

Warning: Single-selection 'listbox' control requires that 'Value' be an integer within Character vector range

Control will not be rendered until all of its parameter values are valid

3. Many many GUI warnings that won't all be copied here.

These file-specific warnings relate to SMAP, but not to LocMoFit specifically. In the example sml file we provide, some additional descriptive annotations of sites are included and cause the warnings, which do not affect the analysis. We now removed this unnecessary information to avoid these. SMAP produces many warnings that can be safely ignored. We are very responsive to bug reports, both by email and Github issues. We now added a function that detects small screens and adjust the size and position of the ROI manger accordingly. We reproduced all of the following reported errors and have them fixed now.

- Came across one error as well that could not be reproduced (also not sure if that is general SMAP functionality, or LocMoFit):

1. Intermediate dot '.' indexing produced a comma-separated list with 0 values, but it must produce a single value when followed by subsequent indexing operations.

Error in imline/getPosition (line 150)

```
pos = obj.api.getPosition();
```

Error in gui.SEExploreGui>plotline (line 844)

```
posin=obj.hlines.(posfield).getPosition;
```

Error in gui.SEExploreGui>anglebutton_callback (line 871)

```
plotline(obj,'rotationpos',obj.guihandles.angle);
```

Related documentation

Error while evaluating UIControl Callback.

This error is caused by previously saved information in the example data. The information is not relevant to the tutorials and we now removed it to prevent the error.

- Error when clicking on 'evaluate current ROI':

1. Dot indexing is not supported for variables of this type.

Error in interfaces.SEEvaluationProcessor/getLocs (line 108)

```
sx=obj.site.image.rangex;
```

Error in ROIManager.Evaluate.generalStatistics/run (line 26)

```
locs=obj.getLocs({'locprecnm','PSFxn','xnm','ynm','phot','bg','numberInGroup'},'layer',k,'size',roisizeh);
```

Error in interfaces.SEEvaluationProcessor/evaluate (line 83)

```
out=obj.run(p);
```

Error in gui.SEEvaluationGui>evaluatesite (line 227)

module.evaluate(site);

Error in gui.SEEvaluationGui>preview_callback (line 211)

evaluatesite(obj,obj.SE.currentsite,1)

Error while evaluating UIControl Callback.

This error can be avoided by selecting any site first. To avoid this confusion, now we included a message 'Nothing is evaluated. Please click on one site in the ROI list first' that will be displayed in the MATLAB Command Window when no site is selected.

- Not clear from the tutorial how to perform a composite model. It stops at step 8.4, and could not really figure out how to advance from there. When some things were tried, errors were thrown, so probably not following the correct steps. This ties in with the fact that the GUI is not really intuitive to use, and without a tutorial, it is difficult to have anything working properly.

Step 8.4 was indeed the end of the tutorial. We now extended and restructured all the tutorials, including this one, to make the workflows clear by explaining the purpose of each main step.

To make LocMoFit easy to use and learn, we now added in the GUI a help button (in the top-right corner) to bring the users to corresponding help pages in the documentation according to the current GUI tab where the user is. We also simplified the GUI, especially the model loading functionalities. Now, built-in models can be loaded with a drop-down menu so that the users do not have to search for specific files. The documentation of built-in models is now connected to the models themselves in the GUI through an information button next to the drop-down menu.

- Simulating data according to the documentation does NOT work.

Cannot continue with the simulations. This is from a clean install, clean settings, Matlab 2021b

1. Error using LocMoFit/setModel (line 163)

Invalid: You are adding a 3D model for fitting 2D data.

Error in LocMoFit/addModel (line 224)

obj.setModel(model,lastMod+1);

Error in ROIManager.Evaluate.LocMoFitGUI>initmodel (line 1266)

fitter.addModel(geoModeltemp);

Error in ROIManager.Evaluate.LocMoFitGUI>loadmodel_callback (line 1205)

initmodel(obj, modelnumber);

Error while evaluating UIControl Callback.

This error occurs in the absence of localization data. We now changed the code and removed this requirement.

2. Click Connect to LocMoFit and check LocMoFitGUI in the new window. -> Cannot find the 'connect to LocMoFit' button anywhere.

This issue is related to the last point. This is not an issue anymore as we now removed the requirement for localization data.

Reviewer #2:

Remarks to the Author:

The work by Wu and coworkers demonstrates an MLE approach to obtaining structural information from SMLM data. The authors use a combination of ground-truth models and the well characterised nuclear pore complex to validate their methodology. They eventually apply it to develop and benchmark a kinetic growth model for Clathrin-mediated endocytosis. The authors have open-sourced their code and is available as integration to SMAP developed by the Ries Lab as well as an API for third-party integration.

The simulations are extensive, the maths is solid, the manuscript is extremely well written and, overall, the work represents a huge leap forward for which I have to congratulate the authors. However, there are important shortcomings that have to be addressed before I can recommend this for publication. We would like to thank Reviewer 2 for the positive comments and useful suggestions.

1, the vision of the authors is to enable in-situ structural biology using SMLM. And, I agree that SMLM is one of few (perhaps the only method) that has the potential to analyse the structure of heterogenous macromolecular complexes inside the cell. However, talking about protein structure, measurements not the angstrom scale are not useful from a structural biology perspective; they are only useful in constructing kinetic growth models as nicely demonstrated by the authors. To what extent is the phrase 'in-situ structural biology' supported in light of the following:

This criticism is not related to LocMoFit, but to our definition of 'in situ structural biology'. We, and parts of the community, work with a somewhat different definition of 'structural biology' that includes not only the precise arrangement of amino acids in a protein, but in general the spatial relationships among proteins in complexes (mentioned on Page 2, Lines 33-34).

For the classical structural biology techniques (crystallography, electron microscopy), indeed a resolution not in the angstrom range (which is required to fit the primary sequence into the electron densities and to generate atomic models) is not very useful, as this prevents the identification of protein domains in a protein or of a protein in a complex. The situation is different in super-resolution microscopy, where the fluorescence label can be identified with highest contrast and with moderate (1-10 nm) precision.

This information on intermediate (nanometer) spatial scales allows probing the relative arrangement of proteins, which in principle can be invaluable information to, e.g., assign protein identities to unassigned electron densities, to determine the approximate position of flexible proteins or to investigate the arrangement of proteins in complexes that are too irregular or disordered to be solved with structural techniques (e.g., the yeast endocytic machinery). When structural models of the parts are known, the superresolution information can help probing the conformation of a complex in the cell.

We now discuss this point on Page 2, Lines 34-36.

a, none of the measurements in the paper, aside from the simulations, are reported with angstrom resolution (including those performed on the highly conserved nuclear pore complex).

As the size of the fluorescence label is >1 nm, optical techniques can never directly reach angstrom localization precisions. But we would like to point out that the parameters that we extract with LocMoFit can in principle have a precision in the Angstrom range, as they are based on fitting numerous localizations. In ED Fig. 2 many of the fitting parameters reach precisions below the localization precision. Also, when pooling many measurements (analogous to averaging in cryo-EM), the average of a specific parameter can be calculated with a much higher precision, the standard error of the mean (SEM). For example, for experimental data on Nup96 (Fig. 2d) we find a value for the radius of $r = 53.4 \pm 0.04$ nm (mean \pm SEM), i.e., a SEM of 0.4 Angstrom. We now add these Angstrom precision values to Page 7, Line 267 but want to point out that they might not be very useful, because they might be smaller than the biological variability or a bias caused by experimental imperfections.

b, this tool is most useful, as suggested by the authors, when applied to study the structure of heterogenous complexes which escape particle averaging by alternative techniques (e.g. CryoEM). However, the structure of the analysed complexes is not studied, one-by-one, at high resolution - instead they are either averaged (through the model free routine) or pooled in samples/bins for further analysis.

In our manuscript, we indeed analyze every structure individually, and from each structure obtain a set of parameters that describes the individual structure. Compared to looking at averages or pooling, this allows studying for instance the variability of individual parameters and with it the variability of the whole structures (e.g., the radius of the nuclear pore complex) and to look at correlations among parameters (e.g., the curvature and the closing angle in mammalian endocytosis, ED Fig. 9 which allowed us to investigate the mechanism of coat formation), something that is only possible when analyzing individual particles. Thus, we disagree that we only look at averages or pooled structures.

None of the above is a deterrent to publishing this beautiful work, but it is important to set the expectations right from the beginning - that this is a tool for analysing the nanoscale structure of protein complexes at the ensemble level as otherwise, for a structural biologist, this tool could image protein structures with angstrom resolution which is not the case.

We decide to continue to use our definition of 'in situ structural biology'. To clarify what it covers, we include a discussion on Page 2, Lines 33-34.

2, the simulations in extended figure 2 are performed with one parameter changed at a time which is not informative for dictating the optimal experimental conditions that would yield measurements with the highest accuracy. To the non-expert reader, the reported method works well even when labelling efficiency is 30% - where in reality, that low labelling efficiency can be accompanied by reductions in the localisation precision, amplification in the background, etc.

ED Fig. 2 gives an overview of how the accuracy changes with respect to certain parameters. Varying all parameters against all others would generate too many graphs. To follow the reviewers suggestion, we now extend ED Fig. 2 and update ED Fig. 1 and 3 to cover different fluorophore brightnesses (i.e., average localization precisions) corresponding to the typical PALM, STORM and DNA-PAINT modality against other parameters that determine the data quality and discuss this point on Page 6, Lines 226-230. Note that the simulations are now based on the experimental PSF and thus are more realistic. In practice, however, the user can easily determine many of the parameters in their experiment (photons, background, blinking rates,...) and perform these simulations themselves to investigate the performance of the fit under their actual experimental conditions. We now include a tutorial on how to generate such simulations in the LocMoFit documentation.

Additionally, the very important parameters, linker length, density of localisations and low- and high-frequency drift are not simulated despite the fact that they could worsen spatial resolution by several tens of nano-metres.

The reviewer points out important additional parameters. We can already consider all of those in the simulations:

a. Linker length: to simulate linker lengths is not trivial. If the fluorophore is free to rotate and to explore the space allowed by the linker, this happens on a time scale much shorter than the camera exposure time and the positions are averaged out. This would correspond to a systematic offset of the fluorophores, which will affect the fitting and cannot easily be corrected for. If instead the fluorophore can only explore small regions, and these are random, then we will get a random displacement for each fluorophore. This was included in the simulations. We now clarify this point on Page 6, Lines 229-230 and Page 21, Line 805-808.

b. The density of localizations corresponds in our case to the labeling efficiency (together with a density of proteins in the model for continuous models), which we investigated (see Page 21, Line 805 and ED Fig. 1c and 2a) and can simulate.

c. Drifts and vibrations lead to additional displacements of the localizations. We simulate them by random displacements of the localizations (see Page 21, Line 805-808). Random linker positions and drifts/vibrations are incorporated into the fitting model by the parameter 'extra uncertainty ϵ' '. In the previous version, we did not explicitly mention the differences between the linkage error, random displacement, and extra uncertainty. Now we clearly differentiate them.

The authors are asked to simulate these parameters alone and combined to guide the end-user in understanding the limitations of the method and/or choosing the right set of parameters that would provide them with an acceptable error in their quantitative structural measurements. This in addition to showing the reader the correlation between the error in these measurements and the sample size (i.e. number of sites/structures).

The sample size is not relevant for the analysis of individual structures, but for the model-based and model-free averaging, and we include simulations showing the impact of the number of structures (ED Fig. 10a-d) and a discussion on Page 13, Lines 455-456.

As described above, most importantly, the simulation engine enables any user to easily generate simulations for their experimental conditions. Now we provide an improved tutorial the LocMoFit documentation to understand the limitations of their workflow and the expected precision and accuracy of their parameter estimates.

3, some readers will be familiar with the work of Szymborska et al. Science, 2013 where the radii of the different nucleoporins was reported with angstrom accuracy using SMLM (see figure 1F). It is important for the authors to establish the correlation between the accuracy of their, and the reported, measurements commenting on the discrepancy between both as well as the role of data curation in improving the accuracy of measurements.

In the work of Szymborska et al., the average radius of thousands of nuclear pore complexes is reported, and not that of individual NPCs. The corresponding error is the standard error of the mean (SEM) and is in the Angstrom range (see reply to comment 1a, this reviewer). If we look at the distributions of the different parameters, we can also calculate the mean and SEM. For the radius of Nup96 (Fig. 2d) we find a value of $r = 53.4 \pm 0.04$ nm (mean \pm SEM), i.e., sub-angstrom precision. The value in Szymborska et al on Nup96 (shown in their Fig. 2E, instead of Fig. 1, which shows the authors' workflow based on another nucleoporin Nup133) reported a value of $r = 59.0 \pm 0.1$ nm. The difference in the mean value can be due to a) the different analysis method (Szymborska: 1. rendering as a pixelated image, 2. Registration of images, which might introduce additional inaccuracies, 3. fitting of the radius to the 1D radial distribution. This work: direct fitting of each particle, then averaging of the individual measurements), b) labels of different size (Szymborska: indirect immunolabeling, this work: SNAPtag), c) 2D (Szymborska) vs 3D (this work) measurement and analysis, d) additional systematic errors such as imprecise calibration of the pixel size. This is now discussed on Page 7, Lines 263-265.

For data curation for the NPC, we excluded structures that do not appear to be rings, that have too few localizations, and that have one ring completely unlabeled. Examples are now included as ED Fig. 4f. We also include the parameter distributions without the exclusion as ED Fig. 4g-i to show the effect of curation. This point is mentioned on Page 27, Lines 1036 and 1047-1048.

4, the mathematical models used in this work are relatively simple. Complexes, particularly those of intrinsically-disordered proteins, assemble into complex geometries that can be modelled with discrete models. The authors demonstrate discrete modelling in the context of model scoring of the conserved NPC structures, but not for more complex heterogenous structures. The authors are asked to perform simulations with heterogenous structures demonstrating the validity of true discrete modelling (i.e. flexible line segments) and model scoring.

LocMoFit is not developed for modeling entirely random geometries (mentioned on Page 5, Lines 186-188) and overlapping paths. We do agree that a flexible structure obeying a certain geometry (e.g.,

flexible line segments, as the reviewer suggests) is a good example to further showcase LocMoFit. For this, we include examples of line segments with different numbers of segments and their scoring (AICC) after fitting in ED Fig. 8. We discuss this on Page 10, Lines 350-354 and Page 21, Lines 793-797 (Methods).

5, it is important to establish the supremacy of MLE compared to the widely used LS/NLS in terms of accuracy or at least both approaches require the same amount of time to execute as with the MLE it requires 10 seconds per site which projects to several tens of hours for tens of thousands of structures. We chose MLE for its ease of implementation. We do not see a straightforward way to formulate the fitting problem (fitting a geometric model to point clouds) using LS/NLS approaches. In principle, we could render both data and the model as images and perform fitting by minimizing the residual sum of squares of the difference. However, this would be computationally more expensive, would introduce the pixilation bias, and would still fit individual bright pixels with a continuous model. Alternatively, one could calculate the distance of each localization to the nearest point in the model and minimize this, but except for very simple geometries (e.g., a circle), this is rather complex.

Reviewer #3:

Remarks to the Author:

The manuscript presents a method to produce 3D structural models of proteins and protein complexes from localization coordinates - such as those obtained from single-molecule localization microscopy (SMLM). In particular, the method enables to fit arbitrary models to localization coordinates and to extract quantitative model parameters with error estimates. In addition, it is able to derive measures of quality of fitness that can be used to select which structural model is more appropriate. Also, it can rely on 2-color imaging to re-assemble multi-color protein distribution maps. Finally, the authors demonstrate the ability of their method to derive model-free averaging.

The manuscript is well written, and the method clearly described and validated. However, a number of issues need to be resolved to ensure that this method becomes of widespread interest to a community interested in using SMLM to reconstruct the 3D structures of protein complexes.

We also would like to thank reviewer 3 for the positive and constructive comments.

1, In their approach for multicolor protein distribution maps, the authors image 2 proteins simultaneously and use one as a reference structure. This approach is fine when there is no variation in the relative spatial distribution between proteins in a given structure. However, it is reasonable to imagine situations where this assumption will not be valid, and will lead to erroneous results. For instance, if there are to conformations with the same reference distribution but different distributions for target 1 and target 2, then the final model will contain 'assembled' distributions for these targets that will be wrong as the reconstructed 3-color structure actually does not exist in nature.

The reviewer is correct. As any averaging method without classification, also LocMoFit's protein distribution maps rely on identical underlying structures. Otherwise, an average of the distribution of

the conformations will be calculated, and we now add this point on Page 10, Lines 384-386. In the future, one could consider adding a classification step that is also based on the target proteins.

2, The software has been tested and validated in pretty stable and well known structures, mainly the NPC. It would be important to test how the method will perform in more realistic cases, where there is structural heterogeneity, different complex compositions, or where there is flexibility. At the end this package will be most useful if it can be applied to these kind of scenarios, rather than just to 'perfect' model systems that are interesting to a limited audience.

Although the NPC is the major example, we also provide other examples which show strong structural heterogeneity: micrometer-long microtubule segments and the 5.2- μm segment, each curved in their own way (Fig. 2g-l); clathrin coats in mammalian cells that are flexible, i.e. they can change in sizes and curvature, corresponding to different shapes (flat, dome-like, spherical) (ED Fig. 9); endocytic proteins in yeast cells with different relative distributions and abundances (i.e., complex compositions, Fig. 4e-i). Thus, we believe that we showcased LocMoFit on realistic cases, where there is structural heterogeneity, different complex compositions, and where there is flexibility.

3, This last point, is in fact, a main claim of the method "As the fitting is performed on individual structures without averaging (e.g., Fig. and Extended Data Figure 7), this will help investigating the vast majority of cellular structures that are heterogeneous and complex.". However, I don't see where this has been tested or validated. In absence of this advantage, where is the gain with respect to cryo-EM? As we detail in the previous reply, we do showcase and validate LocMoFit on individual heterogeneous and complex structures. Note the former ED Fig. 7 is now ED Fig. 9.

In general, LocMoFit in combination with SMLM has the following advantages compared to cryo-EM:

a. Analysis of individual structures: SMLM determines the precise positions of single fluorophores. This happens with ultra-high contrast, but with a position error determined by the localization precision. This high contrast allows analyzing individual particles without averaging, as we showcase with LocMoFit. The distribution of the fitting parameters allowed us to directly investigate biological heterogeneity (shapes of clathrin coats, radius variability of the NPC), which is more challenging to do with averaging-based cryo-EM.

b. Protein distribution maps (template based or template free): due to the high contrast of specific fluorophores, averages are meaningful even if the underlying structures are not exactly the same, they then report average distribution maps. Corresponding average electron densities on the other hand would be largely useless, as they would not have the resolution to identify specific protein domains. Thus, the SMLM-based protein distribution maps would allow determining approximate positions of proteins in a complex that are not visible in the EM maps because they are too small or too flexible, an advantage even for rather homogenous structures like the NPC. For large molecular machineries like the endocytic machinery, atomic models likely do not exist because of the stochastic interactions of the proteins, here LocMoFit with SMLM can still reveal average distribution of proteins, and after pseudo-temporal sorting, even reconstruct dynamic protein distribution maps.

c. In the future, there is the distinct possibility that SMLM and new flavors like MINFLUX can perform measurements directly in the living cell. LocMoFit could analyze such data to result in truly dynamic structural information in the cell.

The disadvantage is of course the limited resolution, determined in the best case by the size of the fluorescent label and potential effects of the label on protein function, which must be excluded with controls.

4, I would have liked to see testing of the performance of the algorithm under different realistic experimental situations. For instance: how many particles or sites are necessary to converge? is this experimentally feasible? how does the model perform with lower efficiencies of detection?

This comment is related to the model-free averaging section. As we present high-quality template-free averages (Fig. 5, Supplementary Movie 3, ED Fig. 10a-d) based on experimental data, we already show that this approach is experimentally feasible.

For addressing the minimal required number of NPC particles, we calculate protein distribution maps based on random subsets with varying size (see the new ED Fig. 10a-d, and the relevant text on Page 13, Lines 455-456). For addressing the minimal required efficiency of detection (i.e., labeling efficiency), we simulated NPC particles with different labeling efficiency and compared their calculated averages with the model used for simulations (see the new ED Fig. 10e-h, and the relevant text on Page 13, Lines 456-458).

5, The code is provided in MATLAB. If I understand correctly, this would require potential users to pay for a MATLAB license to use it. If this is the case, it would be important to provide a way in which users can bypass this, such as with a compiled dynamic library.

As LocMoFit is integrated into SMAP, the compiled version of SMAP can be run without requiring a MATLAB license (this is now mentioned on Page 28, Lines 1082-1084). LocMoFit works in the compiled version with the limitation that own new models cannot be added. But the multiple pre-defined models can be used and combined freely. All examples and tutorials can be followed also with the compiled version.

6, Literature on the previous methods used for particle averaging are not exhaustive and only seem to focus on those written by the author.

We apologize for only focusing on the dedicated frameworks and for unintentionally omitting other important contributions to particle averaging of super-resolution microscopy data.

We now include the following citations on Page 2, Lines 66-68 but are also open to additional suggestions:

- Löschberger et al., Journal of Cell Science, 2012 (2D averaging of NPCs imaged by STORM).
- Van Engelenburg et al., Science, 2014 (Single-cluster averaging of ESCRT proteins at HIV budding sites by iPALM imaging).

- Szyborska et al., Science, 2014 (Determination of the Y-complex orientation within the NPC by STORM imaging and radial averaging).
- Schnitzbauer et al., Nature Protocols, 2017 (Particle averaging of DNA origami imaged by DNA-PAINT).
- Salas et al., PNAS, 2017 (Single-particle reconstruction to extract 3D information from 2D super-resolution images obtained by DNA-PAINT).
- Sieben et al., Nature Methods, 2018 (Multi-color single-particle reconstruction of the human centriole by high-throughput SMLM and analysis pipeline).

7, It is surprising to see a comparison between LocMoFit and PERPL in the Discussion. This comparison should be instead within the results section.

As mentioned in the reply to point 8 of reviewer 1, we take out the comparison with PERPL completely.

Decision Letter, second revision:

9th Sep 2022

Dear Jonas,

Thank you for submitting your revised manuscript "Maximum-likelihood model fitting for quantitative analysis of SMLM data" (NMETH-A47609C). It has now been seen by the original referees and their comments are below. The reviewers find that the paper has improved in revision, and therefore we'll be happy in principle to publish it in Nature Methods, pending minor revisions to satisfy the referees' final requests and to comply with our editorial and formatting guidelines.

With regards to the reviewer feedback, we simply ask that you add the suggested discussion from the rebuttal into the discussion section.

TRANSPARENT PEER REVIEW

Nature Methods offers a transparent peer review option for new original research manuscripts submitted from 17th February 2021. We encourage increased transparency in peer review by publishing the reviewer comments, author rebuttal letters and editorial decision letters if the authors agree. Such peer review material is made available as a supplementary peer review file. Please state in the cover letter 'I wish to participate in transparent peer review' if you want to opt in, or 'I do not wish to

participate in transparent peer review' if you don't. Failure to state your preference will result in delays in accepting your manuscript for publication.

Thank you again for your interest in Nature Methods Please do not hesitate to contact me if you have any questions.

Sincerely,
Rita

Rita Strack, Ph.D.
Senior Editor
Nature Methods

ORCID

Reviewer #1 (Remarks to the Author):

The authors have addressed the concerns raised and the manuscript is now suitable for publication. I only have one additional suggestion. The authors address the method's limitation in terms of assessing the correctness of the model in the response letter but a discussion on this should also be included in the manuscript.

Reviewer #2 (Remarks to the Author):

In their revised manuscript, the authors discuss what they mean by in situ structural biology (and its usefulness to the field of structural biology), explain the discrepancy in the mean error and absolute

radius measurement between their work and that published earlier, appropriately justify their simulation parameters (and extended them as suggested), and develop an extensive tutorial document for LocMoFit.

This is an important development that I, personally, have been eagerly waiting for. I, therefore, thank the authors for this extensive effort and recommend the publication of their revised manuscript without reservations.

John S. H. Danial

Reviewer #3 (Remarks to the Author):

The authors have successfully addressed my concerns. Congratulations on this excellent development that will likely further stimulate the use of SMLM in this field.

Final Decision Letter:

14th Oct 2022

Dear Jonas,

I am pleased to inform you that your Article, "Maximum-likelihood model fitting for quantitative analysis of SMLM data", has now been accepted for publication in Nature Methods. Your paper is tentatively scheduled for publication in our January print issue, and will be published online prior to that. The received and accepted dates will be Nov 19, 2021 and Oct 14, 2022. This note is intended to let you know what to expect from us over the next month or so, and to let you know where to address any further questions.

Once your paper is typeset, you will receive an email with a link to choose the appropriate publishing options for your paper and our Author Services team will be in touch regarding any additional information that may be required.

Please note that Nature Methods is a Transformative Journal (TJ). Authors may publish their research with us through the traditional subscription access route or make their paper immediately

open access through payment of an article-processing charge (APC). Authors will not be required to make a final decision about access to their article until it has been accepted. [Find out more about Transformative Journals](https://www.springernature.com/gp/open-research/transformative-journals)

Your paper will now be copyedited to ensure that it conforms to Nature Methods style. Once proofs are generated, they will be sent to you electronically and you will be asked to send a corrected version within 24 hours. It is extremely important that you let us know now whether you will be difficult to contact over the next month. If this is the case, we ask that you send us the contact information (email, phone and fax) of someone who will be able to check the proofs and deal with any last-minute problems.

If, when you receive your proof, you cannot meet the deadline, please inform us at rjsproduction@springernature.com immediately.

Once your manuscript is typeset and you have completed the appropriate grant of rights, you will receive a link to your electronic proof via email with a request to make any corrections within 48 hours. If, when you receive your proof, you cannot meet this deadline, please inform us at rjsproduction@springernature.com immediately.

Once your paper has been scheduled for online publication, the Nature press office will be in touch to confirm the details.

Once your paper has been scheduled for online publication, the Nature press office will be in touch to confirm the details.

Content is published online weekly on Mondays and Thursdays, and the embargo is set at 16:00 London time (GMT)/11:00 am US Eastern time (EST) on the day of publication. If you need to know the exact publication date or when the news embargo will be lifted, please contact our press office after you have submitted your proof corrections. Now is the time to inform your Public Relations or Press Office about your paper, as they might be interested in promoting its publication. This will allow them time to prepare an accurate and satisfactory press release. Include your manuscript tracking number NMETH-A47609D and the name of the journal, which they will need when they contact our office.

About one week before your paper is published online, we shall be distributing a press release to news organizations worldwide, which may include details of your work. We are happy for your institution or funding agency to prepare its own press release, but it must mention the embargo date and Nature Methods. Our Press Office will contact you closer to the time of publication, but if you or your Press Office have any inquiries in the meantime, please contact press@nature.com.

Nature Portfolio journals [encourage authors to share their step-by-step experimental protocols](https://www.nature.com/nature-research/editorial-policies/reporting-standards#protocols) on a protocol sharing platform of their choice. Nature Portfolio's Protocol Exchange is a free-to-use and open resource for protocols; protocols deposited in Protocol Exchange are citable and can be linked from the published article. More details can be found at www.nature.com/protocolexchange/about.

Please note that you and any of your coauthors will be able to order reprints and single copies of the issue containing your article through Nature Portfolio 's reprint website, which is located at <http://www.nature.com/reprints/author-reprints.html>. If there are any questions about reprints please send an email to author-reprints@nature.com and someone will assist you.

Best regards,
Rita

Rita Strack, Ph.D.
Senior Editor
Nature Methods